

# Lime and zinc application influence soil zinc availability, dry matter yield and zinc uptake by maize grown on Alfisols

**Sanjib K. Behera**[a,*]**, Arvind K. Shukla**[b]**, Brahma S. Dwivedi**[c]**, Brij L. Lakaria**[b]

*ICAR-Indian Institute of Oil Palm Research, Pedavegi, West Godavari District, Andhra Pradesh 534450, India*
*ICAR-Indian Institute of Soil Science, Nabibagh, Berasia Road, Bhopal, Madhya Pradesh 462038, India*
*ICAR-Indian Agricultural Research Institute, Pusa, New Delhi, 110012, India*

*Corresponding author: sanjibkumarbehera123@gmail.com (S. K. Behera), ICAR-Indian Institute of Oil Palm Research, Pedavegi, West Godavari District, Andhra Pradesh 534450, India

ABSTRACT

Zinc (Zn) deficiency is widespread in all types of soils of world including acid soils affecting crop production and nutritional quality of edible plant parts. There is, however, limited information available regarding effects of lime and farmyard manure (FYM) addition on soil properties, phyto-available Zn by different extractants, dry matter yield, Zn concentration and uptake by maize (*Zea mays* L.). Green house pot experiments were carried out in two acid soils to study the effect of five levels of lime (0, 1/10 lime requirement (LR), 1/3 LR, 2/3 LR and LR), three levels of Zn concentration (0, 2.5 and 5.0 mg Zn kg$^{-1}$ soil) and two levels of FYM (0 and 10 t ha$^{-1}$) addition on soil pH, EC and OC content, phyto-available Zn in soil and dry matter yield, Zn concentration and uptake by maize plant grown up to 60 days. Application of lime and FYM improved soil pH. Increased level of lime application reduced Zn extracted by DTPA, Mehlich 1, Mehlich 3, 0.1 N HCl and ABDTPA extractants. However, application of FYM along with lime improved Zn extraction. The amount of Zn extracted by different extractants followed the order DTPA-Zn < ABDTPA-Zn < Mehlich-1 Zn < 0.1 M HCl. Lime rate of 1/3$^{rd}$ LR was found to be optimum as dry matter yield of maize increased significantly with lime application up to 1/3$^{rd}$ LR in soils of both the series and decreased subsequently. Addition of FYM with and without lime increased dry matter yield.



Application of Zn up to 5.0 mg kg$^{-1}$ to soil increased dry matter yield with and without FYM
application in soils of Hariharapur series. Addition of higher doses of lime significantly
reduced Zn concentration in maize crop grown in soils of both the series. Mean Zn uptake
values were at par for no lime, 1/10$^{th}$ LR and 1/3$^{rd}$ LR with and without FYM application and
it was significantly higher than Zn uptake by 2/3$^{rd}$ LR and LR treatments. However, FYM
application improved Zn uptake by maize crop.  Zn extracted by different extractants like
DTPA, ABDTPA, Mehlich 1, Mehlich 3 and 0.1 M HCl was positively and significantly
correlated amongst themselves and with dry matter yield, Zn concentration and Zn uptake by
maize.
*Keywords:*  Alfisols, Lime, Farmyard manure, Zinc, Maize
**1. Introduction**
Globally, zinc (Zn) deficiency is the most widespread micronutrient deficiency problem
resulting in reduced crop production and nutritional quality of edible plant parts (Cakmak,
2002).   It is more prevalent in cereal growing areas and nearly 50% of world's cereal
growing areas are having soils with low plant-available Zn. It has also been reported in
almost all countries (Alloway, 2008) including India in different soil types (Takkar, 1996;
Shukla et al., 2014).  It is commonly prevalent in high pH calcareous soils (Katyal and Vlek,
1985), and leached, heavily weathered and sandy acid soils with low organic matter content
(Rautaray et al., 2003; Behera et al., 2011).
Soil acidity is a serious problem affecting crop production across the world including
India which is having 34.5% of arable land with acid soils (Maji et al., 2012). Ameliorating
acid soils with suitable amendments and proper nutrient especially Zn management in acid
soils are areas of concern for obtaining higher crop yield. Amelioration of acidic soils is
beneficial to plant growth because it improves soil pH and replenishes nutrients (Moon et al.,



2014). Application of liming material is an effective method for amelioration of acid soils
(Ponnette et al., 1991; Quoggio et al., 1995). Lime is normally oxides, carbonates and
hydroxides of calcium or magnesium. There are about four types of lime viz., quicklime
(CaO), slaked lime (Ca(OH)$_2$), limestone (CaCO$_3$) and dolomite. Application CaCO$_3$ to acid
soils reduced soil acidity, improved basic cations status and significantly increased the yields
of crops grown on Ultisol (Cifu et al., 2004). However, adoption of standard recommendation
of lime requirement (LR) for different groups of acid soils is difficult for farmers, which is
uneconomical and unsustainable. Therefore, lower doses of LR like 1/10[th], 1/3[rd] and 2/3[rd] of
LR are applied by the farmers. There is dearth of information regarding the application of
different doses of lime on Zn availability in acid soils.

66        Soil pH and organic matter content are the most important soil factors affecting

phyto-availability of Zn in soil (Suman, 1986; Lindsay, 1992). Increased soil pH due to
addition of lime can influence availability of Zn in soil by altering its equilibrium (Verma and
Minhas, 1987).  Higher level of soil pH results in reduced extractable Zn content due to
increased adsorptive capacity, formation of hydrolyzed forms of zinc, chemisorption on
calcium carbonate and co- precipitation in iron oxides (Cox and Kamprath, 1972). Available
organic materials such as farmyard manure (FYM) are generally used by the farmers along
with chemical fertilizers because it improves soil physical, chemical and biological properties
(Nambiar, 1994). Addition of organic matter to soil results in enhanced microbiological
activity which adds complexing agents as well as influences the redox status of soil.
According to Moody et al. (1997) higher levels of organic matters enhance Zn availability by
increasing exchangeable and organic fractions of Zn and reducing oxide fractions of Zn. The
effect of addition of organic matter on Zn availability in soils has also been reported by
different workers (Murthy, 1982; Ghanem and Mikkelsen, 1987). But the information



regarding influence of addition of lime with and without FYM to acid soils on Zn availability
in soil and Zn concentration and Zn uptake by crops is limited.
Appropriate soil tests for plant available metal are not yet available for all types of
agricultural soils around the world. However, extractants like diethylene triamine penta
acetic acid (DTPA), ethylene diamine tetra acetic acid (EDTA), hydrochloric acid,
ammonium bicarbonate-DTPA (ABDTPA) , Mehlich 1 and Mehlich 3 are used for
extraction of plant available Zn from soils (Alloway, 2008). But DTPA extractant is the most
widely used. The DTPA soil test was originally developed to categorize near-neutral and
calcareous soils with insufficient plant available Zn to support maximum yield of crops
(Lindsay and Norvell, 1978). But the same has been used for acid soils also for extraction of
plant available Zn. According to O'Connor (1988), whenever one strays from the original
design of the test, one should be aware of the possible consequences and pass that awareness
on to others. Based on correlation among the extracted Zn by different extractants and with
soil properties, Behera et al. (2011) reported the usefulness of DTPA, Mehlich 1, Mehlich 3,
0.1 N HCl and ABDTPA extractant for extraction of plant available Zn in acid soils of India.
But there is scanty information available regarding the relationship of extracted Zn by
different extractants with Zn concentration and uptake by crop plants. Therefore, the present
study was carried out to evaluate the influence of lime and FYM addition on soil pH, EC and
OC content, extracted Zn as extracted by different extractants and dry matter yield, Zn
concentration and uptake by maize (*Zea mays* L.) and to analyze the relationship amongst
them.
**2. Materials and methods**
*2.1 Soil characteristics and methods of soil analysis*

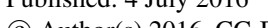



The bulk surface (0-15 cm depth) soils collected from Hariharpur series (Oxic Haplustalfs)
(Bhubaneswar, India) and Debatoli series (Udic Rhodostalfs) (Ranchi, India) were used in the
study. These soils were representative typical soils found in India.  Selected characteristics of
these soils are given in Table 1. The collected soil samples were air dried and stone and
debris were removed and then ground to pass a 2 mm sieve. The samples were then stored for
subsequent analysis. Soil properties like pH and EC were determined done on 1: 2.5 soil
water ratio (w/v) suspension using pH meter and EC meter   following half an hour
equilibrium (Jackson, 1973). Soil organic carbon (OC) content was estimated by chromic
acid digestion-back titration method (Walkley and Black, 1934). The clay, silt and sand per
cent of soils were determined by hydrometer method (Bouyoucos, 1962). Calcium carbonate
($CaCO_3$) content was determined by rapid titration method (Puri, 1930) and cation exchange
capacity (CEC) by neutral normal ammonium acetate method (Richards, 1954). Lime
requirement (LR) of the soil was estimated by extractant buffer method (Shoemaker et al.,
1961). The plant available Zn in soils was extracted by DTPA method (Lindsay and Norvell,
1978). After drying of FYM   at 70 $^o$C for 24 h followed by grinding to pass through 20 mesh
sieve, one gram of ground FYM was dry-ashed at 450 $^o$C for 2h. Ashed samples were
extracted using 0.5 N HCl. Zn concentration was determined in filtered extracts. The OC, N,
P and K concentrations in FYM were estimated by appropriate methods (Jackson, 1973). The
OC, N, P, K and Zn content in FYM (on dry weight basis) were 0.12% 0.48%, 0.10%, 0.55%
and 12 mg kg$^{-1}$ respectively.

123        Replicated soil samples were collected after harvesting of maize plants. Collected soil

samples were processed and analyzed for pH, EC, OC content and DTPA-Zn concentration
by the methodologies described above. The plant available Zn in soils was also extracted by
Mehlich 1 (Perkins, 1970), 0.1 M HCl   (Sorensen et al., 1971) and ABDTPA (Soltanpour
and Schwab, 1977) extractants by following the respective prescribed methods. Estimation of



Zn concentration was done on the clear extract by atomic absorption spectrophotometer
(AAS).
*2.2 Green house study*
Pot experiments were carried out in two Hariharapur and Debatoli series soils. The
experiments were carried out in plastic pots having 4 kg of soil with five levels of LR (0, 1/10
LR, 1/3 LR, 2/3 LR and LR), three levels of Zn concentration (0, 2.5 and 5.0 mg Zn kg$^{-1}$ soil)
and two levels of fresh FYM (35% moisture) (0 and 10 t ha$^{-1}$). All the pots received basal
treatments of N-P$_2$O$_5$-K$_2$O @ 150-60-40 kg ha$^{-1}$. Fertilizer N, P and K were applied through
analytical grade urea, calcium dihydrogen orthophosphate and muriate of potash,
respectively. Lime and Zn were added to soil through laboratory grade CaCO$_3$ and ZnSO$_4$
respectively.  All nutrients were mixed in soil thoroughly before sowing of seeds. The soil in
each plot was then irrigated to field capacity with deionized water and kept for incubation for
one week. Each treatment combination was replicated thrice in a factorial completely
randomized design. Four seeds of cv. KH 101 of maize were sown in each pot.  Two
seedlings of maize per each pot were maintained after emergence. Pots were irrigated with
water daily as per requirement of water on weight basis to maintain the field capacity. Above-
ground biomass of plants from each pot was harvested at the end of 60 days of growth.
*2.3 Plant analysis*
Harvested above-ground biomass of each pot was washed in deionized water, and then dried
in oven at 70 $^\circ$C for 48 h. After drying, dry matter yield (DMY) of each pot was recorded.
Dried plant material was  then ground  in a stainless steel Wiley mill, and digested in a di-
acid mixture of HNO$_3$ and HClO$_4$ (Jackson, 1973). Zn concentration was then determined in
aqueous extracts of the digested plant material by atomic absorption spectrophotometer
(AAS). Zn uptake was calculated as DMY multiplied by the Zn concentration.



*2.4 Statistical analysis*
The data regarding soil properties, DMY, Zn concentration, Zn uptake and extracted Zn by
different extractants    subjected to analysis of variance method (Gomez and Gomez 1984).
Least square difference (LSD) at $P \leq .01$ was used to compare among the treatment means.
Pearson's correlation coefficient values were estimated to establish relationship among soil
properties, DMY, Zn concentration, Zn uptake and extracted Zn by different extractants.
**3. Results**
*3.1 Soil properties*
Application of lime at different rates significantly increased pH in soils of both
Hariharapur and Debatoli series (Table 2, Fig. 1 a). With addition of graded doses of limes
viz. from no lime, 1/10[th] LR, 1/3[rd] LR, 2/3[rd] LR and LR, soil pH increased from 4.58 to 7.16
(without FYM addition) and from 4.89 to 7.23 (with FYM addition) in Hariharapur series and
from 5.83 to 6.95 (without FYM addition) and from 6.04 to 7.02 (with FYM addition) in
Debatoli series. Application of FYM without lime increased soil pH in both the soils (Table
2). Combined application of lime and FYM also enhanced soil pH significantly. Addition of
Zn did not have any effect on soil pH. Application of lime, FYM and Zn did not influence
soil EC levels in soils of both the series (Table 2, Fig. 1 b). However application of FYM
increased soil OC content in soils of both series (Table 2, Fig. 1c). Addition of lime and Zn
did not influence soil OC.
*3.2 Extractable zinc in post-harvest soil*
Data regarding amount Zn extracted by DTPA, Mehlich 1, 0.1 M HCl and ABDTPA
extractants in post harvest soil are given in Table 3. The amount of extracted Zn by DTPA,
Mehlich 1, and ABDTPA extractants decreased with increased level of lime application in
soils of both the series (Fig. 2 a, b, d). But addition of FYM (@ 10 t ha[-1]) in combination of





different levels of lime led to marked enhancement of extracted Zn by different extractants in
both the soils compared to only application of different lime levels (Table 3). Application Zn
at different levels viz. 2.5 and 5.0 mg kg$^{-1}$ with and without FYM increased the concentration
of extracted Zn by the different extractants. The amount of Zn extracted by different
extractants varied widely and it followed the order DTPA-Zn < ABDTPA-Zn < Mehlich-1 Zn
< 0.1 M HCl.

*3.3 Dry matter yield*

DMY of maize increased significantly with lime application up to 1/3$^{rd}$ LR (Table 4, Fig.
3 a) in soils of both the series. This indicated that lime application @ 1/3$^{rd}$ of LR was
optimum for these soils. Application of higher doses of lime (2/3$^{rd}$ LR and LR) did not result
in increased DMY. The mean DMY in 1/3$^{rd}$ LR treatment without FYM and with FYM was
139% and 149% of control respectively in Harihpur series soils. Similarly in Debatoli series
soil, the mean DMY was 84% and 120% of control without and with FYM application
respectively in combination with 1/3$^{rd}$ LR. Application of graded doses of Zn upto 5.0 mg kg$^{-}$
$^{1}$ to soil increased DMY with and without FYM application in Hariharapur series. Whereas in
Debatoli series, application of graded doses of Zn up to 5 mg kg$^{-1}$ without FYM and
application of Zn @ 2.5 mg kg$^{-1}$ with FYM enhanced DMY.

*3.4 Zinc concentration and uptake by maize*

Addition of higher doses of lime significantly reduced Zn concentration in maize crop
grown in soils of both the series (Table 4, Fig. 3 b). In contrast, application of Zn (@ 2.5 and
5.0 mg kg$^{-1}$) and FYM (@ 10 t ha$^{-1}$) increased Zn concentration in maize crop significantly in
soils of both the series (Table 4). In soils of Hariharapur series, application Zn @ 2.5 and 5
mg kg$^{-1}$ without and with FYM augmented Zn concentration in maize by 67.5 and 93.5 to 109
% respectively as compared to control (No Zn). Similarly, increased Zn concentrations of 22



to 35 and 58 to 73% were recorded with application of Zn @ 2.5 and 5 mg kg$^{-1}$ without and
with FYM respectively in comparison to no Zn control in soils of Debatoli series.  Mean Zn
uptake values were at par for no lime, 1/10$^{th}$ LR and 1/3$^{rd}$ LR with and without FYM
application and it was significantly higher than Zn uptake by 2/3$^{rd}$ LR and LR treatments in
soils of both the series (Table 4, Fig. 3 c). However, Zn and FYM application improved Zn
uptake by maize crop in soils of both series. Addition of Zn @ 2.5 and 5 mg kg$^{-1}$ enhanced
Zn uptake by 67 to 100 and 122 to 150% respectively as compared to no Zn control in soils
of Hariharapur series. Whereas, the enhancements in Zn uptake were 36 to50, 73 to 117%
due to application of Zn @ 2.5 and 5 mg kg$^{-1}$ respectively as compared to no Zn control in
soils of Debatoli series.
**4. Discussion**
Lime is a basic chemical and its application neutralizes soil acidity (H$^+$ and Al$^{3+}$ ions) and
makes soil more basic. In this study, application of increased rate of lime also enhanced soil
pH.  Anikwe et al. (2016) also reported increase in soil pH due to lime addition in an Ultisol
of Nigeria.  Application of lime along with FYM also enhanced soil pH. This is in line with
the findings of Saha et al. (2012). Normally, addition of organic matter lowers soil pH by
releasing H$^+$ ions    associated with organic anions or by nitrification in an open system
(Porter et al., 1980). But in contrary, it may cause pH increases either by mineralization of
organic anions to CO$_2$ and water (thereby removing H$^+$ ions) or because of the 'alkaline'
nature of the organic material (Helyar, 1976). Increase in soil pH due to addition FYM in our
study may be due to operation of the second mechanism.
Application of lime reduced the concentrations of extractable Zn extracted by DTPA,
Mehlich 1 and ABDTPA extractants. Reduced availability of Zn in soil due to liming has also
been reported by Tlustos et al. (2006) and Vondrackova et al. (2013). It is because of



conversion of plant available fractions of Zn to plant unavailable fractions resulting in
effective immobilisation (Davis-Carter and Shuman, 1993). But application of FYM
improved the concentrations of extracted Zn.  Addition of organic matter led to formation of
organic acids by microbial decomposition, which mobilize soil bound Zn and restrict the
fixation of soluble Zn by chelating it (Shukla, 1971; Sarkar and Deb, 1982; Tagwira et al.,
1992). It has also been reported by Saha et al. (1999) that application of organic matter to
cultivated acid soils was essential to counteract the adverse effect of lime application on Zn
availability. Application Zn with and without FYM enhanced the concentrations of extracted
Zn significantly. Rupa et al. (2003) also reported increased concentration of exchangeable
plus water soluble, inorganically, organically and oxide bound Zn in two Alfisols due to
addition of increased Zn rates.
Among the extractants used in this study, DTPA extracted lowest amount of Zn. This is
in agreement with the findings of Behera et al. (2011) who reported lowest amount of Zn
extracted by DTPA compared other extractants like Mehlich 1, Mehlich 3, 0.1 M HCl and
ABDTPA, by analysing four hundred soil samples collected from cultivated acid soils of
India. This may be ascribed to lower extracting power of DTPA in these soils owing to
reduced active sites of DTPA at lower pH values. Higher extractability of ABDTPA
compared to DTPA in these soils because of ABDTPA solution pH of 7.6 which allowed
DTPA to chelate and extract more Zn from soil. Mehlich 1 extractant which was originally
developed for prediction of plant available P in acidic coastal plain soil (pH<6.5) with low
cation exchange capacity (CEC<10meq/100g) and low organic matter (<5%), extracted more
amount of Zn compared to DTPA and ABDTPA extractants.  Higher extractability of Zn by
0.1 M HCl has also been reported by Naik and Das (2010) as compared to DTPA and 0.05 M
HCl extracted Zn in low land rice soils. This is because 0.1 M HCl extracts Zn from freshly
adsorbed iron and manganese oxides, carbonates, or decomposing organic matter and Zn



bound with the octahedral-OH in layer silicates (Hodgson, 1963). Dilute mineral acids of pH
1-2 showed the greatest extracting power for extraction of Zn, followed by buffered solutions
of pH 7-9 containing chelating agents and buffers or very dilute acids of pH 4-5 (Misra et al.,
1989). Zhang et al. (2010) reported Zn extraction capacity of different extractants in the order
of EDTA > Mehlich 3 > Mehlich 1 > DTPA > $NH_4OAc$ > $CaCl_2$ in polluted soils of rice in
south-eastern China. The amount Zn extracted in polluted soils of central Iran followed the
order Mehlich 3 > ABDTPA > DTPA > Mehlich 2 > $CaCl_2$ > HCl (Hosseiwwnpur and
Motaghian, 2015).

257        Significant increase in DMY was recorded with application of lime up to 1/3[rd] LR.

Increase in DMY with lime application up to 1/3[rd] LR may be ascribed to increase in soil pH
and positive influence on nutrient availability in soil (Tisdale, 2005). Increased DMY due to
FYM addition may be due to positive influence of on nutrient availability and uptake.
Increased DMY due to Zn addition in soils of Hariharapur series revealed that Zn is a limiting
nutrient in this soil. It was evident from low initial DTPA-Zn status (0.47 mg $kg^{-1}$) of this
soil. Grain and vegetative tissue (stover) yield of maize increased significantly with
successive application of Zn up to 1 kg $ha^{-1}$  in a Zn-deficient  (DTPA-Zn 0.38 mg $kg^{-1}$)
Vertisol of India (Behera et al., 2015). Zn addition to a soil with 0.18 mg $kg^{-1}$ Zn enhanced
wheat grain yield (Cakmak et al., 2010a; Cakmak et al., 2010b). However in Debatoli series,
DMY response to Zn application was obtained in spite of high initial DTPA-Zn status (1.45
mg $kg^{-1}$) which needs further investigation. In contrast to our findings, Zhang et al. (2012)
and Wang et al. (2012) reported that zinc fertilizer application did not improve the biomass
and grain yields of wheat and maize in rain-fed and low Zn calcareous soils of China. This
may be attributed to   Zn availability in soil influenced by several factors (Alloway, 2009)
and efficiency of the crops/genotypes to utilize available Zn in soils (Cakmak et al., 1998).





Addition of lime significantly reduced Zn concentration. This may be due to reduced
availability Zn in soil due to increased soil pH. Soil pH significantly influences Zn
distribution among different fractions and availability in soil (Sims, 1986; Smith, 1994) and
the plant uptake is primarily related with different Zn fractions (Behera et al., 2008).
However, FYM and Zn application improved Zn concentration in maize. Application of 5
and 10 mg Zn kg$^{-1}$ enhanced Zn concentration of navy bean shoot from 19.93 mg kg$^{-1}$ to
38.12 and 54.8 mg kg$^{-1}$ respectively (Gonzalez et al., 2008).  Significant increase in Zn
concentration in ear leaves of spring maize, shoots of wheat and in maize and wheat grains
was also reported by Wang et al. (2012). Payne et al. (1988) also reported increased Zn
concentration in maize grain under highest $ZnSO_4$ application from a long-term experiment.
Soil pH was negatively and significantly correlated with Zn concentration (r = -0.509**, r
= -0.343**) and Zn uptake by maize (r = -0.397**, r = -0.326**) in both the soil series (Table
5). This revealed that increased soil pH resulted in decreased Zn concentration and Zn uptake
in maize and vice versa. Wang et al. (2006) also recorded increased Zn concentration in
*Thlaspi caerulescens* with decreased soil pH. Soil OC content was positively and
significantly correlated with DMY (r = 0.221*), Zn concentration (r = 0.232*) and Zn uptake
(r = 0.294**) in Hariharpur series only.  It was also positively and significantly correlated
with DTPA, Mehlich 1 and 0.1 M HCl extracted Zn in soils of both the series. This is in line
with the findings of Katyal and Sharma (1991) and Shidhu and Sharma (2010).  DMY was
positively and significantly correlated with Zn uptake(r = 0.605**, 0.727**) in soils of both
the series. It was also positively and significantly correlated with Zn extracted by DTPA,
Mehlich 1, 0.1 M HCl and ABDTPA extractants in Hariharpur series and Zn extracted by
Mehlich 1, 0.1 M HCl and ABDTPA extractants in Debatoli series. Zn concentration in
maize was positively and significantly correlated with Zn uptake by maize and extracted Zn
by different extractants in soils of both the series. Positive and significant correlation





coefficient values were also obtained for Zn uptake vs Zn extracted by different extractants in
soils of both the series. Zn extracted by different extractants in soils of both series were
positively and significantly correlated with each other. The values of correlation coefficients
ranged from r = 0.811** to r = 0.937**. This indicated that the trend of extraction of Zn from
both the soils, by different extractants used in the study is similar. It corroborates the findings
of Gartley et al. (2002), Mylavarapu et al. (2002) , Nascimento et al. (2007) and Behera et al.
(2011) who have reported the suitability of extractants like DTPA, ABDTPA, Mehlich 1,
Mehlich 3 and 0.1 M HCl for extraction of phyto-available Zn in acids of different parts of
the world.  Since Zn extracted by different extractants like DTPA, ABDTPA, Mehlich 1,
Mehlich 3 and 0.1 M HCl was positively and significantly correlated  amongst themselves
and with DMY, Zn concentarion and Zn uptake by maize, all these extractants can be used
for extraction of Zn from acid soils.
**5. Conclusion**
From the study, it is concluded that application of lime with and without FYM
influenced phyto-available Zn extracted by different extractants like DTPA, ABDTPA,
Mehlich 1, Mehlich 3 and 0.1 M HCl in two acid soils of India. Increased level of lime
application led to enhancement of soil pH and reduction in extractable Zn in soils of both the
series and Zn concentration in maize. Lime application of 1/3LR was found to be optimum
for amelioration in these soils. Application of FYM along with lime improved the
concentration of extractable Zn in soil. Soil OC content was positively and significantly
correlated with Zn extracted by different extractants. Since   DTPA, ABDTPA, Mehlich 1,
Mehlich 3 and 0.1 M HCl extractable Zn in soils of both the series were positively and
significantly correlated with dry matter yield, Zn concentration and Zn uptake, these
extractants could be used for extraction of Zn in acid soils.




**Acknowledgements**

323        The study was supported by the grant from Indian Council of Agricultural Research,

New Delhi. We thank the Director of ICAR-Indian Institute of Soil Science, Bhopal, Madhya
Pradesh, India for providing necessary facilities for conducting the research work. We
acknowledge the help rendered by Ms. P. Singh, Mr. R. Singh and Mr. D. K. Verma during
the execution of the work.

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





**Table 1** Some selected characteristics of the experimental soils.

| Soil characteristics | Hariharapur series | Debatoli series |
|---|---|---|
| Taxonomic classification | Oxic Haplustalfs | Udic Rhodustalfs |
| pH (1:2.5) | 4.50 | 5.80 |
| EC (dS m$^{-1}$) | 0.14 | 0.23 |
| Organic carbon (%) | 0.31 | 0.22 |
| Clay (%) | 12.1 | 14.2 |
| Silt (%) | 15.0 | 11.6 |
| Sand (%) | 73.2 | 75.1 |
| CaCO$_3$ (%) | 20.0 | 32.0 |
| CEC (cmol(p$^+$) kg$^{-1}$) | 3.90 | 5.10 |
| Lime requirement (g kg$^{-1}$) | 3.34 | 1.51 |
| DTAP-Zn (mg kg$^{-1}$) | 0.47 | 1.45 |



















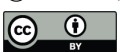

**Table 2** Soil pH, EC and OC content as influence by FYM, lime and Zn application.

| Treatments | No FYM | | | | FYM (10 t ha⁻¹) | | | | Overall mean |
|---|---|---|---|---|---|---|---|---|---|
| | No Zn | 2.5 mg Zn kg⁻¹ | 5.0 mg Zn kg⁻¹ | Mean | No Zn | 2.5 mg Zn kg⁻¹ | 5.0 mg Zn kg⁻¹ | Mean | |
| Hariharapur series | | | | | | | | | |
| **pH** | | | | | | | | | |
| No lime | 4.56 | 4.57 | 4.61 | 4.58 | 5.16 | 5.10 | 5.34 | 5.20 | 4.89 |
| 1/10th LR | 4.80 | 5.01 | 4.83 | 4.88 | 5.46 | 5.42 | 5.44 | 5.44 | 5.16 |
| 1/3rd LR | 5.69 | 6.14 | 5.57 | 5.80 | 5.93 | 6.49 | 5.97 | 6.13 | 5.97 |
| 2/3 LR | 6.45 | 6.53 | 6.62 | 6.53 | 6.92 | 7.08 | 6.57 | 6.86 | 6.70 |
| LR | 7.23 | 7.25 | 6.99 | 7.16 | 7.37 | 7.17 | 7.38 | 7.31 | 7.23 |
| Mean | 5.75 | 5.90 | 5.72 | - | 6.17 | 6.25 | 6.14 | - | - |
| LSD (0.01) | Lime = 0.19, Zn level = ns, FYM level = 0.25, Lime x Zn level = ns, Lime x FYM level = ns, Lime x Zn level x FYM level = 0.51, Zn level x FYM level = ns | | | | | | | | |
| **EC (dS m⁻¹)** | | | | | | | | | |
| No lime | 0.14 | 0.11 | 0.13 | 0.13 | 0.13 | 0.15 | 0.14 | 0.14 | 0.13 |
| 1/10th LR | 0.14 | 0.10 | 0.10 | 0.12 | 0.15 | 0.11 | 0.12 | 0.13 | 0.12 |
| 1/3rd LR | 0.13 | 0.13 | 0.11 | 0.12 | 0.12 | 0.10 | 0.14 | 0.12 | 0.12 |
| 2/3 LR | 0.12 | 0.13 | 0.11 | 0.12 | 0.12 | 0.15 | 0.10 | 0.12 | 0.12 |
| LR | 0.13 | 0.14 | 0.12 | 0.13 | 0.15 | 0.14 | 0.15 | 0.15 | 0.14 |
| Mean | 0.13 | 0.12 | 0.11 | - | 0.13 | 0.13 | 0.13 | - | 0.13 |
| LSD (0.01) | Lime = ns, Zn level = ns, FYM level = ns, Lime x Zn level = ns, Lime x FYM level = ns, Lime x Zn level x FYM level = ns, Zn level x FYM level = ns | | | | | | | | |
| **OC (%)** | | | | | | | | | |
| No lime | 0.26 | 0.27 | 0.25 | 0.26 | 0.32 | 0.37 | 0.34 | 0.34 | 0.30 |
| 1/10th LR | 0.27 | 0.24 | 0.27 | 0.26 | 0.33 | 0.34 | 0.39 | 0.35 | 0.31 |
| 1/3rd LR | 0.25 | 0.24 | 0.27 | 0.25 | 0.31 | 0.36 | 0.37 | 0.35 | 0.30 |
| 2/3 LR | 0.27 | 0.25 | 0.23 | 0.25 | 0.30 | 0.34 | 0.32 | 0.32 | 0.29 |
| LR | 0.24 | 0.21 | 0.22 | 0.22 | 0.25 | 0.34 | 0.33 | 0.31 | 0.27 |
| Mean | 0.26 | 0.24 | 0.25 | - | 0.30 | 0.35 | 0.35 | - | - |
| LSD (0.01) | Lime = ns, Zn level = ns, FYM level = 0.03, Lime x Zn level = ns, Lime x FYM level = ns, Lime x Zn level x FYM level = ns, Zn level x FYM level = ns | | | | | | | | |



**Debatoli series**

**pH**

| | | | | | | | | | |
|---|---|---|---|---|---|---|---|---|---|
| No lime | 5.88 | 5.85 | 5.77 | 5.83 | 6.14 | 6.17 | 6.45 | 6.25 | 6.04 |
| 1/10th LR | 5.93 | 5.88 | 5.94 | 5.92 | 6.28 | 6.42 | 6.56 | 6.42 | 6.17 |
| 1/3rd LR | 6.38 | 6.21 | 6.21 | 6.27 | 6.44 | 6.58 | 6.58 | 6.53 | 6.40 |
| 2/3rd LR | 6.64 | 6.67 | 6.6 | 6.64 | 6.76 | 6.75 | 6.64 | 6.72 | 6.68 |
| LR | 6.96 | 6.99 | 6.9 | 6.95 | 7.27 | 6.87 | 7.14 | 7.09 | 7.02 |
| Mean | 6.36 | 6.32 | 6.28 | - | 6.58 | 6.56 | 6.67 | - | - |

LSD (0.01): Lime = 0.17, Zn level = ns, FYM level = 0.20, Lime x Zn level = ns, Lime x FYM level = 0.47, Zn level x FYM level = ns

**EC (dS m⁻¹)**

| | | | | | | | | | |
|---|---|---|---|---|---|---|---|---|---|
| No lime | 0.23 | 0.22 | 0.27 | 0.24 | 0.21 | 0.26 | 0.23 | 0.23 | 0.24 |
| 1/10th LR | 0.27 | 0.27 | 0.23 | 0.25 | 0.21 | 0.20 | 0.21 | 0.21 | 0.24 |
| 1/3rd LR | 0.23 | 0.23 | 0.24 | 0.23 | 0.17 | 0.29 | 0.25 | 0.24 | 0.24 |
| 2/3rd LR | 0.23 | 0.21 | 0.21 | 0.21 | 0.23 | 0.19 | 0.24 | 0.22 | 0.22 |
| LR | 0.24 | 0.24 | 0.29 | 0.23 | 0.19 | 0.30 | 0.26 | 0.25 | 0.24 |
| Mean | 0.24 | 0.22 | 0.25 | - | 0.20 | 0.25 | 0.24 | - | - |

LSD (0.01): Lime = ns, Zn level = ns, FYM level = 0.04, Lime x Zn level = ns, Lime x FYM level = ns, Zn level x FYM level = ns

**OC (%)**

| | | | | | | | | | |
|---|---|---|---|---|---|---|---|---|---|
| No lime | 0.21 | 0.28 | 0.22 | 0.24 | 0.22 | 0.29 | 0.30 | 0.27 | 0.25 |
| 1/10th LR | 0.22 | 0.22 | 0.21 | 0.22 | 0.28 | 0.28 | 0.28 | 0.28 | 0.25 |
| 1/3rd LR | 0.21 | 0.23 | 0.24 | 0.23 | 0.28 | 0.26 | 0.29 | 0.28 | 0.26 |
| 2/3rd LR | 0.18 | 0.21 | 0.21 | 0.21 | 0.31 | 0.25 | 0.28 | 0.28 | 0.25 |
| LR | 0.21 | 0.25 | 0.24 | 0.24 | 0.28 | 0.30 | 0.28 | 0.25 | 0.26 |
| Mean | 0.21 | 0.24 | 0.25 | - | 0.27 | 0.29 | - | - | - |

LSD (0.01): Lime = ns, Zn level = ns, FYM level = ns, Lime x Zn level = ns, Lime x FYM level = ns, Zn level x FYM level = ns



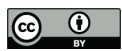

**Table 3** Effect of FYM, lime and Zn application on extractable Zn in soils.

| Treatments | No FYM | | | | FYM (10 t ha$^{-1}$) | | | | Overall mean |
|---|---|---|---|---|---|---|---|---|---|
| | No Zn | 2.5 mg Zn kg$^{-1}$ | 5.0 mg Zn kg$^{-1}$ | Mean | No Zn | 2.5 mg Zn kg$^{-1}$ | 5.0 mg Zn kg$^{-1}$ | Mean | |
| **Hariharapur series** | | | | | | | | | |
| **DTPA-Zn (mg kg$^{-1}$)** | | | | | | | | | |
| No lime | 0.40 | 1.44 | 2.95 | 1.60 | 0.88 | 1.68 | 3.21 | 1.92 | 1.76 |
| 1/10th LR | 0.40 | 1.24 | 2.30 | 1.31 | 0.66 | 1.67 | 3.20 | 1.84 | 1.58 |
| 1/3rd LR | 0.38 | 1.06 | 1.64 | 1.03 | 0.61 | 1.62 | 2.68 | 1.64 | 1.33 |
| 2/3 LR | 0.37 | 0.86 | 1.45 | 0.89 | 0.44 | 1.59 | 2.55 | 1.53 | 1.21 |
| LR | 0.34 | 0.77 | 1.25 | 0.79 | 0.44 | 1.27 | 2.53 | 1.41 | 1.10 |
| Mean | 0.38 | 1.08 | 1.92 | - | 0.61 | 1.57 | 2.83 | - | - |
| LSD (0.01) | Lime = 0.02, Zn level = 0.25, FYM level = 0.20, Lime x Zn level =0. 35, Lime x FYM level = 0.28, Zn level x FYM level = 0.47 | | | | | | | | |
| **Mehlich 1-Zn (mg kg$^{-1}$)** | | | | | | | | | |
| No lime | 0.78 | 1.68 | 3.85 | 2.10 | 1.23 | 3.70 | 5.08 | 3.34 | 2.72 |
| 1/10th LR | 0.77 | 1.66 | 3.74 | 2.06 | 1.17 | 3.20 | 4.88 | 3.08 | 2.57 |
| 1/3rd LR | 0.74 | 1.50 | 3.27 | 1.84 | 1.05 | 2.64 | 4.79 | 2.83 | 2.33 |
| 2/3 LR | 0.66 | 1.48 | 2.26 | 1.47 | 1.03 | 2.54 | 4.49 | 2.69 | 2.08 |
| LR | 0.51 | 1.24 | 1.92 | 1.22 | 0.94 | 2.54 | 4.25 | 2.58 | 1.90 |
| Mean | 0.69 | 1.51 | 3.01 | - | 1.09 | 2.92 | 4.70 | - | - |
| LSD (0.01) | Lime = 0.10, Zn level = 0.42, FYM level = 0.25, Lime x Zn level =0.55, Lime x FYM level = 0.37, Zn level x FYM level = 0.70 | | | | | | | | |
| **0.1 M HCl-Zn (mg kg$^{-1}$)** | | | | | | | | | |
| No lime | 0.90 | 2.50 | 4.62 | 2.67 | 1.50 | 3.81 | 6.24 | 3.85 | 3.26 |
| 1/10th LR | 0.89 | 2.31 | 4.61 | 2.60 | 1.34 | 3.72 | 6.20 | 3.75 | 3.18 |
| 1/3rd LR | 0.84 | 2.25 | 4.28 | 2.46 | 1.33 | 3.39 | 5.68 | 3.47 | 2.96 |
| 2/3 LR | 0.84 | 2.18 | 3.94 | 2.32 | 1.22 | 3.05 | 5.62 | 3.30 | 2.81 |
| LR | 0.84 | 1.93 | 3.91 | 2.23 | 1.06 | 3.03 | 5.43 | 3.17 | 2.70 |





| | | | | | | | | |
|---|---|---|---|---|---|---|---|---|
| Mean | 0.86 | 2.23 | 4.27 | - | 1.29 | 3.40 | 5.83 | - |
| LSD (0.01) | Lime = 0.02, Zn level = 0.30, FYM level = 0.27, Lime x Zn level =0. 37, Lime x FYM level = 0.30, Zn level x FYM level = 0.60 | | | | | | | |

**ABDTPA-Zn (mg kg⁻¹)**

| | | | | | | | | |
|---|---|---|---|---|---|---|---|---|
| No lime | 0.71 | 2.03 | 4.06 | 2.27 | 1.16 | 2.54 | 3.98 | 2.56 |
| 1/10th LR | 0.68 | 1.98 | 3.19 | 1.95 | 1.11 | 2.43 | 3.92 | 2.49 |
| 1/3rd LR | 0.59 | 1.70 | 2.62 | 1.64 | 1.00 | 2.43 | 3.84 | 2.42 |
| 2/3rd LR | 0.52 | 1.52 | 2.29 | 1.44 | 0.95 | 2.37 | 3.61 | 2.31 |
| LR | 0.49 | 1.25 | 2.12 | 1.29 | 0.93 | 2.21 | 3.31 | 2.15 |
| Mean | 0.60 | 1.70 | 2.85 | - | 1.03 | 2.40 | 3.73 | - |
| LSD (0.01) | Lime = 0.05, Zn level =0.28, FYM level = 0.32, Lime x Zn level =0. 32, Lime x FYM level = 0.41, Zn level x FYM level = 0.62 | | | | | | | |

Debatoli series

**DTPA-Zn (mg kg⁻¹)**

| | | | | | | | | |
|---|---|---|---|---|---|---|---|---|
| No lime | 1.45 | 2.62 | 3.29 | 2.45 | 1.63 | 2.80 | 4.33 | 2.92 |
| 1/10th LR | 1.30 | 2.32 | 2.93 | 2.18 | 1.37 | 2.54 | 4.01 | 2.64 |
| 1/3rd LR | 1.08 | 1.94 | 2.91 | 1.98 | 1.32 | 2.37 | 3.79 | 2.49 |
| 2/3rd LR | 0.99 | 1.78 | 2.80 | 1.86 | 1.08 | 2.25 | 2.95 | 2.09 |
| LR | 0.77 | 1.72 | 2.73 | 1.74 | 0.99 | 2.21 | 2.48 | 1.89 |
| Mean | 1.12 | 2.08 | 2.93 | - | 1.28 | 2.43 | 3.51 | - |
| LSD (0.01) | Lime = 0.21, Zn level = 0.50, FYM level = 0.35, Lime x Zn level =0.75, Lime x FYM level = 0.78, Zn level x FYM level = 0.98 | | | | | | | |

**Mehlich 1-Zn (mg kg⁻¹)**

| | | | | | | | | |
|---|---|---|---|---|---|---|---|---|
| No lime | 1.73 | 3.61 | 6.78 | 4.04 | 2.64 | 4.78 | 6.78 | 4.73 |
| 1/10th LR | 1.63 | 3.60 | 6.59 | 3.94 | 2.44 | 4.20 | 6.28 | 4.31 |
| 1/3rd LR | 1.51 | 3.44 | 6.12 | 3.69 | 2.42 | 4.10 | 6.21 | 4.24 |
| 2/3rd LR | 1.49 | 3.33 | 4.13 | 2.98 | 2.40 | 4.06 | 5.69 | 4.05 |
| LR | 1.26 | 3.15 | 4.06 | 2.82 | 2.37 | 3.74 | 5.46 | 3.86 |
| Mean | 1.53 | 3.43 | 5.54 | - | 2.45 | 4.18 | 6.08 | - |
| LSD (0.01) | Lime = 0.09, Zn level = 0.50, FYM level = 0.28, Lime x Zn level =0. 45, Lime x FYM level = 0.42, Zn level x FYM level = 0.85 | | | | | | | |

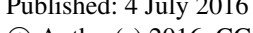



**0.1 M HCl-Zn (mg kg⁻¹)**

| | | | | | | | | | |
|---|---|---|---|---|---|---|---|---|---|
| No lime | 4.26 | 2.35 | 4.66 | 3.76 | 2.80 | 4.54 | 6.69 | 4.68 | 4.22 |
| 1/10th LR | 4.42 | 2.32 | 5.34 | 4.03 | 2.75 | 4.70 | 6.93 | 4.79 | 4.41 |
| 1/3rd LR | 4.40 | 2.22 | 6.07 | 4.23 | 2.86 | 5.25 | 7.61 | 5.24 | 4.74 |
| 2/3 LR | 3.87 | 2.23 | 7.46 | 4.52 | 2.91 | 5.14 | 7.01 | 5.02 | 4.77 |
| LR | 4.53 | 2.22 | 6.96 | 4.57 | 2.85 | 6.06 | 7.79 | 5.57 | 5.07 |
| Mean | 4.30 | 2.27 | 6.10 | - | 2.83 | 5.14 | 7.21 | - | - |
| LSD (0.01) | Lime = 0.06, Zn level = 0.35, FYM level = 0.37, Lime x Zn level =0. 45, Lime x FYM level = 0.45, Zn level x FYM level = 0.79 | | | | | | | | |

**ABDTPA-Zn (mg kg⁻¹)**

| | | | | | | | | | |
|---|---|---|---|---|---|---|---|---|---|
| No lime | 3.19 | 2.10 | 4.23 | 3.18 | 2.12 | 3.34 | 5.17 | 3.54 | 3.36 |
| 1/10th LR | 3.46 | 1.82 | 4.19 | 3.16 | 1.98 | 3.37 | 5.89 | 3.75 | 3.46 |
| 1/3rd LR | 2.77 | 1.61 | 4.60 | 2.99 | 1.93 | 3.46 | 5.17 | 3.52 | 3.26 |
| 2/3 LR | 2.05 | 1.36 | 5.12 | 2.84 | 1.75 | 3.02 | 4.26 | 3.01 | 2.93 |
| LR | 2.17 | 1.22 | 4.22 | 2.54 | 1.53 | 3.36 | 4.42 | 3.10 | 2.82 |
| Mean | 2.73 | 1.62 | 4.47 | - | 1.86 | 3.31 | 4.98 | - | - |
| LSD (0.01) | Lime = 0.10, Zn level = 0.35, FYM level = 0.35, Lime x Zn level =0. 47, Lime x FYM level = 0.20, Lime x Zn level x FYM level = 0.40, Zn level x FYM level = 0.70 | | | | | | | | |



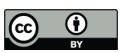

**Table 4** Effect of FYM, lime and Zn application on dry matter yield, Zn concentration and Zn uptake by maize.

| Treatments | No FYM | | | | FYM (10 t ha$^{-1}$) | | | | Overall mean |
|---|---|---|---|---|---|---|---|---|---|
| | No Zn | 2.5 mg Zn kg$^{-1}$ | 5.0 mg Zn kg$^{-1}$ | Mean | No Zn | 2.5 mg Zn kg$^{-1}$ | 5.0 mg Zn kg$^{-1}$ | Mean | |
| Hariharapur series | | | | | | | | | |
| *Dry matter (g pot$^{-1}$)* | | | | | | | | | |
| No lime | 1.64 | 2.02 | 2.04 | 1.90 | 2.06 | 2.60 | 2.23 | 2.30 | 2.10 |
| 1/10th LR | 2.43 | 2.37 | 2.16 | 2.32 | 2.21 | 2.74 | 2.66 | 2.53 | 2.43 |
| 1/3rd LR | 2.88 | 2.87 | 2.96 | 2.83 | 2.57 | 2.89 | 3.66 | 2.98 | 2.91 |
| 2/3rd LR | 2.65 | 2.37 | 2.66 | 2.64 | 2.40 | 2.40 | 3.01 | 2.66 | 2.65 |
| LR | 1.77 | 2.06 | 2.52 | 2.12 | 1.94 | 2.05 | 2.71 | 2.23 | 2.18 |
| Mean | 2.27 | 2.34 | 2.47 | - | 2.23 | 2.53 | 2.85 | - | - |
| LSD (0.01) | Lime = 0.30, Zn level = 0.11, FYM level = 0.11, FYM level = 0.25, Lime x Zn level =0.50, Lime x FYM level = 0.61, Zn level x FYM level = 0.42 | | | | | | | | |
| *Zn concentration (mg kg$^{-1}$)* | | | | | | | | | |
| No lime | 54.0 | 84.0 | 112 | 83.3 | 57.4 | 104 | 119 | 93.2 | 88.4 |
| 1/10th LR | 53.3 | 87.4 | 113 | 84.6 | 59.2 | 99.5 | 119 | 92.7 | 88.6 |
| 1/3rd LR | 38.5 | 63.5 | 75.0 | 59.0 | 46.3 | 72.8 | 80.0 | 66.4 | 62.7 |
| 2/3rd LR | 27.4 | 52.7 | 60.8 | 47.0 | 35.4 | 59.8 | 67.6 | 54.2 | 50.6 |
| LR | 25.2 | 44.8 | 54.2 | 41.4 | 31.2 | 48.9 | 58.1 | 46.1 | 43.7 |
| Mean | 39.7 | 66.5 | 83.0 | - | 45.9 | 76.9 | 88.8 | - | - |
| LSD (0.01) | Lime = 3.50, Zn level = 0.11, FYM level = 0.11, FYM level =3.21, Lime x Zn level = 2.00, Lime x FYM level = 5.70, Zn level x FYM level = 3.15 | | | | | | | | |
| *Zn uptake (mg pot$^{-1}$)* | | | | | | | | | |
| No lime | 0.11 | 0.14 | 0.23 | 0.16 | 0.12 | 0.27 | 0.26 | 0.22 | 0.19 |
| 1/10th LR | 0.13 | 0.21 | 0.24 | 0.19 | 0.13 | 0.27 | 0.32 | 0.24 | 0.22 |
| 1/3rd LR | 0.10 | 0.18 | 0.22 | 0.17 | 0.11 | 0.21 | 0.29 | 0.20 | 0.19 |
| 2/3rd LR | 0.08 | 0.13 | 0.16 | 0.12 | 0.09 | 0.14 | 0.20 | 0.15 | 0.13 |




| | | | | | | | | | |
|---|---|---|---|---|---|---|---|---|---|
| LR | 0.05 | 0.09 | 0.14 | 0.09 | 0.06 | 0.10 | 0.16 | 0.11 | 0.10 |
| Mean | 0.09 | 0.15 | 0.20 | - | 0.10 | 0.20 | 0.25 | - | - |
| LSD (0.01) | Lime = 0.002, Zn level = 0.005, FYM level = 0.004, Lime x Zn level =0.008, Lime x FYM level = 0.007, Zn level x FYM level = 0.012 | | | | | | | | |

**Debatoli series**

**Dry matter (g pot⁻¹)** — $Dry\ matter\ (g\ pot^{-1})$

| | | | | | | | | | |
|---|---|---|---|---|---|---|---|---|---|
| No lime | 2.84 | 3.55 | 4.19 | 3.53 | 3.45 | 3.72 | 3.44 | 3.57 | 3.55 |
| 1/10th LR | 3.37 | 3.94 | 4.52 | 3.94 | 3.56 | 4.06 | 4.21 | 3.91 | 3.93 |
| 1/3rd LR | 3.71 | 4.32 | 4.54 | 4.19 | 3.80 | 4.84 | 4.46 | 4.37 | 4.28 |
| 2/3 LR | 3.55 | 3.67 | 4.43 | 3.88 | 3.53 | 3.74 | 3.76 | 3.68 | 3.78 |
| LR | 3.27 | 3.54 | 3.46 | 3.42 | 3.46 | 3.59 | 3.55 | 3.54 | 3.48 |
| Mean | 3.35 | 3.80 | 4.23 | - | 3.56 | 3.99 | 3.88 | - | - |
| LSD (0.01) | Lime = 0.32, Zn level = 0.22, FYM level = ns, Lime x Zn level =0. 58, Lime x FYM level = ns, Zn level x FYM level =ns | | | | | | | | |

**Zn concentration (mg kg⁻¹)** — $Zn\ concentration\ (mg\ kg^{-1})$

| | | | | | | | | | |
|---|---|---|---|---|---|---|---|---|---|
| No lime | 62.2 | 85.0 | 119 | 88.7 | 71.0 | 86.2 | 126 | 94.4 | 91.6 |
| 1/10th LR | 60.4 | 78.4 | 105 | 81.3 | 70.7 | 84.3 | 116 | 90.3 | 85.8 |
| 1/3rd LR | 55.3 | 68.9 | 94.8 | 73.0 | 71.6 | 77.3 | 97.9 | 82.3 | 77.6 |
| 2/3 LR | 47.8 | 66.5 | 75.2 | 63.2 | 52.4 | 69.5 | 80.2 | 67.4 | 65.3 |
| LR | 39.7 | 60.6 | 64.8 | 55.0 | 44.8 | 62.6 | 70.6 | 59.4 | 57.2 |
| Mean | 53.1 | 71.9 | 91.8 | - | 62.1 | 76.0 | 98.1 | - | - |
| LSD (0.01) | Lime = 1.80, Zn level = 0.20, FYM level = 1.50, Lime x Zn level =2.10, Lime x FYM level = 3.80, Zn level x FYM level = 2.10 | | | | | | | | |

**Zn uptake (mg pot⁻¹)** — $Zn\ uptake\ (mg\ pot^{-1})$

| | | | | | | | | | |
|---|---|---|---|---|---|---|---|---|---|
| No lime | 0.18 | 0.30 | 0.50 | 0.33 | 0.25 | 0.32 | 0.44 | 0.34 | 0.33 |
| 1/10th LR | 0.20 | 0.31 | 0.47 | 0.33 | 0.24 | 0.34 | 0.49 | 0.36 | 0.34 |
| 1/3rd LR | 0.21 | 0.30 | 0.43 | 0.31 | 0.27 | 0.37 | 0.44 | 0.36 | 0.34 |
| 2/3 LR | 0.17 | 0.24 | 0.33 | 0.25 | 0.19 | 0.26 | 0.30 | 0.25 | 0.25 |
| LR | 0.13 | 0.21 | 0.23 | 0.19 | 0.15 | 0.23 | 0.25 | 0.21 | 0.20 |
| Mean | 0.18 | 0.27 | 0.39 | - | 0.22 | 0.30 | 0.38 | - | - |
| LSD (0.01) | Lime = 0.03, Zn level = 0.11, FYM level = 0.02, Lime x Zn level =ns, Lime x FYM level = 0.08, Zn level x FYM level = ns | | | | | | | | |



**Table 5** Pearson's correlation coefficient values revealing relationship among soil properties, dry matter yield, Zn concentration, Zn uptake and extracted Zn in soils (n = 90).

| | pH | EC | OC | Dry matter yield | Zn conc. | Zn uptake | DTPA-Zn | Mehlich 1-Zn | 0.1 M HCl-Zn | ABDTPA-Zn |
|---|---|---|---|---|---|---|---|---|---|---|
| **Hariharapur series** | | | | | | | | | | |
| pH | 1 | | | | | | | | | |
| EC | 0.058 | 1 | | | | | | | | |
| OC | -0.089 | -0.084 | 1 | | | | | | | |
| Dry matter yield | 0.059 | 0.093 | 0.221* | 1 | | | | | | |
| Zn conc. | -0.590** | -0.029 | 0.232* | 0.047 | 1 | | | | | |
| Zn uptake | -0.397** | 0.036 | 0.294** | 0.605** | 0.792** | 1 | | | | |
| DTPA-Zn | 0.010 | -0.073 | 0.211* | 0.391** | 0.610** | 0.523** | 1 | | | |
| Mehlich 1-Zn | 0.130 | -0.045 | 0.272** | 0.281** | 0.510** | 0.545** | 0.897** | 1 | | |
| 0.1 M HCl-Zn | 0.046 | -0.076 | 0.242* | 0.260* | 0.633** | 0.626** | 0.871** | 0.929** | 1 | |
| ABDTPA-Zn | -0.011 | -0.013 | 0.136 | 0.285** | 0.656** | 0.673** | 0.887** | 0.922** | 0.923** | 1 |
| **Debatoli series** | | | | | | | | | | |
| pH | 1 | | | | | | | | | |
| EC | 0.032 | 1 | | | | | | | | |
| OC | 0.113 | -0.098 | 1 | | | | | | | |
| Dr matter yield | -0.154 | 0.096 | 0.011 | 1 | | | | | | |
| Zn conc. | -0.343** | 0.042 | 0.158 | 0.384** | 1 | | | | | |
| Zn uptake | -0.326** | 0.086 | 0.110 | 0.727** | 0.905** | 1 | | | | |
| DTPA-Zn | -0.087 | 0.061 | 0.290** | 0.133 | 0.741** | 0.715** | 1 | | | |
| Mehlich 1-Zn | 0.168 | 0.091 | 0.317** | 0.330** | 0.589** | 0.568** | 0.811** | 1 | | |
| 0.1 M HCl-Zn | 0.188 | 0.130 | 0.294** | 0.333** | 0.562** | 0.545** | 0.822** | 0.937** | 1 | |
| ABDTPA-Zn | -0.074 | 0.108 | 0.193 | 0.419** | 0.772** | 0.748** | 0.889** | 0.890** | 0.887** | 1 |

*$p \leq 0.05$; **$p \leq 0.01$.





Fig. 1. Soil pH, EC and OC as influenced by interaction of Zn application and lime rate in Hariharapur and Debatoli series. Error bars represent ± SE.

Fig. 2. Extractable Zn by different extractants as influenced by interaction of Zn application and lime rate in Hariharapur and Debatoli series. Error bars represent ± SE.

Fig. 3. Dry matter yield, Zn concentration and Zn uptake by maize as influenced by interaction of Zn application and lime rate in Hariharapur and Debatoli series. Error bars represent ± SE.

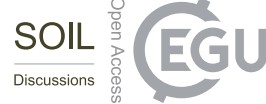



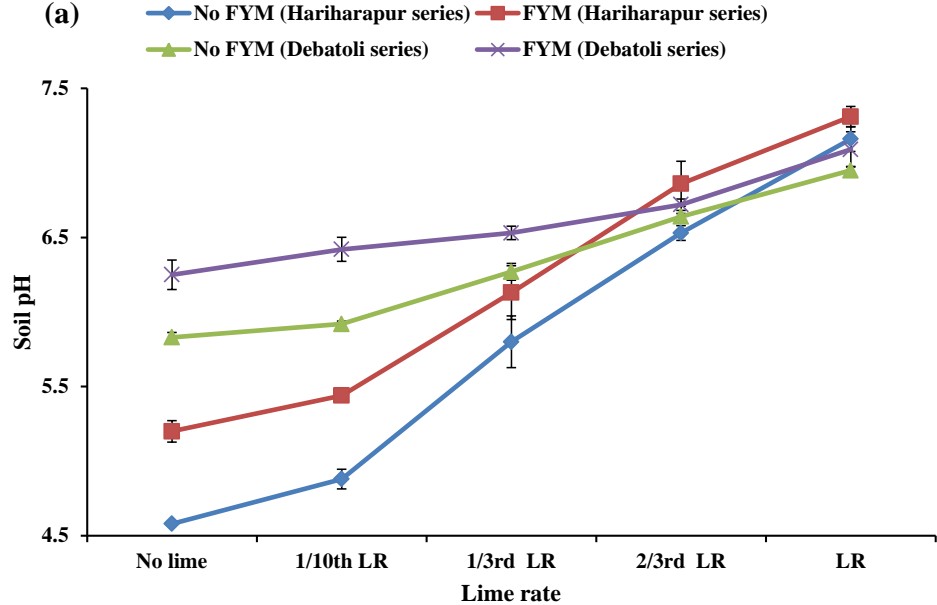

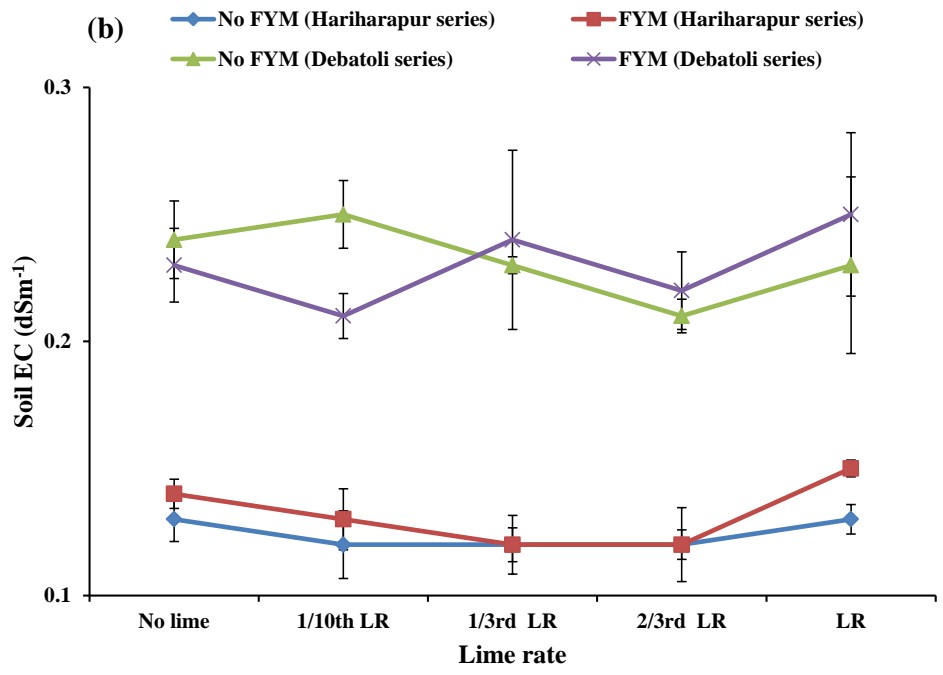



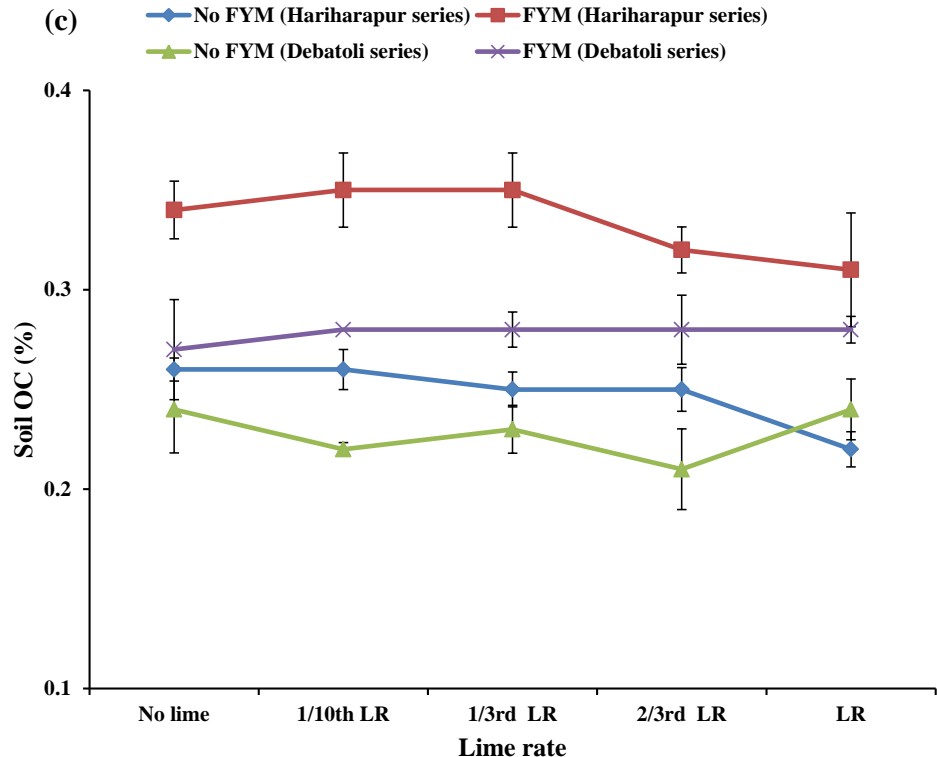

Fig. 1. Soil pH, EC and OC as influenced by interaction of Zn application and lime rate in Hariharapur and Debatoli series. Error bars represent ± SE.



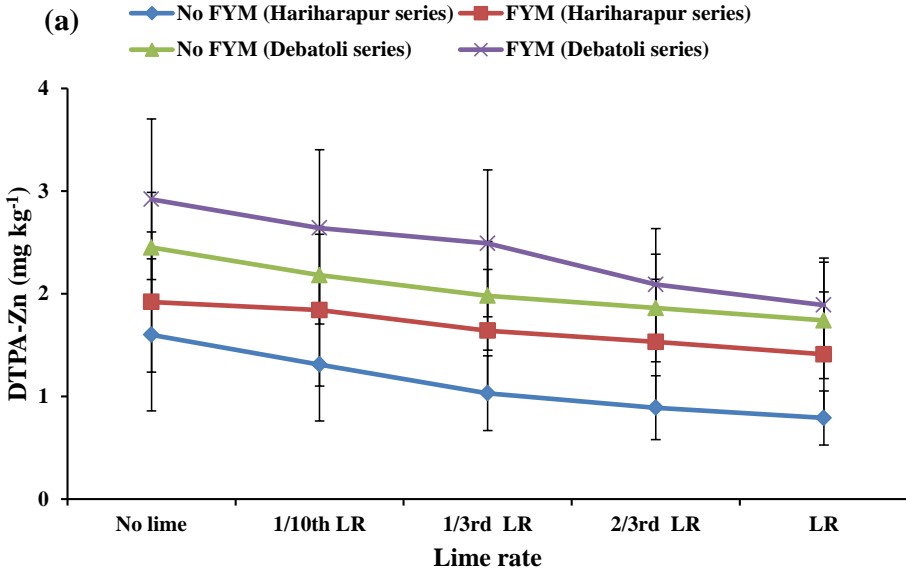




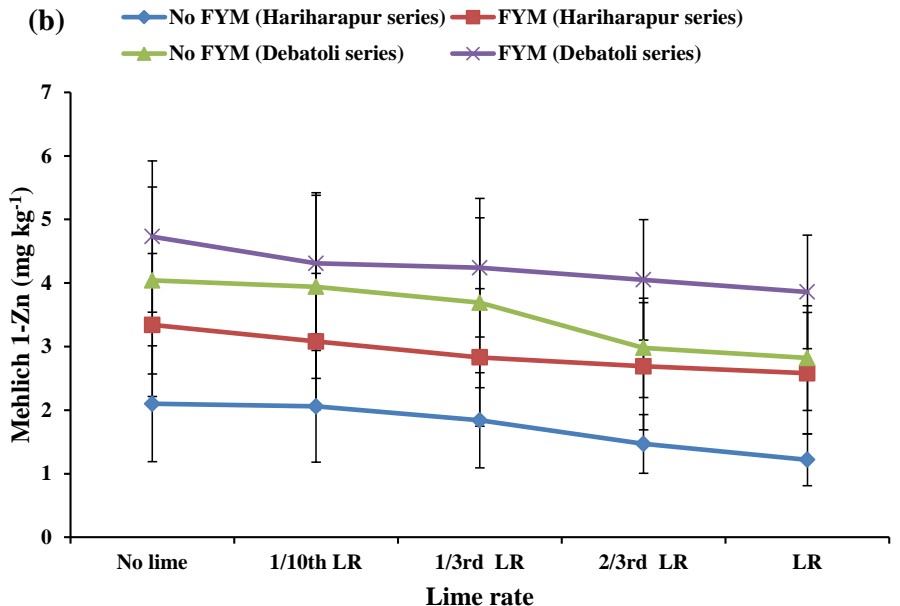

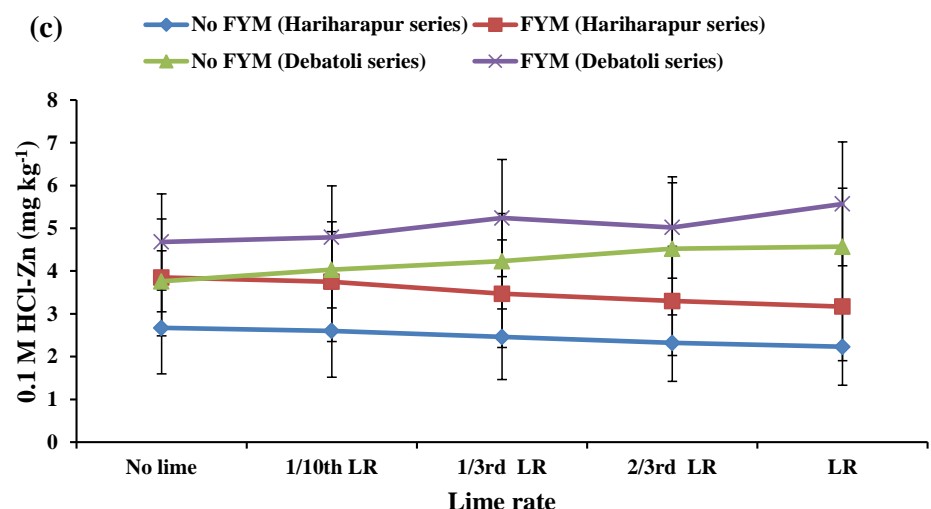

**(d)**




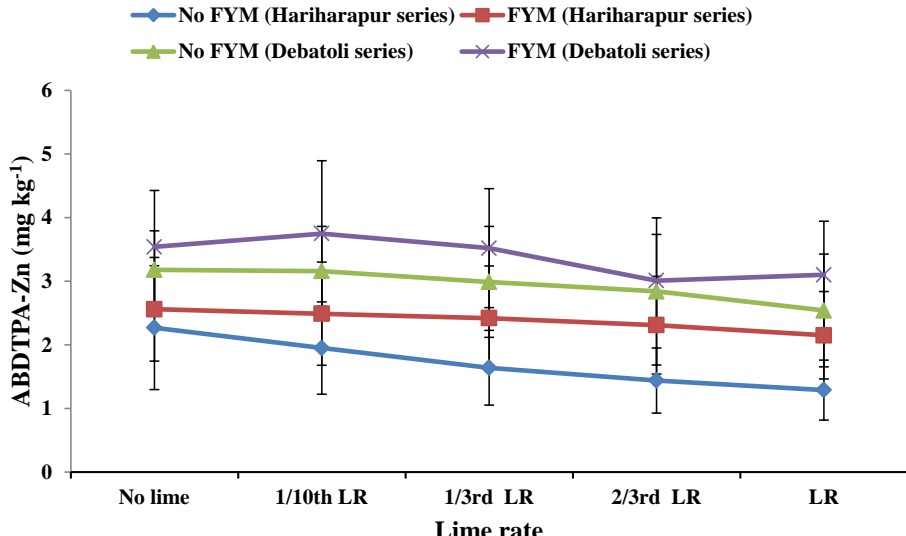

Fig. 2. Extractable Zn by different extractants as influenced by interaction of Zn application and lime rate in Hariharapur and Debatoli series. Error bars represent ± SE.





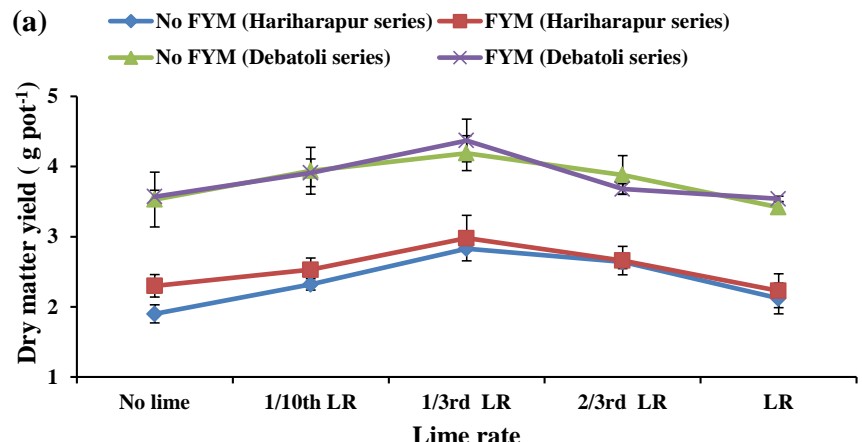

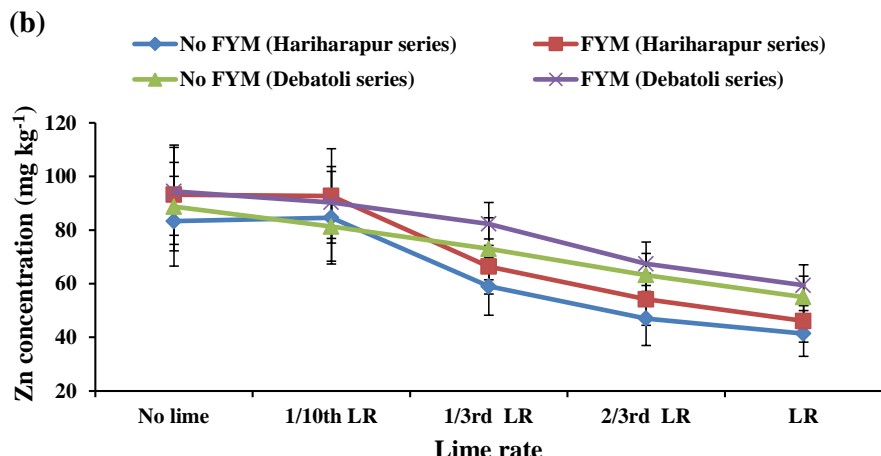



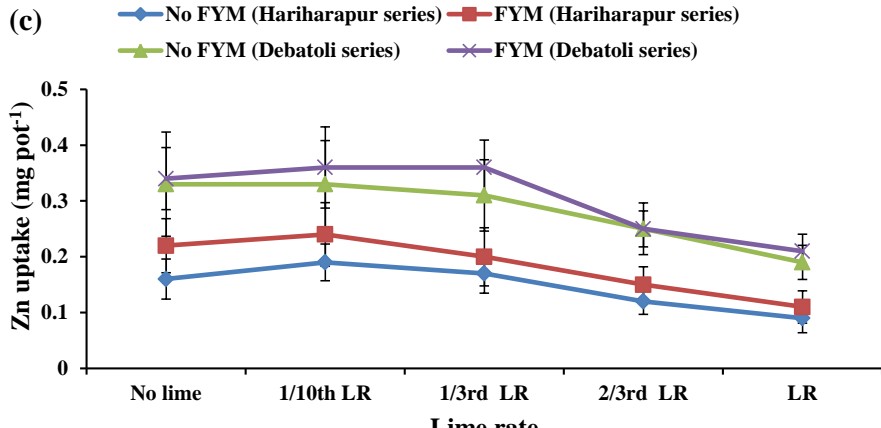

Fig. 3.  Dry matter yield, Zn concentration and Zn uptake by maize as influenced by interaction of
Zn application and lime rate in Hariharapur and Debatoli series. Error bars represent ± SE.