# Peer review of "Lime and zinc application influence soil zinc availability, dry"

_SOIL, 2016_

## Referee Comment (RC1) · Anonymous Referee #1 · 6 Oct 2016

The article is of limited scientific merit for publication in "SOIL".

Some suggestions are given below:

1. Introduction. - It is not clear as to where the research is leading to: the problem of soil acidity regarding to crop production? Zinc-deficient soils for crops? the combination of acid soils and zinc-deficient soils? You should clearly define the starting situation that generates the need for research and avoid redundancy in drafting the text. - The evaluation of the different Zn-extractants is not relevant for this research, although this assessment could be the subject of another more specific work. - Objectives should be reworked once the problem has been well defined (and also the title of the article, according to them). What relevance does the assessment have of the effect of the

[Figure]

application of farmyard manure on the soil OC content (since OM was added)? (as well as the influence of lime application in pH). This is not relevant.

2. Materials and methods The experiment was in pots, of which the diameter was not indicated, although the weight of soil was. However, the added amount of farmyard manure was expressed in t/ha. You should indicate the amount (g) of FYM applied to each pot for to know the nutrients (i.e., Zn) added to the soil with FYM as you are evaluating Zn extraction by crop (and by extractants) by varying Zn and lime doses. What type of farmyard manure is it? The results of chemical analysis indicate FYM: OC 0.12

3. Results - Irrelevant results were included (e.g., adding farmyard manure increased the soil OC, the addition of lime increased soil pH, ..., adding Zn (and FYM) to soil increased Zn concentration in plant) - No critical levels of Zn in soil and/or plant tissues were indicated. Was the concentration of Zn in plants for unfavorable treatment below the critical values (literature)? Was there observed Zn deficiency symptoms in the plants with lower Zn concentration? - The Figures presented are redundant (and unnecessary), since data are also shown in tables. The Tables do not clarify the results of statistical analysis (comparison of means). The differences observed between means of the different treatments should be indicated by adding the corresponding letter (a, b, c...) to each mean value.

---

## Referee Comment (RC2) · Anonymous Referee #2 · 7 Oct 2016

My overall assessment of this manuscript is that although it covers a subject of potential interest to the journal, it does it without a clear objective and combining information on subjects that are well proven (e.g. liming) and very little in others (such as in a more detailed discussion of the interaction among treatments).

I have indicated several comments in the annotated version of the manuscript, but I summarize here some major points in case the authors want to rework the manuscript for a possible resubmission.

1- The article lacks clear objectives, stated at the end of the introduction. There are apparently three overlapping studies: 1. Field experiments, greenhouse pot experiments, and effect of the extractant used in determining Zn concentration. However, it

[Figure]

**SOILD**

is unclear how they are coordinated for a final objective, giving the impression of been three related (but not properly coordinated) experiments. The manuscript might be re-organized and edited, particularly in the introduction and M&Methods to address this problem.

2- There is missing some key information in the material and methods sections (for instance a better definition of the soil sampling in the field studies, or the properties of the manure, . . .). There are many other examples of these in the annotated version of the manuscript. They should be addressed.

3-Tthere is duplication in results presented in the Tables and Graphs while at the same time the statistical models uses (and in their major results, particularly in the case of interactions between variables) This should be addressed.

4- The discussion ad conclusions suffer the same lack of focus already mentioned in the overall organization for the manuscript. This should also been addressed-

For these reasons my recommendation is that the manuscript should be returned to the authors for major modifications before been reconsidered for possible publication.

Please also note the supplement to this comment:
http://www.soil-discuss.net/soil-2016-41/soil-2016-41-RC2-supplement.pdf

**Supplement:**

[revised manuscript text omitted]

---

## Editor Comment (EC1) · J. A. Gomez (Editor) · 7 Oct 2016

Both reviewers indicate that the manuscript is of limited merit, in its current form, for publication in soil. Both indicates concisely which are the major points ot be addressed for a possible resubmission, and one of them also expand these comments in an annotated version of the manuscript.

I agree with both reviewers and endorse their recommendation for rejection, considering possible resubmission if the authors considered that they can rework the manuscript to address the comments indicated by the two reviewers.

---

## Author Comment (AC1) · 18 Oct 2016

Dear Editor and Referees,

We thank you very much for providing constructive and useful suggestion for our manuscript. We have modified the manuscript incorporating the suggestions. The details of our responses and revisions are given below.

Comments of Editor:- Both reviewers indicate that the manuscript is of limited merit, in its current form, for publication in soil. Both indicates concisely which are the major points to be addressed for a possible resubmission, and one of them also expand these comments in an annotated version of the manuscript. I agree with both reviewers and endorse their recommendation for rejection, considering possible resubmission if the authors considered that they can rework the manuscript to address the comments indicated by the two reviewers.

Our response:- We have revised the manuscript according to the comments of both of the reviewers. We feel that the quality of the manuscript has improved by incorporating the suggestions of the reviewers. We have attached the revised manuscript for your kind perusal. Thanks a lot to you and to the reviewers.

Comments of Anonymous referee # 1:-

Comment:- The article is of limited scientific merit for publication in "SOIL". Some suggestions are given below: 1. Introduction. - It is not clear as to where the research is leading to: the problem of soil acidity regarding to crop production? Zinc-deficient soils for crops? the combination of acid soils and zinc-deficient soils? You should clearly define the starting situation that generates the need for research and avoid redundancy in drafting the text.

Our response:- Thank you sir for this suggestion. We have modified the introduction part of the manuscript and deleted unwanted portions. The present study was carried out to examine the influence of lime and farmyard manure and Zn addition on dry matter yield, Zn concentration and uptake by maize and soil properties and extractable Zn by different extractants in acid soils. The information would be useful for assessment of extractable Zn and its management in acid soils where Zn availability is one of the main problems and Zn application is imminent and application of lime and FYM is a common practice to obtain higher crop yield. This information has been incorporated in introduction part to bring the clarity of the study.

Comment:- The evaluation of the different Zn-extractants is not relevant for this research, although this assessment could be the subject of another more specific work.

Our response: Yes, we do agree with the reviewer that evaluation of different Zn- extractants is more specific work. We have used different extractants to extract Zn in post harvest soil in order to establish relationship among the extracted Zn by different extractants and dry matter yield and Zn concentration and uptake by maize. Based on this relationship, we have identified the suitability of different extracts for extraction of Zn in acid soils.

Comment:- Objectives should be reworked once the problem has been well defined (and also the title of the article, according to them).

Our response: We have clarified the problem in the introduction part of the manuscript as mentioned earlier. We have also modified the title of the article and the objectives of the study.

Comment:-What relevance does the assessment have of the effect of the application of farmyard manure on the soil OC content (since OM was added)? (as well as the influence of lime application in pH). This is not relevant.

Our response: We agree with the reviewer. We have assessed the soil properties like Ph, EC and OC content along with extracted Zn in post-harvest soils to visualize the influence lime and FYM addition and to assess the relationship among the soil properties, dry matter yield and Zn concentration and uptake by maize.

Comment:- Materials and methods The experiment was in pots, of which the diameter was not indicated, although the weight of soil was.

Our response:- We have included diameter of each pot in the manuscript.

Comment:- However, the added amount of farmyard manure was expressed in t/ha. You should indicate the amount (g) of FYM applied to each pot for to know the nutrients (i.e., Zn) added to the soil with FYM as you are evaluating Zn extraction by crop (and by extractants) by varying Zn and lime doses.

Our response: Yes, we agree with the reviewer. We have indicated the amount of FYM in g added in the manuscript.

 Comment:-What type of farmyard manure is it? The results of chemical analysis indicate FYM: OC 0.12.

Our response:- Locally available farmyard manure was used for the study and it was decomposed mixture of left over fodder (predominantly) fed to farm animals, animal dung and animal urine. There was a typo error in providing the OC content of FYM. It is 0.22% instead of 0.12% as mentioned earlier. We have corrected it in the manuscript and the information has been provided in Table 1.

Comment:-Results - Irrelevant results were included (e.g., adding farmyard manure increased the soil OC, the addition of lime increased soil pH, ..., adding Zn (and FYM) to soil increased Zn concentration in plant).

Our response: We have modified the result as per the suggestions. We have also changed the sequence of the results presented in the manuscript to make it more relevant.

 Comment:- No critical levels of Zn in soil and/or plant tissues were indicated. Was the concentration of Zn in plants for unfavorable treatment below the critical values (literature)? Was there observed Zn deficiency symptoms in the plants with lower Zn concentration?

Our response:- We have include critical concentration of DTPA-Zn in soils  (0.8 mg kg$^{-1}$)(in table 1) and plant tissues in the manuscript. We have also compared the values of Zn concentration in plant tissues under different treatments with critical values available in literature. We have mentioned that the Zn concentration in maize under all the treatments were well above the critical Zn concentration of 15 to 22 mg kg$^{-1}$ for maize crop (Alloway,

2008) and no visual Zn deficiency symptoms in plants were recorded.

Comment:- The Figures presented are redundant (and unnecessary), since data are also shown in tables.

Our response: In agreement with the reviewer, we have deleted the figures no. 1 from the manuscript. We have modified the figures no. 2 and 3 as per the suggestions of the referee #2.

Comment:-The Tables do not clarify the results of statistical analysis (comparison of means).

The differences observed between means of the different treatments should be indicated by adding the corresponding letter (a, b, c...) to each mean value.

Our response: Yes it is correct. We have provided different letters to identify the observed differences between means in the tables.

Comments of Anonymous referee # 2:-

Comment:-My overall assessment of this manuscript is that although it covers a subject of potential interest to the journal, it does it without a clear objective and combining information on subjects that are well proven (e.g. liming) and very little in others (such as in a more detailed discussion of the interaction among treatments).

Our response:- We thank the reviewer for visualizing the importance of our study. We have modified the introduction part of the manuscript to clarify the problem and clearly stated the objectives of the study. Since liming and farmyard manure application is common by the farmers in acid soils, many researchers have worked in this line. But the information regarding the influence of lime, farmyard manure and Zn in acid soils on crop yield, Zn concentration in plant tissue and extracted Zn and their relationship is lacking. Therefore, the present study was carried out. We have tried our best to improve the discussion part of the manuscript by incorporating information about the significant interaction effects among the treatments.

Comment:-I have indicated several comments in the annotated version of the manuscript, but

I summarize here some major points in case the authors want to rework the manuscript for a possible resubmission.

Our response:-We have gone through the comments given in the annotated version of the manuscript. We have modified the manuscript as per the comments provided in the different parts of the manuscript.

Comment:-The article lacks clear objectives, stated at the end of the introduction. There are apparently three overlapping studies: 1. Field experiments, greenhouse pot experiments, and effect of the extractant used in determining Zn concentration. However, it is unclear how they are coordinated for a final objective, giving the impression of been three related (but not properly coordinated) experiments.

Our response: We agree with you that we have collected bulk soil from field to conduct green house study. The objective of the present study was to study the influence of lime and farmyard manure and Zn addition on dry matter yield, Zn concentration and uptake by maize and soil properties and extractable Zn by different extractants in acid soils. The information would be useful for assessment of extractable Zn and its management in acid soils where Zn availability is one of the main problems and Zn application is imminent and application of lime and FYM is a common practice to obtain higher crop yield. This information has been incorporated in introduction part to bring the clarity of the study. We have modified and properly coordinated the introduction part of the manuscript to make it better understandable.

Comment:-The manuscript might be reorganized and edited, particularly in the introduction and M&Methods to address this problem.

Our response:- We have modified and reorganized the introduction and material and method section of the manuscript and made it systematic.

Comment:- There is missing some key information in the material and methods sections (for instance a better definition of the soil sampling in the field studies, or the properties of the manure, . . .).

Our response: We have incorporated the information regarding soils collected from field and methods used for analysis of manures as per the comments given in the annotated version of the manuscript.

 Comment:-There are many other examples of these in the annotated version of the manuscript. They should be addressed.

Our response:- We have modified the manuscript as per the comments given.

Comment:-There is duplication in results presented in the Tables and Graphs while at the same time the statistical models uses (and in their major results, particularly in the case of interactions between variables) This should be addressed.

Our response: In agreement with the reviewer, we have deleted the figures no. 1 from the manuscript. We have modified the figures no. 2 and 3 as per the suggestions of the reviewers given in the manuscript. We have also tried our best to describe the results including the interaction effects of different treatments as per the statistical test used in the study.

Comment:- The discussion ad conclusions suffer the same lack of focus already mentioned in the overall organization for the manuscript. This should also been addressed.

Our response: We have modified the discussion and conclusion parts of the manuscript. Now we feel that is properly ordered and systematic.

Comment:-For these reasons my recommendation is that the manuscript should be returned to
the authors for major modifications before been reconsidered for possible publication.

Our response:- Thank you very much. We have modified the manuscript as per the
suggestions.

With above modifications, we are hereby submitting the revised manuscript for your kind
perusal.

With kind regards,

Sanjib Kumar Behera

**Effect of lime, farmyard manure and zinc application on soil properties, dry matter yield, zinc concentration and uptake by maize and extractable zinc in Alfisols**

**Sanjib K. Behera[a,\*], Arvind K. Shukla[b], Brahma S. Dwivedi[c], Brij L. Lakaria[b]**

*ICAR-Indian Institute of Oil Palm Research, Pedavegi, West Godavari District, Andhra Pradesh 534450, India*

*ICAR-Indian Institute of Soil Science, Nabibagh, Berasia Road, Bhopal, Madhya Pradesh 462038, India*

*ICAR-Indian Agricultural Research Institute, Pusa, New Delhi, 110012, India*

\*Corresponding author: sanjibkumarbehera123@gmail.com (S. K. Behera), ICAR-Indian Institute of Oil Palm Research, Pedavegi, West Godavari District, Andhra Pradesh 534450, India

ABSTRACT

Zinc (Zn) deficiency is widespread in all types of soils of world including acid soils affecting crop production and nutritional quality of edible plant parts. There is, however, limited information available regarding effects of lime and farmyard manure (FYM) and Zn addition to acid soils on dry matter yield, Zn concentration and uptake by maize (*Zea mays* L.) and soil properties and extractable Zn by different extractants. Green house pot experiments were carried out in two acid soils to study the effect of five levels of lime (0, 1/10 lime requirement (LR), 1/3 LR, 2/3 LR and LR), three levels of Zn concentration (0, 2.5 and 5.0 mg Zn kg$^{-1}$ soil) and two levels of FYM (0 and 10 t ha$^{-1}$) addition on dry matter yield, Zn concentration and uptake by maize plant grown up to 60 days and soil pH, EC and OC content and extractable Zn in soil. Lime rate of 1/3$^{rd}$ LR was found to be optimum as dry matter yield of maize increased significantly with lime application up to 1/3$^{rd}$ LR in soils of both the series and decreased subsequently. Addition of FYM with and without lime increased dry matter yield. Application of Zn up to 5.0 mg kg$^{-1}$ to soil increased dry matter yield with and without FYM application in soils of Hariharapur series. Addition of higher doses of lime significantly reduced Zn concentration in maize crop grown in soils of both the series. Mean Zn uptake values were at par for no lime, 1/10th LR and 1/3rd LR with and without FYM application and it was significantly higher than Zn uptake by 2/3rd LR and LR

treatments. However, FYM application improved Zn uptake by maize crop.  Increased level of lime application reduced Zn extracted by DTPA, Mehlich 1, 0.1 N HCl and ABDTPA

extractants. However, application of FYM along with lime improved Zn extraction. The amount of Zn extracted by different extractants followed the order DTPA-Zn < ABDTPA-

Zn < Mehlich-1 Zn < 0.1 M HCl. Zn extracted by different extractants like DTPA, ABDTPA,

Mehlich 1 and 0.1 M HCl was positively and significantly correlated amongst themselves and with dry matter yield, Zn concentration and Zn uptake by maize. Among the extractants,

ABDTPA was found to be the best extractant for extraction of Zn in acid soils.

*Keywords:* Alfisol, Dry matter yield, Farmyard manure, Lime, Zinc concentration

**1. Introduction**

Soil acidity is a serious problem affecting crop production across the world including

India which is having 34.5% of arable land with acid soils (Maji et al., 2012). Ameliorating acid soils with suitable amendments and proper nutrient especially zinc (Zn) management in

Zn-deficient acid soils (Rautaray et al., 2003; Behera et al., 2011) are areas of concern for obtaining higher crop yield. Amelioration of acidic soils is beneficial to plant growth because it improves soil pH and replenishes nutrients (Moon et al., 2014). Application of liming material is an effective method for amelioration of acid soils (Ponnette et al., 1991; Quoggio et al., 1995). Lime is normally oxides, carbonates and hydroxides of calcium or magnesium.

There are about four types of lime viz., quicklime (CaO), slaked lime (Ca(OH)$_2$), limestone (CaCO$_3$) and dolomite. Application CaCO$_3$ to acid soils reduces soil acidity, improves basic cations status and significantly increases the yields of crops grown on Ultisol (Cifu et al.,

2004). It also improves physical structure in nitric soils. However, adoption of standard recommendation of lime requirement (LR) for different groups of acid soils is difficult for farmers, which is uneconomical and unsustainable (Barman et al., 2014). Therefore, lower doses of LR like 1/10[th], 1/3[rd] and 2/3[rd] of LR are applied by the farmers.

Soil pH and organic matter content are the most important soil factors affecting phyto-availability of Zn in soil (Suman, 1986; Lindsay, 1992). Increased soil pH due to addition of lime can influence availability of Zn in soil by altering its equilibrium (Verma and

Minhas, 1987).  Higher level of soil pH results in reduced extractable Zn content due to increased adsorptive capacity, formation of hydrolyzed forms of zinc, chemisorption on calcium carbonate and co- precipitation in iron oxides (Cox and Kamprath, 1972). Available organic materials such as farmyard manure (FYM) are generally used by the farmers along with chemical fertilizers because it improves soil physical, chemical and biological properties (Nambiar, 1994). Addition of organic matter to soil results in enhanced microbiological activity which adds complexing agents as well as influences the redox status of soil.

According to Moody et al. (1997), higher levels of organic matters enhance Zn availability by increasing exchangeable and organic fractions of Zn and reducing oxide fractions of Zn. The effect of addition of organic matter on Zn availability in soils has also been reported by different workers (Murthy, 1982; Ghanem and Mikkelsen, 1987). But the information regarding influence of addition of lime with and without FYM to acid soils on Zn availability in soil and Zn concentration and Zn uptake by crops is limited.

Appropriate soil tests for plant available Zn is not yet available for all types of agricultural soils around the world. However, extractants like  diethylene triamine penta acetic acid (DTPA), ethylene diamine tetra acetic acid (EDTA), hydrochloric acid, ammonium bicarbonate-DTPA (ABDTPA) , Mehlich 1 and Mehlich 3  are used for extraction of plant available Zn from soils (Alloway, 2008). But DTPA extractant is the most widely used.  The DTPA soil test was originally developed to categorize near-neutral and calcareous soils with insufficient plant available Zn to support maximum yield of crops (Lindsay and Norvell, 1978). But the same has been used for acid soils also for extraction of plant available Zn.  According to O'Connor (1988), whenever one strays from the original design of the test, one should be aware of the possible consequences and pass that awareness on to others. Based on correlation among the extracted Zn by different extractants and with soil properties, Behera et al. (2011) reported the usefulness of DTPA, Mehlich 1, Mehlich 3,

0.1 N HCl and ABDTPA extractant for extraction of plant available Zn in acid soils of India.

However, 0.1 N HCl was found to be best extractant (based on higher values of correlation coefficient with soil pH and OC) for extraction of plant available Zn in acid soils. But there is scanty information available regarding the relationship of extracted Zn by different extractants with Zn concentration and uptake by crop plants.

The information from the present study would be useful for assessment of extractable

Zn and its management in acid soils where Zn availability is one of the main problems and

Zn application is imminent and application of lime and FYM is a common practice.  Keeping above facts in view, the present study was carried out (i) to evaluate the influence of lime,

FYM and Zn addition on dry matter yield, Zn concentration and uptake by maize (*Zea mays*

L.) crop and (ii) to evaluate the influence of lime, FYM and Zn addition to acid soils on soil pH, EC and OC content, extractable Zn as extracted by different extractants.

**2.  Materials and methods**

*2.1 Soil and farmyard manure characteristics*

The bulk surface (0-15 cm depth) soils collected from Hariharpur series (Oxic Haplustalf,

Alfisol (Soil Survey Staff, 2014)) and Debatoli series (Udic Rhodostalf, Alfisol (Soil Survey

Staff, 2014)) of Bhubaneswar and Ranchi (India), respectively were used in the study. The collected soils were air dried and stone and debris were removed and then ground to pass a 2

mm sieve and analysed for selected properties (Table 1). Soil properties like pH and EC were determined done on 1: 2.5 soil water ratio (w/v) suspension using pH meter and EC meter following half an hour equilibrium (Jackson, 1973). Soil organic carbon (OC) content was estimated by chromic acid digestion-back titration method (Walkley and Black, 1934). The clay, silt and sand per cent of soils were determined by hydrometer method (Bouyoucos,

1962). Calcium carbonate ($CaCO_3$) content was determined by rapid titration method (Puri,

1930) and cation exchange capacity (CEC) by neutral normal ammonium acetate method (Richards, 1954). Lime requirement (LR) of the soil was estimated by extractant buffer method (Shoemaker et al., 1961). The plant available Zn in soils was extracted by DTPA

method (Lindsay and Norvell, 1978). Estimation of Zn concentration was done on the clear extract by atomic absorption spectrophotometer (AAS). After drying of FYM at 70 $^{o}$C for

24 h followed by grinding to pass through 20 mesh sieve, one gram of ground FYM was dry- ashed at 450 $^{o}$C for 2h. Ashed samples were extracted using 0.5 N HCl. Zn concentration was determined in filtered extracts. The total OC (loss on ignition), N (Kjeldahl method), P

(nitric-perchloric 9:4 digestion) and K (nitric-perchloric 9:4 digestion) concentrations in

FYM were estimated according to Tandon (2009) (Table 1).

*2.2 Green house study, soil and plant analysis*

Pot experiments were carried out in two Hariharapur and Debatoli series soils. The experiments were carried out in plastic pots (each with diameter of 20 cm) having 4 kg of soil with five levels of LR (0, 1/10 LR, 1/3 LR, 2/3 LR and LR), three levels of Zn concentration (0, 2.5 and 5.0 mg Zn kg$^{-1}$ soil) and two levels of fresh FYM (35% moisture) (0 and 4.5 g

FYM kg$^{-1}$ soil viz., 0 and 10 t FYM ha$^{-1}$). Locally available FYM was used for the study and it was decomposed mixture of left over fodder fed to farm animals, animal dung and urine.

All the pots received basal treatments of N-$P_2O_5$-$K_2O$ @ 150-60-40 kg ha$^{-1}$ (equivalent to

66.7-26.7-17.8 mg N-$P_2O_5$-$K_2O$ kg$^{-1}$ soil, respectively). Fertilizer N, P and K were applied through analytical grade urea, calcium dihydrogen orthophosphate and muriate of potash, respectively. Lime and Zn were added to soil through laboratory grade $CaCO_3$ and $ZnSO_4$

respectively.  All nutrients were mixed in soil thoroughly before sowing of seeds. The soil in each pot was then irrigated to field capacity with deionized water and kept for incubation for one week. Each treatment combination was replicated thrice in a factorial completely randomized design. Four seeds of cv. KH 101 of maize were sown in each pot.  Two seedlings of maize per each pot were maintained after emergence. Pots were irrigated with water daily as per requirement of water on weight basis to maintain the field capacity. Above- ground biomass of plants from each pot was harvested at the end of 60 days of growth.

Harvested above-ground biomass of each pot was washed in deionized water, and then dried in oven at 70 $^o$C for 48 h. After drying, dry matter yield (DMY) of each pot was recorded.

Dried plant material was  then ground  in a stainless steel Wiley mill, and digested in a di- acid mixture of $HNO_3$ and $HClO_4$ (Jackson, 1973). Zn concentration was then determined in aqueous extracts of the digested plant material by atomic absorption spectrophotometer (AAS). Zn uptake was calculated as DMY multiplied by the Zn concentration.

Soil sample from each pot were collected after harvesting of maize plants. Collected soil samples were processed and analyzed for pH, EC, OC content and DTPA-Zn concentration following the methods described above. The plant available Zn in soils was also extracted by

DTPA (Lindsay and Norvell, 1978), Mehlich 1 (Perkins, 1970), 0.1 M HCl  (Sorensen et al.,

1971) and ABDTPA (Soltanpour and Schwab, 1977) extractants by following the respective prescribed methods. Estimation of Zn concentration was done on the clear extract by AAS.

*2.3 Statistical analysis*

The data regarding soil properties, DMY, Zn concentration, Zn uptake and extracted Zn by different extractants   subjected to analysis of variance method (Gomez and Gomez 1984).

Least square difference (LSD) at $P \leq .01$ was used to compare among the treatment means.

Pearson's correlation coefficient values were estimated to establish relationship among soil properties, DMY, Zn concentration, Zn uptake and extracted Zn by different extractants.

**3. Results**

*3.1 Dry matter yield*

DMY of maize increased significantly with lime application up to $1/3^{rd}$ LR (Table 2, Fig. 1 a)

in soils of both the series. This indicated that lime application @ $1/3^{rd}$ of LR was optimum for these soils. Application of higher doses of lime ($2/3^{rd}$ LR and LR) did not result in increased

DMY. However, this finding needs to be verified by conducting field experiment. The mean

DMY in $1/3^{rd}$ LR treatment without FYM and with FYM was 139% and 149% of control respectively in Harihpur series soils. Similarly in Debatoli series soil, the mean DMY was

84% and 120% of control without and with FYM application respectively in combination with $1/3^{rd}$ LR. Application of graded doses of Zn upto 5.0 mg $kg^{-1}$ to soil increased DMY

with and without FYM application in Hariharapur series. Whereas in Debatoli series, application of graded doses of Zn up to 5 mg $kg^{-1}$ without FYM and  application of Zn @ 2.5

mg $kg^{-1}$ with FYM enhanced DMY.

*3.2 Zinc concentration and uptake by maize*

Addition of higher doses of lime significantly reduced Zn concentration in maize crop grown in soils of both the series (Table 2, Fig. 1 b). In contrast, application of Zn (@ 2.5 and 5.0 mg

$kg^{-1}$) and FYM (@ 10 t $ha^{-1}$) increased Zn concentration in maize crop significantly in soils of both the series (Table 2, Fig 1c). In soils of Hariharapur series, application Zn @ 2.5 and 5

mg $kg^{-1}$ without and with FYM augmented Zn concentration in maize by 67.5 and 93.5 to 109

% respectively, as compared to control (No Zn). Similarly, increased Zn concentrations of 22

to 35 and 58 to 73% were recorded with application of Zn @ 2.5 and 5 mg $kg^{-1}$ without and with FYM respectively in comparison to no Zn control in soils of Debatoli series. However, the Zn concentration in maize under all the treatments were well above the critical Zn concentration of 15 to 22 mg kg$^{-1}$ for maize crop (Alloway, 2008) and no visual Zn deficiency symptoms in plants were recorded.  Mean Zn uptake values were at par for no lime, 1/10$^{th}$ LR

and 1/3$^{rd}$ LR with and without FYM application and it was significantly higher than Zn uptake by 2/3$^{rd}$ LR and LR treatments in soils of both the series (Table 2, Fig. 1 d). However,

Zn and FYM application improved Zn uptake by maize crop in soils of both series (Fig. 1 e).

Addition of Zn @ 2.5 and 5 mg kg$^{-1}$ enhanced Zn uptake by 67 to 100 and 122 to 150%

respectively as compared to no Zn control in soils of Hariharapur series. Whereas, the enhancements in Zn uptake were 36 to50, 73 to 117% due to application of Zn @ 2.5 and 5

mg kg$^{-1}$ respectively as compared to no Zn control in soils of Debatoli series.

*3.3 Soil properties*

Application of lime at different rates significantly increased pH in soils of both Hariharapur and Debatoli series (Table 3). With addition of graded doses of limes viz. from no lime,

1/10$^{th}$ LR, 1/3$^{rd}$ LR, 2/3$^{rd}$ LR and LR, soil pH increased from 4.58 to 7.16 (without FYM

addition) and from 4.89 to 7.23 (with FYM addition) in Hariharapur series and from 5.83 to

6.95 (without FYM addition) and from 6.04 to 7.02 (with FYM addition) in Debatoli series.

Application of FYM without lime increased soil pH in both the soils (Table 3). Interaction effect of combined application of lime and FYM on soil pH was significant. Soil pH values obtained by addition of 2/3$^{rd}$ LR and LR along with FYM were at par.  Addition of Zn did not have any effect on soil pH. Sole application of lime, FYM and Zn and their interaction did not influence soil EC levels in soils of both the series (Table 3). However application of FYM

increased soil OC content in soils of both series. Addition of lime and Zn and their interaction did not influence soil OC.

*3.4 Extractable zinc in post-harvest soil*

Data regarding amount Zn extracted by DTPA, Mehlich 1, 0.1 M HCl and ABDTPA

extractants in post harvest soil are given in Table 4 and Figure 2. Perusal of data revealed significant effect of individual application of lime, FYM and Zn and their interaction on extracted Zn by different extractants. The amount of extracted Zn by DTPA, Mehlich 1, and

ABDTPA extractants decreased with increased level of lime application in soils of both the series (Fig. 2 a, b, d). But addition of FYM (@ 10 t ha$^{-1}$) in combination of different levels of lime led to marked enhancement of extracted Zn by different extractants in both the soils compared to only application of different lime levels (Table 4). Application Zn at different levels viz. 2.5 and 5.0 mg kg$^{-1}$ with and without FYM increased the concentration of extracted Zn by the different extractants. The amount of Zn extracted by DTPA, Mehlich 1,

0.1 M HCl and ABDTPA extractant varied from 1.10 to 1.76, 1.90 to 2.72, 2.70 to 3.26 and

1.72 to 2.42 mg kg$^{-1}$ respectively, under different levels of lime application across FYM and

Zn application in soils of Hariharpur series. Whereas, the Zn extracted by DTPA, Mehlich 1,

0.1 M HCl and ABDTPA extractant varied from  1.82 to 2.69, 3.34 to 4.39, 4.22 to 5.07 and

2.82 to 3.36 mg kg$^{-1}$ respectively, under different levels of lime application across FYM and

Zn application in soils of Debatoli series. In both the series, the extracted Zn followed the order DTPA-Zn < ABDTPA-Zn < Mehlich1-Zn < 0.1 M HCl-Zn.

**4. Discussion**

Significant increase in DMY was recorded with application of lime up to 1/3$^{rd}$ LR. Increase in DMY with lime application up to 1/3$^{rd}$ LR may be ascribed to increase in soil pH and positive influence on nutrient availability in soil (Tisdale, 2005). Our finding is in line with the observations made by Barman et al. (2014) who reported lime application at 1/3rd LR

was optimum for obtaining cauliflower yield in Typic Fluvaquent soil of West Bengal, India.

There was reduction in DMY with lime application at 2/3$^{rd}$ LR and LR in soils of both the series. This may be ascribed to reduced availability Zn in soil with 2/3$^{rd}$ LR and LR rate of lime application and adverse effect on other soil properties. This needs to be verified by conducting filed experiment. 
[revised manuscript text omitted]

**5. Conclusion**

Lime application of 1/3LR was found to be optimum for amelioration of acid soils. The concentration of Zn in maize tissue and extracted Zn by different extractants like DTPA, Mehlich 1, 0.1 M HCl and ABDTPA in both the soils reduced with lime application. Application of FYM along with lime improved the Zn concentration in maize plant and extractable Zn in soils. Since  DTPA, Mehlich 1, 0.1 M HCl and ABDTPA extractable Zn in soils of both the series were positively and significantly correlated with dry matter yield, Zn concentration and Zn uptake, these extractants could be used for extraction of Zn in acid soils. However based on higher correlation coefficient values, ABDTPA was found to be best extractant for extraction of Zn in acid soils.

**Acknowledgements**

The study was supported by the grant from Indian Council of Agricultural Research, New Delhi. We thank the Director of ICAR-Indian Institute of Soil Science, Bhopal, Madhya Pradesh, India for providing necessary facilities for conducting the research work. We acknowledge the help rendered by Ms. P. Singh, Mr. R. Singh and Mr. D. K. Verma during the execution of the work. The authors thank the topical editor and the anonymous reviewers for their useful suggestions for improvement of the manuscript.

**Table 1** Some selected characteristics of the experimental soils and farmyard manure.

| Characteristics | Experimental soils | |
|---|---|---|
| | Hariharapur series | Debatoli series |
| Taxonomic classification | Oxic Haplustalfs | Udic Rhodustalfs |
| pH (1:2.5) | 4.50 | 5.80 |
| EC (dS m$^{-1}$) | 0.14 | 0.23 |
| Organic carbon (%) | 0.31 | 0.22 |
| Clay (%) | 12.1 | 14.2 |
| Silt (%) | 15.0 | 11.6 |
| Sand (%) | 73.2 | 75.1 |
| CaCO$_3$ (%) | 20.0 | 32.0 |
| CEC (cmol(p$^+$) kg$^{-1}$) | 3.90 | 5.10 |
| Lime requirement (g kg$^{-1}$) | 3.34 | 1.51 |
| DTAP-Zn (mg kg$^{-1}$) | 0.47 | 1.45 |
| | Farmyard manure | |
| Total organic carbon (%) | 0.22 | |
| Total N (%) | 0.48 | |
| Total P (%) | 0.10 | |
| Total K (%) | 0.55 | |
| Total Zn (mg kg$^{-1}$) | 12 | |

*Critical concentration of DTPA-Zn is 0.80 mg kg$^{-1}$

**T able 2** Effect of FYM, lime and Zn application on dry matter yield, Zn concentration and Zn uptake by maize.

| Treatments | No FYM | | | | FYM (10 t ha$^{-1}$) | | | | Overall mean |
|---|---|---|---|---|---|---|---|---|---|
| | No Zn | 2.5 mg Zn kg$^{-1}$ | 5.0 mg Zn kg$^{-1}$ | Mean | No Zn | 2.5 mg Zn kg$^{-1}$ | 5.0 mg Zn kg$^{-1}$ | Mean | Overall mean |
| Hariharapur series | | | | | | | | | |
| | | | | Dry matter (g pot$^{-1}$) | | | | | |
| No lime | 1.64 | 2.02 | 2.04 | 1.90a | 2.06 | 2.60 | 2.23 | 2.30d | 2.10 |
| 1/10$^{th}$ LR | 2.43 | 2.37 | 2.16 | 2.32b | 2.21 | 2.74 | 2.66 | 2.53ef | 2.43 |
| 1/3$^{rd}$ LR | 2.88 | 2.87 | 2.96 | 2.83c | 2.57 | 2.89 | 3.66 | 2.98f | 2.91 |
| 2/3$^{rd}$ LR | 2.65 | 2.37 | 2.66 | 2.64c | 2.40 | 2.40 | 3.01 | 2.66d | 2.65 |
| LR | 1.77 | 2.06 | 2.52 | 2.12ab | 1.94 | 2.05 | 2.71 | 2.23d | 2.18 |
| Mean | 2.27aa | 2.34aa | 2.47bb | - | 2.23cc | 2.53cce | 2.85dde | - | - |
| LSD (0.01) | Lime = 0.30, Zn level = 0.11, FYM level = 0.25, Lime x Zn level =0. 50, Lime x FYM level = 0.61, Zn level x FYM level = 0.42 | | | | | | | | |
| | | | | Zn concentration (mg kg$^{-1}$) | | | | | |
| No lime | 54.0 | 84.0 | 112 | 83.3a | 57.4 | 104 | 119 | 93.2e | 88.4 |
| 1/10$^{th}$ LR | 53.3 | 87.4 | 113 | 84.6a | 59.2 | 99.5 | 119 | 92.7e | 88.6 |
| 1/3$^{rd}$ LR | 38.5 | 63.5 | 75.0 | 59.0b | 46.3 | 72.8 | 80.0 | 66.4f | 62.7 |
| 2/3$^{rd}$ LR | 27.4 | 52.7 | 60.8 | 47.0c | 35.4 | 59.8 | 67.6 | 54.g | 50.6 |
| LR | 25.2 | 44.8 | 54.2 | 41.4d | 31.2 | 48.9 | 58.1 | 46.1h | 43.7 |
| Mean | 39.7aa | 66.5bb | 83.0cc | - | 45.9dd | 76.9ee | 88.8ff | - | - |
| LSD (0.01) | Lime = 3.50, Zn level = 0.11, FYM level = 2.00, Lime x Zn level =3.21, Lime x FYM level = 5.70, Zn level x FYM level = 3.15 | | | | | | | | |
| | | | | Zn uptake (mg pot$^{-1}$) | | | | | |
| No lime | 0.11 | 0.14 | 0.23 | 0.16a | 0.12 | 0.27 | 0.26 | 0.22f | 0.19 |
| 1/10$^{th}$ LR | 0.13 | 0.21 | 0.24 | 0.19b | 0.13 | 0.27 | 0.32 | 0.24g | 0.22 |
| 1/3$^{rd}$ LR | 0.10 | 0.18 | 0.22 | 0.17c | 0.11 | 0.21 | 0.29 | 0.20h | 0.19 |
| 2/3$^{rd}$ LR | 0.08 | 0.13 | 0.16 | 0.12d | 0.09 | 0.14 | 0.20 | 0.15i | 0.13 |
| LR | 0.05 | 0.09 | 0.14 | 0.09e | 0.06 | 0.10 | 0.16 | 0.11j | 0.10 |
| Mean | 0.09aa | 0.15bb | 0.20cc | - | 0.10dd | 0.20dd | 0.25dd | - | - |
| LSD (0.01) | Lime = 0.002, Zn level = 0.005, FYM level = 0.004, Lime x Zn level =0.008, Lime x FYM level = 0.007, Zn level x FYM level = 0.012 | | | | | | | | |

Debatoli series

**Dry matter (g pot$^{-1}$)**

| | | | | | | | | | |
|---|---|---|---|---|---|---|---|---|---|
| No lime | 2.84 | 3.55 | 4.19 | 3.53a | 3.45 | 3.72 | 3.44 | 3.57d | 3.55 |
| 1/10$^{th}$ LR | 3.37 | 3.94 | 4.52 | 3.94b | 3.56 | 4.06 | 4.21 | 3.91d | 3.93 |
| 1/3$^{rd}$ LR | 3.71 | 4.32 | 4.54 | 4.19b | 3.80 | 4.84 | 4.46 | 4.37d | 4.28 |
| 2/3$^{rd}$ LR | 3.55 | 3.67 | 4.43 | 3.88b | 3.53 | 3.74 | 3.76 | 3.68d | 3.78 |
| LR | 3.27 | 3.54 | 3.46 | 3.42c | 3.46 | 3.59 | 3.55 | 3.54d | 3.48 |
| Mean | 3.35aa | 3.80bb | 4.23cc | - | 3.56dd | 3.99dd | 3.88dd | - | - |
| LSD (0.01) | Lime = 0.32, Zn level = 0.22, FYM level = ns, Lime x Zn level =0. 58,  Lime x FYM level = ns, Zn level x FYM level =ns ||||||||| |

**Zn concentration (mg kg$^{-1}$)**

| | | | | | | | | | |
|---|---|---|---|---|---|---|---|---|---|
| No lime | 62.2 | 85.0 | 119 | 88.7a | 71.0 | 86.2 | 126 | 94.4f | 91.6 |
| 1/10$^{th}$ LR | 60.4 | 78.4 | 105 | 81.3b | 70.7 | 84.3 | 116 | 90.3g | 85.8 |
| 1/3$^{rd}$ LR | 55.3 | 68.9 | 94.8 | 73.0c | 71.6 | 77.3 | 97.9 | 82.3h | 77.6 |
| 2/3$^{rd}$ LR | 47.8 | 66.5 | 75.2 | 63.2d | 52.4 | 69.5 | 80.2 | 67.4i | 65.3 |
| LR | 39.7 | 60.6 | 64.8 | 55.0e | 44.8 | 62.6 | 70.6 | 59.4j | 57.2 |
| Mean | 53.1aa | 71.9bb | 91.8cc | - | 62.1dd | 76.0ee | 98.1ff | - | - |
| LSD (0.01) | Lime = 1.80, Zn level = 0.20, FYM level = 1.50, Lime x Zn level =2.10,  Lime x FYM level = 3.80, Zn level x FYM level = 2.10 ||||||||| |

**Zn uptake (mg pot$^{-1}$)**

| | | | | | | | | | |
|---|---|---|---|---|---|---|---|---|---|
| No lime | 0.18 | 0.30 | 0.50 | 0.33a | 0.25 | 0.32 | 0.44 | 0.34e | 0.33 |
| 1/10$^{th}$ LR | 0.20 | 0.31 | 0.47 | 0.33a | 0.24 | 0.34 | 0.49 | 0.36f | 0.34 |
| 1/3$^{rd}$ LR | 0.21 | 0.30 | 0.43 | 0.31b | 0.27 | 0.37 | 0.44 | 0.36f | 0.34 |
| 2/3$^{rd}$ LR | 0.17 | 0.24 | 0.33 | 0.25c | 0.19 | 0.26 | 0.30 | 0.25g | 0.25 |
| LR | 0.13 | 0.21 | 0.23 | 0.19d | 0.15 | 0.23 | 0.25 | 0.21h | 0.20 |
| Mean | 0.18aa | 0.27bb | 0.39cc | - | 0.22dd | 0.30ee | 0.38ff | - | - |
| LSD (0.01) | Lime = 0.03, Zn level = 0.11, FYM level = 0.02, Lime x Zn level =ns,  Lime x FYM level = 0.08, Zn level x FYM level = ns ||||||||| |

*Letters indicate observed differences among the means of different treatments

**T able 3** Soil pH, EC and OC content as influence by FYM, lime and Zn application.

| Treatments | No FYM | | | | FYM (10 t ha$^{-1}$) | | | | Overall mean |
|---|---|---|---|---|---|---|---|---|---|
| | No Zn | 2.5 mg Zn kg$^{-1}$ | 5.0 mg Zn kg$^{-1}$ | Mean | No Zn | 2.5 mg Zn kg$^{-1}$ | 5.0 mg Zn kg$^{-1}$ | Mean | |
| Hariharapur series | | | | | | | | | |
| | | | | pH | | | | | |
| No lime | 4.56 | 4.57 | 4.61 | 4.58a | 5.16 | 5.10 | 5.34 | 5.20f | 4.89 |
| 1/10$^{th}$ LR | 4.80 | 5.01 | 4.83 | 4.88b | 5.46 | 5.42 | 5.44 | 5.44f | 5.16 |
| 1/3$^{rd}$ LR | 5.69 | 6.14 | 5.57 | 5.80c | 5.93 | 6.49 | 5.97 | 6.13g | 5.97 |
| 2/3$^{rd}$ LR | 6.45 | 6.53 | 6.62 | 6.53d | 6.92 | 7.08 | 6.57 | 6.86h | 6.70 |
| LR | 7.23 | 7.25 | 6.99 | 7.16e | 7.37 | 7.17 | 7.38 | 7.31h | 7.23 |
| Mean | 5.75aa | 5.90aa | 5.72aa | - | 6.17bb | 6.25bb | 6.14bb | - | - |
| LSD (0.01) | Lime = 0.19, Zn level = ns, FYM level = 0.25, Lime x Zn level = ns, Lime x FYM level = 0.51, Zn level x FYM level = ns | | | | | | | | |
| | | | | EC (dS m$^{-1}$) | | | | | |
| No lime | 0.14 | 0.11 | 0.13 | 0.13a | 0.13 | 0.15 | 0.14 | 0.14a | 0.13 |
| 1/10$^{th}$ LR | 0.14 | 0.10 | 0.10 | 0.12a | 0.15 | 0.11 | 0.12 | 0.13a | 0.12 |
| 1/3$^{rd}$ LR | 0.13 | 0.13 | 0.11 | 0.12a | 0.12 | 0.10 | 0.14 | 0.12a | 0.12 |
| 2/3$^{rd}$ LR | 0.12 | 0.13 | 0.11 | 0.12a | 0.12 | 0.15 | 0.10 | 0.12a | 0.12 |
| LR | 0.13 | 0.14 | 0.12 | 0.13a | 0.15 | 0.14 | 0.15 | 0.15a | 0.14 |
| Mean | 0.13aa | 0.12aa | 0.11aa | - | 0.13aa | 0.13aa | 0.13aa | - | 0.13 |
| LSD (0.01) | Lime = ns, Zn level = ns, FYM level = ns, Lime x Zn level = ns, Lime x FYM level = ns, Zn level x FYM level = ns | | | | | | | | |
| | | | | OC (%) | | | | | |
| No lime | 0.26 | 0.27 | 0.25 | 0.26a | 0.32 | 0.37 | 0.34 | 0.34b | 0.30 |
| 1/10$^{th}$ LR | 0.27 | 0.24 | 0.27 | 0.26a | 0.33 | 0.34 | 0.39 | 0.35b | 0.31 |
| 1/3$^{rd}$ LR | 0.25 | 0.24 | 0.27 | 0.25a | 0.31 | 0.36 | 0.37 | 0.35b | 0.30 |
| 2/3$^{rd}$ LR | 0.27 | 0.25 | 0.23 | 0.25a | 0.30 | 0.34 | 0.32 | 0.32b | 0.29 |
| LR | 0.24 | 0.21 | 0.22 | 0.22a | 0.25 | 0.34 | 0.33 | 0.31b | 0.27 |
| Mean | 0.26aa | 0.24aa | 0.25aa | - | 0.30bb | 0.35bb | 0.35bb | - | - |
| LSD (0.01) | Lime = ns, Zn level = ns, FYM level = 0.03, Lime x Zn level = ns, Lime x FYM level = ns, Zn level x FYM level = ns | | | | | | | | |

Debatoli series

**pH**

| | | | | | | | | | |
|---|---|---|---|---|---|---|---|---|---|
| No lime | 5.88 | 5.85 | 5.77 | 5.83a | 6.14 | 6.17 | 6.45 | 6.25f | 6.04 |
| 1/10th LR | 5.93 | 5.88 | 5.94 | 5.92b | 6.28 | 6.42 | 6.56 | 6.42f | 6.17 |
| 1/3rd LR | 6.38 | 6.21 | 6.21 | 6.27c | 6.44 | 6.57 | 6.58 | 6.53f | 6.40 |
| 2/3rd LR | 6.64 | 6.67 | 6.6 | 6.64d | 6.76 | 6.75 | 6.65 | 6.73g | 6.68 |
| LR | 6.96 | 6.99 | 6.9 | 6.95e | 7.27 | 6.87 | 7.14 | 7.09g | 7.02 |
| Mean | 6.36aa | 6.32aa | 6.28aa | - | 6.58bb | 6.56bb | 6.67bb | - | - |

LSD (0.01)    Lime  = 0.17, Zn level  = ns, FYM level = 0.20, Lime x Zn level = ns,  Lime x FYM level = 0.47, Zn level x FYM level = ns

**EC (dS m$^{-1}$)**

| | | | | | | | | | |
|---|---|---|---|---|---|---|---|---|---|
| No lime | 0.23 | 0.22 | 0.27 | 0.24a | 0.21 | 0.26 | 0.23 | 0.23a | 0.24 |
| 1/10th LR | 0.27 | 0.27 | 0.23 | 0.25a | 0.21 | 0.23 | 0.20 | 0.21a | 0.24 |
| 1/3rd LR | 0.23 | 0.23 | 0.24 | 0.23a | 0.17 | 0.29 | 0.25 | 0.24a | 0.24 |
| 2/3rd LR | 0.23 | 0.21 | 0.21 | 0.21a | 0.23 | 0.19 | 0.24 | 0.22a | 0.22 |
| LR | 0.24 | 0.17 | 0.29 | 0.23a | 0.19 | 0.30 | 0.26 | 0.25a | 0.24 |
| Mean | 0.24aa | 0.22aa | 0.25aa | - | 0.20aa | 0.25aa | 0.24aa | - | - |

LSD (0.01)    Lime  = ns, Zn level  = ns, FYM level = 0.04, Lime x Zn level = ns,  Lime x FYM level = ns, Zn level x FYM level = ns

**OC (%)**

| | | | | | | | | | |
|---|---|---|---|---|---|---|---|---|---|
| No lime | 0.21 | 0.28 | 0.22 | 0.24a | 0.22 | 0.29 | 0.30 | 0.27b | 0.25 |
| 1/10th LR | 0.22 | 0.22 | 0.21 | 0.22a | 0.28 | 0.28 | 0.28 | 0.28b | 0.25 |
| 1/3rd LR | 0.21 | 0.25 | 0.24 | 0.23a | 0.28 | 0.26 | 0.29 | 0.28b | 0.26 |
| 2/3rd LR | 0.18 | 0.22 | 0.25 | 0.21a | 0.31 | 0.25 | 0.28 | 0.28b | 0.25 |
| LR | 0.21 | 0.25 | 0.26 | 0.24a | 0.28 | 0.30 | 0.28 | 0.28b | 0.26 |
| Mean | 0.21aa | 0.24aa | 0.24aa | - | 0.27bb | 0.27bb | 0.29bb | - | - |

LSD (0.01)    Lime  = ns, Zn level  = ns, FYM level = 0.04, Lime x Zn level = ns,  Lime x FYM level = ns, Zn level x FYM level = ns

*Letters indicate observed differences among the means of different treatments

**Table 4** Effect of FYM, lime and Zn application on extractable Zn in soils.

| Treatments | No FYM | | | | FYM (10 t ha$^{-1}$) | | | | |
|---|---|---|---|---|---|---|---|---|---|
| | No Zn | 2.5 mg Zn kg$^{-1}$ | 5.0 mg Zn kg$^{-1}$ | Mean | No Zn | 2.5 mg Zn kg$^{-1}$ | 5.0 mg Zn kg$^{-1}$ | Mean | Overall mean |
| Hariharapur series | | | | | | | | | |
| | | | | DTPA-Zn (mg kg$^{-1}$) | | | | | |
| No lime | 0.40 | 1.44 | 2.95 | 1.60a | 0.88 | 1.68 | 3.21 | 1.92f | 1.76 |
| 1/10$^{th}$ LR | 0.40 | 1.24 | 2.30 | 1.31b | 0.66 | 1.67 | 3.20 | 1.84f | 1.58 |
| 1/3$^{rd}$ LR | 0.38 | 1.06 | 1.64 | 1.03c | 0.61 | 1.62 | 2.68 | 1.64fh | 1.33 |
| 2/3$^{rd}$ LR | 0.37 | 0.86 | 1.45 | 0.89d | 0.44 | 1.59 | 2.55 | 1.53gh | 1.21 |
| LR | 0.34 | 0.77 | 1.25 | 0.79e | 0.44 | 1.27 | 2.53 | 1.41gh | 1.10 |
| Mean | 0.38aa | 1.08bb | 1.92cc | - | 0.61dd | 1.57ee | 2.83ff | - | - |
| LSD (0.01) | Lime = 0.02, Zn level = 0.25, FYM level = 0.20, Lime x Zn level =0. 35, Lime x FYM level = 0.28, Zn level x FYM level = 0.47 | | | | | | | | |
| | | | | Mehlich 1-Zn (mg kg$^{-1}$) | | | | | |
| No lime | 0.78 | 1.68 | 3.85 | 2.10a | 1.23 | 3.70 | 5.08 | 3.34f | 2.72 |
| 1/10$^{th}$ LR | 0.77 | 1.66 | 3.74 | 2.06b | 1.17 | 3.20 | 4.88 | 3.08f | 2.57 |
| 1/3$^{rd}$ LR | 0.74 | 1.50 | 3.27 | 1.84c | 1.05 | 2.64 | 4.79 | 2.83gi | 2.33 |
| 2/3$^{rd}$ LR | 0.66 | 1.48 | 2.26 | 1.47d | 1.03 | 2.54 | 4.49 | 2.69hi | 2.08 |
| LR | 0.51 | 1.24 | 1.92 | 1.22e | 0.94 | 2.54 | 4.25 | 2.58hi | 1.90 |
| Mean | 0.69aa | 1.51bb | 3.01cc | - | 1.09dd | 2.92ee | 4.70ff | - | - |
| LSD (0.01) | Lime = 0.10, Zn level = 0.42, FYM level = 0.25, Lime x Zn level =0. 55, Lime x FYM level = 0.37, Zn level x FYM level = 0.70 | | | | | | | | |
| | | | | 0.1 M HCl-Zn (mg kg$^{-1}$) | | | | | |
| No lime | 0.90 | 2.50 | 4.62 | 2.67a | 1.50 | 3.81 | 6.24 | 3.85f | 3.26 |
| 1/10$^{th}$ LR | 0.89 | 2.31 | 4.61 | 2.60b | 1.34 | 3.72 | 6.20 | 3.75fi | 3.18 |
| 1/3$^{rd}$ LR | 0.84 | 2.25 | 4.28 | 2.46c | 1.33 | 3.39 | 5.68 | 3.47gi | 2.96 |
| 2/3$^{rd}$ LR | 0.84 | 2.18 | 3.94 | 2.32d | 1.22 | 3.05 | 5.62 | 3.30g | 2.81 |
| LR | 0.84 | 1.93 | 3.91 | 2.23e | 1.06 | 3.03 | 5.43 | 3.17g | 2.70 |

| | | | | | | | | | |
|---|---|---|---|---|---|---|---|---|---|
| Mean | 0.86aa | 2.23bb | 4.27cc | - | 1.29dd | 3.40ee | 5.83ff | - | - |
| LSD (0.01) | Lime = 0.02, Zn level = 0.30, FYM level = 0.27, Lime x Zn level =0. 37,  Lime x FYM level = 0.30, Zn level x FYM level = 0.60 | | | | | | | | |

ABDTPA-Zn (mg kg$^{-1}$)

| | | | | | | | | | |
|---|---|---|---|---|---|---|---|---|---|
| No lime | 0.71 | 2.03 | 4.06 | 2.27a | 1.16 | 2.54 | 3.98 | 2.56f | 2.42 |
| 1/10$^{th}$ LR | 0.68 | 1.98 | 3.19 | 1.95b | 1.11 | 2.43 | 3.92 | 2.49f | 2.22 |
| 1/3$^{rd}$ LR | 0.59 | 1.70 | 2.62 | 1.64c | 1.00 | 2.43 | 3.84 | 2.42f | 2.03 |
| 2/3$^{rd}$ LR | 0.52 | 1.52 | 2.29 | 1.44d | 0.95 | 2.37 | 3.61 | 2.31f | 1.88 |
| LR | 0.49 | 1.25 | 2.12 | 1.29e | 0.93 | 2.21 | 3.31 | 2.15f | 1.72 |
| Mean | 0.60aa | 1.70bb | 2.85cc | - | 1.03dd | 2.40ee | 3.73ff | - | - |
| LSD (0.01) | Lime = 0.05, Zn level = 0.28, FYM level = 0.32, Lime x Zn level =0. 32,  Lime x FYM level = 0.41, Zn level x FYM level = 0.62 | | | | | | | | |

Debatoli series

DTPA-Zn (mg kg$^{-1}$)

| | | | | | | | | | |
|---|---|---|---|---|---|---|---|---|---|
| No lime | 1.45 | 2.62 | 3.29 | 2.45a | 1.63 | 2.80 | 4.33 | 2.92f | 2.69 |
| 1/10$^{th}$ LR | 1.30 | 2.32 | 2.93 | 2.18b | 1.37 | 2.54 | 4.01 | 2.64fh | 2.41 |
| 1/3$^{rd}$ LR | 1.08 | 1.94 | 2.91 | 1.98bd | 1.32 | 2.37 | 3.79 | 2.49fh | 2.24 |
| 2/3$^{rd}$ LR | 0.99 | 1.78 | 2.80 | 1.86cd | 1.08 | 2.25 | 2.95 | 2.09gh | 1.98 |
| LR | 0.77 | 1.72 | 2.73 | 1.74c | 0.99 | 2.21 | 2.48 | 1.89g | 1.82 |
| Mean | 1.12aa | 2.08bb | 2.93cc | - | 1.28dd | 2.43ee | 3.51ff | - | - |
| LSD (0.01) | Lime = 0.21, Zn level = 0.50, FYM level = 0.35, Lime x Zn level =0. 75,  Lime x FYM level = 0.78, Zn level x FYM level = 0.98 | | | | | | | | |

Mehlich 1-Zn (mg kg$^{-1}$)

| | | | | | | | | | |
|---|---|---|---|---|---|---|---|---|---|
| No lime | 1.73 | 3.61 | 6.78 | 4.04a | 2.64 | 4.78 | 6.78 | 4.73a | 4.39 |
| 1/10$^{th}$ LR | 1.63 | 3.60 | 6.59 | 3.94b | 2.44 | 4.20 | 6.28 | 4.31b | 4.12 |
| 1/3$^{rd}$ LR | 1.51 | 3.44 | 6.12 | 3.69c | 2.42 | 4.10 | 6.21 | 4.24b | 3.97 |
| 2/3$^{rd}$ LR | 1.49 | 3.33 | 4.13 | 2.98d | 2.40 | 4.06 | 5.69 | 4.05b | 3.52 |
| LR | 1.26 | 3.15 | 4.06 | 2.82e | 2.37 | 3.74 | 5.46 | 3.86b | 3.34 |
| Mean | 1.53aa | 3.43bb | 5.54cc | - | 2.45dd | 4.18ee | 6.08ff | - | - |
| LSD (0.01) | Lime = 0.09, Zn level = 0.50, FYM level = 0.28, Lime x Zn level =0. 45,  Lime x FYM level = 0.42, Zn level x FYM level = 0.85 | | | | | | | | |

**0.1 M HCl-Zn (mg kg$^{-1}$)**

| | | | | | | | | | |
|---|---|---|---|---|---|---|---|---|---|
| No lime | 2.35 | 4.26 | 4.66 | 3.76a | 2.80 | 4.54 | 6.69 | 4.68e | 4.22 |
| 1/10$^{th}$ LR | 2.32 | 4.42 | 5.34 | 4.03b | 2.75 | 4.70 | 6.93 | 4.79e | 4.41 |
| 1/3$^{rd}$ LR | 2.22 | 4.40 | 6.07 | 4.23c | 2.86 | 5.25 | 7.61 | 5.24f | 4.74 |
| 2/3$^{rd}$ LR | 2.23 | 3.87 | 7.46 | 4.52d | 2.91 | 5.14 | 7.01 | 5.02f | 4.77 |
| LR | 2.22 | 4.53 | 6.96 | 4.57d | 2.85 | 6.06 | 7.79 | 5.57g | 5.07 |
| Mean | 2.27aa | 4.30bb | 6.10cc | - | 2.83dd | 5.14ee | 7.21ff | - | - |
| LSD (0.01) | Lime = 0.06, Zn level = 0.35, FYM level = 0.37, Lime x Zn level =0. 45,  Lime x FYM level = 0.45, Zn level x FYM level = 0.79 ||||||||| |

**ABDTPA-Zn (mg kg$^{-1}$)**

| | | | | | | | | | |
|---|---|---|---|---|---|---|---|---|---|
| No lime | 2.10 | 3.19 | 4.23 | 3.18a | 2.12 | 3.34 | 5.17 | 3.54e | 3.36 |
| 1/10$^{th}$ LR | 1.82 | 3.46 | 4.19 | 3.16a | 1.98 | 3.37 | 5.89 | 3.75e | 3.46 |
| 1/3$^{rd}$ LR | 1.61 | 2.77 | 4.60 | 2.99b | 1.93 | 3.46 | 5.17 | 3.52e | 3.26 |
| 2/3$^{rd}$ LR | 1.36 | 2.05 | 5.12 | 2.84c | 1.75 | 3.02 | 4.26 | 3.01f | 2.93 |
| LR | 1.22 | 2.17 | 4.22 | 2.54d | 1.53 | 3.36 | 4.42 | 3.10f | 2.82 |
| Mean | 1.62aa | 2.73bb | 4.47cc | - | 1.86dd | 3.31ee | 4.98ff | - | - |
| LSD (0.01) | Lime = 0.10, Zn level = 0.35, FYM level = 0.20, Lime x Zn level =0. 47,  Lime x FYM level = 0.40, Zn level x FYM level = 0.70 ||||||||| |

*Letters indicate observed differences among the means of different treatments

**Table 5** Pearson's correlation coefficient values revealing relationship among soil properties, dry matter yield, Zn concentration, Zn uptake and extracted Zn in soils (n = 90).

| | pH | EC | OC | Dry matter yield | Zn conc. | Zn uptake | DTPA-Zn | Mehlich 1-Zn | 0.1 M HCl-Zn | ABDTPA-Zn |
|---|---|---|---|---|---|---|---|---|---|---|
| | | | | | Hariharapur series | | | | | |
| pH | 1 | | | | | | | | | |
| EC | 0.058 | 1 | | | | | | | | |
| OC | -0.089 | -0.084 | 1 | | | | | | | |
| Dry matter yield | 0.059 | 0.093 | 0.221* | 1 | | | | | | |
| Zn conc. | -0.590** | -0.029 | 0.232* | 0.047 | 1 | | | | | |
| Zn uptake | -0.397** | 0.036 | 0.294** | 0.605** | 0.792** | 1 | | | | |
| DTPA-Zn | 0.010 | -0.073 | 0.211* | 0.391** | 0.610** | 0.523** | 1 | | | |
| Mehlich 1-Zn | 0.130 | -0.045 | 0.272** | 0.281** | 0.510** | 0.545** | 0.897** | 1 | | |
| 0.1 M HCl-Zn | 0.046 | -0.076 | 0.242* | 0.260* | 0.633** | 0.626** | 0.871** | 0.929** | 1 | |
| ABDTPA-Zn | -0.011 | -0.013 | 0.136 | 0.285** | 0.656** | 0.673** | 0.887** | 0.922** | 0.923** | 1 |
| | | | | | Debatoli series | | | | | |
| pH | 1 | | | | | | | | | |
| EC | 0.032 | 1 | | | | | | | | |
| OC | 0.113 | -0.098 | 1 | | | | | | | |
| Dr matter yield | -0.154 | 0.096 | 0.011 | 1 | | | | | | |
| Zn conc. | -0.343** | 0.042 | 0.158 | 0.384** | 1 | | | | | |
| Zn uptake | -0.326** | 0.086 | 0.110 | 0.727** | 0.905** | 1 | | | | |
| DTPA-Zn | -0.087 | 0.061 | 0.290** | 0.133 | 0.741** | 0.715** | 1 | | | |
| Mehlich 1-Zn | 0.168 | 0.091 | 0.317** | 0.330** | 0.589** | 0.568** | 0.811** | 1 | | |
| 0.1 M HCl-Zn | 0.188 | 0.130 | 0.294** | 0.333** | 0.562** | 0.545** | 0.822** | 0.937** | 1 | |
| ABDTPA-Zn | -0.074 | 0.108 | 0.193 | 0.419** | 0.772** | 0.748** | 0.889** | 0.890** | 0.887** | 1 |

$*p \leq 0.05$; $**p \leq 0.01$.

Fig. 1.  Dry matter yield, Zn concentration and Zn uptake by maize as influenced by interaction of Zn application and lime rate in Hariharapur and Debatoli series. Error bars represent ± SE.

Fig. 2.  Extractable Zn by different extractants as influenced by interaction of Zn application and lime rate in Hariharapur and Debatoli series. Error bars represent ± SE.

[Figure]

[Figure]

[Figure]

[Figure]

[Figure]

Fig. 1.  Dry matter yield, Zn concentration and Zn uptake by maize as influenced by interaction of Zn application and lime rate in Hariharapur and Debatoli series. Error bars represent ± SE.

[Figure]

[Figure]

**(c)**

[Figure]

**(d)**

[Figure]

Fig. 2. Extractable Zn by different extractants as influenced by interaction of Zn application and lime rate in Hariharapur and Debatoli series. Error bars represent ± SE.

---

## Editor Comment (EC2) · J. A. Gomez (Editor) · 9 Nov 2016

Lime and zinc application influence soil zinc availability, dry matter yield and zinc uptake by maize grown on Alfisols .

My overall assessment of the revised version of this manuscript, published on 07/10/2016,is that it addresses most of the questions raised by the reviewers , albeit some minor ones remain.

First I list those points raised by both reviewers that, in my view, have been address properly.  1- Make a clear definition of the situation and objectives of the experiment to avoid the impression that the manuscript presents three related (but not properly

coordinated) experiments. The changes made by the authors with a new version of introduction and M&Methods, including clearly stated objectives have addressed properly this issue. 2- The need for a better description of the experiments and methods, among them a better description of the pots and manure used. This has also been properly addressed. 3- Elimination of duplication of results presented in Tables and Graphs. This has also been properly addressed with the elimination of some Figures.

The fourth, and major, issue raised by both reviewers was the need for a much improved presentation and discussion of the results. From reviewer 1 I quote, among some of them, "a) Irrelevant results were included (e.g., adding farmyard manure increased the soil OC, the addition of lime increased soil pH, ..., adding Zn (and FYM) to soil increased Zn concentration in plant)" b) "No critical levels of Zn in soil and/or plant tissues were indicated". c) "Was the concentration of Zn in plants for unfavorable treatment below the critical values (literature)? Was there observed Zn deficiency symptoms in the plants with lower Zn concentration? –" d) "The Tables do not clarify the results of statistical analysis (comparison of means). The differences observed between means of the different treatments should be indicated by adding the corresponding letter (a, b, c...) to each mean value" I also quote some of the major comments by reviewer 2. "a) To evaluate the interaction between the two variables (FYM, and lime dose) in the statistical model" b) "Been more critical when extrapolating optimum lime application from initial stages of the crop. 60 days in a 4 l pot, to an adult plant exploring a larger soil volume not considered in their experiment" c) "Use same symbol in the same soil to facilitate identification to reader in Figures."

Many of these have been addressed but a few of them remain problematic. These issues can be summarized in:

a- The statistical results presented in Tables 2 to 4 remain difficult to understand. A better explanation of the statistical model used (a three factors analysis of variance with interaction among these factors by pairs as it seems reading the Tables?) should be included in the material and methods section (section 2.3) and also in an improved, more comprehensive, caption for these Tables. In its present form it is clear how to interpretate the difference between means for any given FYM and soil according to the LR and the Zinc levels. However it remains complicated to understand the statistical significance (or not) of the different treatments, and their interactions. For instance, in Table 2 Hariharapur soil series, dry matter results, what is the interpretation of the LSD(0.01) for Lime x FYM level = 0.61 in your results. Is it a significant or a non-significant interaction across the experimental results? This need be revised. Also indicate in the caption that double letters (e.g. aa, bb and so on) are used as single symbol for the mean values across the Zinc levels.

2- Graphs remain non-intuitive. Please try to use a more straightforward design. For instance use the same color in all the lines, use the symbol to identify the soil series. I mean the same symbol for the same soil series (e.g. solid square for Hariharapur and non-solid circle for Debatoli) and use the line style to differentiate between added or non-added FYM (e.g. continuous for added FYM and dotted for non-added FYM). Using for different colors and symbols makes more complicated a quick interpretation of the graphs. 3- Please in the conclusions be a bit more cautious about the need to extrapolate these results to field recommendations. 4- Some misspellings and sections that should be double checked for proper English editing remain. I have added same comments in the PDF version of the manuscript in order to help you to deal with these final issues, and also to improve the edition in English . So my recommendation is that this revised version merits publication in SOIL but after the authors revise this, now minor changes, before final publication.

Please also note the supplement to this comment:
http://www.soil-discuss.net/soil-2016-41/soil-2016-41-EC2-supplement.pdf

**Supplement:**

Dear Editor and Referees,

We thank you very much for providing constructive and useful suggestion for our
manuscript. We have modified the manuscript incorporating the suggestions. The details of
our responses and revisions are given below.

Comments of Editor:- Both reviewers indicate that the manuscript is of limited merit, in its
current form, for publication in soil. Both indicates concisely which are the major points to be
addressed for a possible resubmission, and one of them also expand these comments in an
annotated version of the manuscript. I agree with both reviewers and endorse their
recommendation for rejection, considering possible resubmission if the authors considered
that they can rework the manuscript to address the comments indicated by the two reviewers.

Our response:- We have revised the manuscript according to the comments of both of the
reviewers. We feel that the quality of the manuscript has improved by incorporating the
suggestions of the reviewers. We have attached the revised manuscript for your kind perusal.
Thanks a lot to you and to the reviewers.

Comments of Anonymous referee # 1:-

Comment:- The article is of limited scientific merit for publication in "SOIL". Some
suggestions are given below: 1. Introduction. - It is not clear as to where the research is
leading to: the problem of soil acidity regarding to crop production? Zinc-deficient soils for
crops? the combination of acid soils and zinc-deficient soils? You should clearly define the
starting situation that generates the need for research and avoid redundancy in drafting the
text.

Our response:- Thank you sir for this suggestion. We have modified the introduction part of
the manuscript and deleted unwanted portions. The present study was carried out to examine
the influence of lime and farmyard manure and Zn addition on dry matter yield, Zn
concentration and uptake by maize and soil properties and extractable Zn by different
extractants in acid soils. The information would be useful for assessment of extractable Zn
and its management in acid soils where Zn availability is one of the main problems and Zn
application is imminent and application of lime and FYM is a common practice to obtain
higher crop yield. This information has been incorporated in introduction part to bring the
clarity of the study.

Comment:- The evaluation of the different Zn-extractants is not relevant for this research,
although this assessment could be the subject of another more specific work.

Our response: Yes, we do agree with the reviewer that evaluation of different Zn- extractants
is more specific work. We have used different extractants to extract Zn in post harvest soil in
order to establish relationship among the extracted Zn by different extractants and dry matter
yield and Zn concentration and uptake by maize. Based on this relationship, we have
identified the suitability of different extracts for extraction of Zn in acid soils.

Comment:- Objectives should be reworked once the problem has been well defined (and also the title of the article, according to them).

Our response: We have clarified the problem in the introduction part of the manuscript as mentioned earlier. We have also modified the title of the article and the objectives of the study.

Comment:-What relevance does the assessment have of the effect of the application of farmyard manure on the soil OC content (since OM was added)? (as well as the influence of lime application in pH). This is not relevant.

Our response: We agree with the reviewer. We have assessed the soil properties like Ph, EC and OC content along with extracted Zn in post-harvest soils to visualize the influence lime and FYM addition and to assess the relationship among the soil properties, dry matter yield and Zn concentration and uptake by maize.

Comment:- Materials and methods The experiment was in pots, of which the diameter was not indicated, although the weight of soil was.

Our response:- We have included diameter of each pot in the manuscript.

Comment:- However, the added amount of farmyard manure was expressed in t/ha. You should indicate the amount (g) of FYM applied to each pot for to know the nutrients (i.e., Zn) added to the soil with FYM as you are evaluating Zn extraction by crop (and by extractants) by varying Zn and lime doses.

Our response: Yes, we agree with the reviewer. We have indicated the amount of FYM in g added in the manuscript.

 Comment:-What type of farmyard manure is it? The results of chemical analysis indicate FYM: OC 0.12.

Our response:- Locally available farmyard manure was used for the study and it was decomposed mixture of left over fodder (predominantly) fed to farm animals, animal dung and animal urine. There was a typo error in providing the OC content of FYM. It is 0.22% instead of 0.12% as mentioned earlier. We have corrected it in the manuscript and the information has been provided in Table 1.

Comment:-Results - Irrelevant results were included (e.g., adding farmyard manure increased the soil OC, the addition of lime increased soil pH, ..., adding Zn (and FYM) to soil increased Zn concentration in plant).

Our response: We have modified the result as per the suggestions. We have also changed the sequence of the results presented in the manuscript to make it more relevant.

 Comment:- No critical levels of Zn in soil and/or plant tissues were indicated. Was the concentration of Zn in plants for unfavorable treatment below the critical values (literature)? Was there observed Zn deficiency symptoms in the plants with lower Zn concentration?

Our response:- We have include critical concentration of DTPA-Zn in soils (0.8 mg kg$^{-1}$)(in table 1) and plant tissues in the manuscript. We have also compared the values of Zn concentration in plant tissues under different treatments with critical values available in literature. We have mentioned that the Zn concentration in maize under all the treatments were well above the critical Zn concentration of 15 to 22 mg kg$^{-1}$ for maize crop (Alloway,

2008) and no visual Zn deficiency symptoms in plants were recorded.

Comment:- The Figures presented are redundant (and unnecessary), since data are also shown in tables.

Our response: In agreement with the reviewer, we have deleted the figures no. 1 from the manuscript. We have modified the figures no. 2 and 3 as per the suggestions of the referee #2.

Comment:-The Tables do not clarify the results of statistical analysis (comparison of means).

The differences observed between means of the different treatments should be indicated by adding the corresponding letter (a, b, c...) to each mean value.

Our response: Yes it is correct. We have provided different letters to identify the observed differences between means in the tables.

Comments of Anonymous referee # 2:-

Comment:-My overall assessment of this manuscript is that although it covers a subject of potential interest to the journal, it does it without a clear objective and combining information on subjects that are well proven (e.g. liming) and very little in others (such as in a more detailed discussion of the interaction among treatments).

Our response:- We thank the reviewer for visualizing the importance of our study. We have modified the introduction part of the manuscript to clarify the problem and clearly stated the objectives of the study. Since liming and farmyard manure application is common by the farmers in acid soils, many researchers have worked in this line. But the information regarding the influence of lime, farmyard manure and Zn in acid soils on crop yield, Zn concentration in plant tissue and extracted Zn and their relationship is lacking. Therefore, the present study was carried out. We have tried our best to improve the discussion part of the manuscript by incorporating information about the significant interaction effects among the treatments.

Comment:-I have indicated several comments in the annotated version of the manuscript, but

I summarize here some major points in case the authors want to rework the manuscript for a possible resubmission.

Our response:-We have gone through the comments given in the annotated version of the manuscript. We have modified the manuscript as per the comments provided in the different parts of the manuscript.

Comment:-The article lacks clear objectives, stated at the end of the introduction. There are apparently three overlapping studies: 1. Field experiments, greenhouse pot experiments, and effect of the extractant used in determining Zn concentration. However, it is unclear how they are coordinated for a final objective, giving the impression of been three related (but not properly coordinated) experiments.

Our response: We agree with you that we have collected bulk soil from field to conduct green house study. The objective of the present study was to study the influence of lime and farmyard manure and Zn addition on dry matter yield, Zn concentration and uptake by maize and soil properties and extractable Zn by different extractants in acid soils. The information would be useful for assessment of extractable Zn and its management in acid soils where Zn availability is one of the main problems and Zn application is imminent and application of lime and FYM is a common practice to obtain higher crop yield. This information has been incorporated in introduction part to bring the clarity of the study. We have modified and properly coordinated the introduction part of the manuscript to make it better understandable.

Comment:-The manuscript might be reorganized and edited, particularly in the introduction and M&Methods to address this problem.

Our response:- We have modified and reorganized the introduction and material and method section of the manuscript and made it systematic.

Comment:- There is missing some key information in the material and methods sections (for instance a better definition of the soil sampling in the field studies, or the properties of the manure, . . .).

Our response: We have incorporated the information regarding soils collected from field and methods used for analysis of manures as per the comments given in the annotated version of the manuscript.

 Comment:-There are many other examples of these in the annotated version of the manuscript. They should be addressed.

Our response:- We have modified the manuscript as per the comments given.

Comment:-There is duplication in results presented in the Tables and Graphs while at the same time the statistical models uses (and in their major results, particularly in the case of interactions between variables) This should be addressed.

Our response: In agreement with the reviewer, we have deleted the figures no. 1 from the manuscript. We have modified the figures no. 2 and 3 as per the suggestions of the reviewers given in the manuscript. We have also tried our best to describe the results including the interaction effects of different treatments as per the statistical test used in the study.

Comment:- The discussion ad conclusions suffer the same lack of focus already mentioned in the overall organization for the manuscript. This should also been addressed.

Our response: We have modified the discussion and conclusion parts of the manuscript. Now we feel that is properly ordered and systematic.

Comment:-For these reasons my recommendation is that the manuscript should be returned to
the authors for major modifications before been reconsidered for possible publication.

Our response:- Thank you very much. We have modified the manuscript as per the
suggestions.

With above modifications, we are hereby submitting the revised manuscript for your kind
perusal.

With kind regards,

Sanjib Kumar Behera

**Effect of lime, farmyard manure and zinc application on soil properties, dry matter yield, zinc concentration and uptake by maize and extractable zinc in Alfisols**

**Sanjib K. Behera[a,*], Arvind K. Shukla[b], Brahma S. Dwivedi[c], Brij L. Lakaria[b]**

*ICAR-Indian Institute of Oil Palm Research, Pedavegi, West Godavari District, Andhra Pradesh 534450, India*

*ICAR-Indian Institute of Soil Science, Nabibagh, Berasia Road, Bhopal, Madhya Pradesh 462038, India*

*ICAR-Indian Agricultural Research Institute, Pusa, New Delhi, 110012, India*

*Corresponding author: sanjibkumarbehera123@gmail.com (S. K. Behera), ICAR-Indian Institute of Oil Palm Research, Pedavegi, West Godavari District, Andhra Pradesh 534450, India

ABSTRACT

Zinc (Zn) deficiency is widespread in all types of soils of world including acid soils affecting crop production and nutritional quality of edible plant parts. There is, however, limited information available regarding effects of lime and farmyard manure (FYM) and Zn addition to acid soils on dry matter yield, Zn concentration and uptake by maize (*Zea mays* L.) and soil properties and extractable Zn by different extractants. Green house pot experiments were carried out in two acid soils to study the effect of five levels of lime (0, 1/10 lime requirement (LR), 1/3 LR, 2/3 LR and LR), three levels of Zn concentration (0, 2.5 and 5.0 mg Zn kg$^{-1}$ soil) and two levels of FYM (0 and 10 t ha$^{-1}$) addition on dry matter yield, Zn concentration and uptake by maize plant grown up to 60 days and soil pH, EC and OC content and extractable Zn in soil. Lime rate of 1/3$^{rd}$ LR was found to be optimum as dry matter yield of maize increased significantly with lime application up to 1/3$^{rd}$ LR in soils of both the series and decreased subsequently. Addition of FYM with and without lime increased dry matter yield. Application of Zn up to 5.0 mg kg$^{-1}$ to soil increased dry matter yield with and without FYM application in soils of Hariharapur series. Addition of higher doses of lime significantly reduced Zn concentration in maize crop grown in soils of both the series. Mean Zn uptake values were at par for no lime, $1/10^{th}$ LR and $1/3^{rd}$ LR with and without FYM application and it was significantly higher than Zn uptake by $2/3^{rd}$ LR and LR

treatments. However, FYM application improved Zn uptake by maize crop. Increased level of lime application reduced Zn extracted by DTPA, Mehlich 1, 0.1 N HCl and ABDTPA

extractants. However, application of FYM along with lime improved Zn extraction. The amount of Zn extracted by different extractants followed the order DTPA-Zn < ABDTPA-

Zn < Mehlich-1 Zn < 0.1 M HCl. Zn extracted by different extractants like DTPA, ABDTPA,

Mehlich 1 and 0.1 M HCl was positively and significantly correlated amongst themselves and with dry matter yield, Zn concentration and Zn uptake by maize. Among the extractants,

ABDTPA was found to be the best extractant for extraction of Zn in acid soils.

*Keywords:* Alfisol, Dry matter yield, Farmyard manure, Lime, Zinc concentration

**1. Introduction**

[revised manuscript text omitted]

The information from the present study would be useful for assessment of extractable Zn and its management in acid soils where Zn availability is one of the main problems and Zn application is imminent and application of lime and FYM is a common practice. Keeping above facts in view, the present study was carried out (i) to evaluate the influence of lime, FYM and Zn addition on dry matter yield, Zn concentration and uptake by maize (*Zea mays* L.) crop and (ii) to evaluate the influence of lime, FYM and Zn addition to acid soils on soil pH, EC and OC content, extractable Zn as extracted by different extractants.

**2. Materials and methods**

*2.1 Soil and farmyard manure characteristics*

The bulk surface (0-15 cm depth) soils collected from Hariharpur series (Oxic Haplustalf, Alfisol (Soil Survey Staff, 2014)) and Debatoli series (Udic Rhodostalf, Alfisol (Soil Survey Staff, 2014)) of Bhubaneswar and Ranchi (India), respectively were used in the study. The collected soils were air dried and stone and debris were removed and then ground to pass a 2 mm sieve and analysed for selected properties (Table 1). Soil properties like pH and EC were determined done on 1: 2.5 soil water ratio (w/v) suspension using pH meter and EC meter following half an hour equilibrium (Jackson, 1973). Soil organic carbon (OC) content was estimated by chromic acid digestion-back titration method (Walkley and Black, 1934). The clay, silt and sand per cent of soils were determined by hydrometer method (Bouyoucos,

1962). Calcium carbonate ($CaCO_3$) content was determined by rapid titration method (Puri,

1930) and cation exchange capacity (CEC) by neutral normal ammonium acetate method (Richards, 1954). Lime requirement (LR) of the soil was estimated by extractant buffer method (Shoemaker et al., 1961). The plant available Zn in soils was extracted by DTPA

method (Lindsay and Norvell, 1978). Estimation of Zn concentration was done on the clear extract by atomic absorption spectrophotometer (AAS). After drying of FYM at 70 $^o$C for

24 h followed by grinding to pass through 20 mesh sieve, one gram of ground FYM was dry- ashed at 450 $^o$C for 2h. Ashed samples were extracted using 0.5 N HCl. Zn concentration was determined in filtered extracts. The total OC (loss on ignition), N (Kjeldahl method), P

(nitric-perchloric 9:4 digestion) and K (nitric-perchloric 9:4 digestion) concentrations in

FYM were estimated according to Tandon (2009) (Table 1).

*2.2 Green house study, soil and plant analysis*

Pot experiments were carried out in two Hariharapur and Debatoli series soils. The experiments were carried out in plastic pots (each with diameter of 20 cm) having 4 kg of soil with five levels of LR (0, 1/10 LR, 1/3 LR, 2/3 LR and LR), three levels of Zn concentration (0, 2.5 and 5.0 mg Zn kg$^{-1}$ soil) and two levels of fresh FYM (35% moisture) (0 and 4.5 g

FYM kg$^{-1}$ soil viz., 0 and 10 t FYM ha$^{-1}$). Locally available FYM was used for the study and it was decomposed mixture of left over fodder fed to farm animals, animal dung and urine.

All the pots received basal treatments of N-$P_2O_5$-$K_2O$ @ 150-60-40 kg ha$^{-1}$ (equivalent to

66.7-26.7-17.8 mg N-$P_2O_5$-$K_2O$ kg$^{-1}$ soil, respectively). Fertilizer N, P and K were applied through analytical grade urea, calcium dihydrogen orthophosphate and muriate of potash, respectively. Lime and Zn were added to soil through laboratory grade $CaCO_3$ and $ZnSO_4$

respectively.  All nutrients were mixed in soil thoroughly before sowing of seeds. The soil in each pot was then irrigated to field capacity with deionized water and kept for incubation for one week. Each treatment combination was replicated thrice in a factorial completely randomized design. Four seeds of cv. KH 101 of maize were sown in each pot.  Two seedlings of maize per each pot were maintained after emergence. Pots were irrigated with water daily as per requirement of water on weight basis to maintain the field capacity. Above- ground biomass of plants from each pot was harvested at the end of 60 days of growth.

Harvested above-ground biomass of each pot was washed in deionized water, and then dried in oven at 70 $^o$C for 48 h. After drying, dry matter yield (DMY) of each pot was recorded.

Dried plant material was  then ground  in a stainless steel Wiley mill, and digested in a di- acid mixture of $HNO_3$ and $HClO_4$ (Jackson, 1973). Zn concentration was then determined in aqueous extracts of the digested plant material by atomic absorption spectrophotometer (AAS). Zn uptake was calculated as DMY multiplied by the Zn concentration.

Soil sample from each pot were collected after harvesting of maize plants. Collected soil samples were processed and analyzed for pH, EC, OC content and DTPA-Zn concentration following the methods described above. The plant available Zn in soils was also extracted by

DTPA (Lindsay and Norvell, 1978), Mehlich 1 (Perkins, 1970), 0.1 M HCl   (Sorensen et al.,

1971) and ABDTPA (Soltanpour and Schwab, 1977) extractants by following the respective prescribed methods. Estimation of Zn concentration was done on the clear extract by AAS.

*2.3 Statistical analysis*

The data regarding soil properties, DMY, Zn concentration, Zn uptake and extracted Zn by different extractants   subjected to analysis of variance method (Gomez and Gomez 1984).

Least square difference (LSD) at $P \leq .01$ was used to compare among the treatment means.

Pearson's correlation coefficient values were estimated to establish relationship among soil properties, DMY, Zn concentration, Zn uptake and extracted Zn by different extractants.

**3. Results**

*3.1 Dry matter yield*

DMY of maize increased significantly with lime application up to $1/3^{rd}$ LR (Table 2, Fig. 1 a)

in soils of both the series. This indicated that lime application @ $1/3^{rd}$ of LR was optimum for these soils. Application of higher doses of lime ($2/3^{rd}$ LR and LR) did not result in increased

DMY. However, this finding needs to be verified by conducting field experiment. The mean

DMY in $1/3^{rd}$ LR treatment without FYM and with FYM was 139% and 149% of control respectively in Harihpur series soils. Similarly in Debatoli series soil, the mean DMY was

84% and 120% of control without and with FYM application respectively in combination with $1/3^{rd}$ LR. Application of graded doses of Zn upto 5.0 mg $kg^{-1}$ to soil increased DMY

with and without FYM application in Hariharapur series. Whereas in Debatoli series, application of graded doses of Zn up to 5 mg $kg^{-1}$ without FYM and  application of Zn @ 2.5

mg $kg^{-1}$ with FYM enhanced DMY.

*3.2 Zinc concentration and uptake by maize*

Addition of higher doses of lime significantly reduced Zn concentration in maize crop grown in soils of both the series (Table 2, Fig. 1 b). In contrast, application of Zn (@ 2.5 and 5.0 mg

$kg^{-1}$) and FYM (@ 10 t $ha^{-1}$) increased Zn concentration in maize crop significantly in soils of both the series (Table 2, Fig 1c). In soils of Hariharapur series, application Zn @ 2.5 and 5

mg $kg^{-1}$ without and with FYM augmented Zn concentration in maize by 67.5 and 93.5 to 109

% respectively, as compared to control (No Zn). Similarly, increased Zn concentrations of 22

to 35 and 58 to 73% were recorded with application of Zn @ 2.5 and 5 mg $kg^{-1}$ without and with FYM respectively in comparison to no Zn control in soils of Debatoli series. However, the Zn concentration in maize under all the treatments were well above the critical Zn concentration of 15 to 22 mg kg$^{-1}$ for maize crop (Alloway, 2008) and no visual Zn deficiency symptoms in plants were recorded. Mean Zn uptake values were at par for no lime, 1/10$^{th}$ LR

and 1/3$^{rd}$ LR with and without FYM application and it was significantly higher than Zn uptake by 2/3$^{rd}$ LR and LR treatments in soils of both the series (Table 2, Fig. 1 d). However,

Zn and FYM application improved Zn uptake by maize crop in soils of both series (Fig. 1 e).

Addition of Zn @ 2.5 and 5 mg kg$^{-1}$ enhanced Zn uptake by 67 to 100 and 122 to 150%

respectively as compared to no Zn control in soils of Hariharapur series. Whereas, the enhancements in Zn uptake were 36 to50, 73 to 117% due to application of Zn @ 2.5 and 5

mg kg$^{-1}$ respectively as compared to no Zn control in soils of Debatoli series.

*3.3 Soil properties*

Application of lime at different rates significantly increased pH in soils of both Hariharapur and Debatoli series (Table 3). With addition of graded doses of limes viz. from no lime,

1/10$^{th}$ LR, 1/3$^{rd}$ LR, 2/3$^{rd}$ LR and LR, soil pH increased from 4.58 to 7.16 (without FYM

addition) and from 4.89 to 7.23 (with FYM addition) in Hariharapur series and from 5.83 to

6.95 (without FYM addition) and from 6.04 to 7.02 (with FYM addition) in Debatoli series.

Application of FYM without lime increased soil pH in both soils (Table 3). Interaction effect of combined application of lime and FYM on soil pH was significant. Soil pH values obtained by addition of 2/3$^{rd}$ LR and LR along with FYM were at par. Addition of Zn did not have any effect on soil pH. Sole application of lime, FYM and Zn and their interaction did not influence soil EC levels in soils of both the series (Table 3). However application of FYM

increased soil OC content in soils of both series. Addition of lime and Zn and their interaction did not influence soil OC.

*3.4 Extractable zinc in post-harvest soil*

Data regarding amount Zn extracted by DTPA, Mehlich 1, 0.1 M HCl and ABDTPA

extractants in post harvest soil are given in Table 4 and Figure 2. Perusal of data revealed significant effect of individual application of lime, FYM and Zn and their interaction on extracted Zn by different extractants. The amount of extracted Zn by DTPA, Mehlich 1, and

ABDTPA extractants decreased with increased level of lime application in soils of both the series (Fig. 2 a, b, d). But addition of FYM (@ 10 t ha$^{-1}$) in combination of different levels of lime led to marked enhancement of extracted Zn by different extractants in both the soils compared to only application of different lime levels (Table 4). Application Zn at different levels viz. 2.5 and 5.0 mg kg$^{-1}$ with and without FYM increased the concentration of extracted Zn by the different extractants. The amount of Zn extracted by DTPA, Mehlich 1,

0.1 M HCl and ABDTPA extractant varied from 1.10 to 1.76, 1.90 to 2.72, 2.70 to 3.26 and

1.72 to 2.42 mg kg$^{-1}$ respectively, under different levels of lime application across FYM and

Zn application in soils of Hariharpur series. Whereas, the Zn extracted by DTPA, Mehlich 1,

0.1 M HCl and ABDTPA extractant varied from  1.82 to 2.69, 3.34 to 4.39, 4.22 to 5.07 and

2.82 to 3.36 mg kg$^{-1}$ respectively, under different levels of lime application across FYM and

Zn application in soils of Debatoli series. In both the series, the extracted Zn followed the order DTPA-Zn < ABDTPA-Zn < Mehlich1-Zn < 0.1 M HCl-Zn.

**4. Discussion**

Significant increase in DMY was recorded with application of lime up to 1/3$^{rd}$ LR. Increase in DMY with lime application up to 1/3$^{rd}$ LR may be ascribed to increase in soil pH and positive influence on nutrient availability in soil (Tisdale, 2005). Our finding is in line with the observations made by Barman et al. (2014) who reported lime application at 1/3rd LR

was optimum for obtaining cauliflower yield in Typic Fluvaquent soil of West Bengal, India.

There was reduction in DMY with lime application at 2/3$^{rd}$ LR and LR in soils of both the series. This may be ascribed to reduced availability Zn in soil with 2/3$^{rd}$ LR and LR rate of lime application and adverse effect on other soil properties. This needs to be verified by conducting filed experiment. 
[revised manuscript text omitted]

**5. Conclusion**

Lime application of 1/3LR was found to be optimum for amelioration of acid soil. The concentration of Zn in maize tissue and extracted Zn by different extractants like DTPA, Mehlich 1, 0.1 M HCl and ABDTPA in both the soils reduced with lime application. Application of FYM along with lime improved the Zn concentration in maize plant and extractable Zn in soils. Since DTPA, Mehlich 1, 0.1 M HCl and ABDTPA extractable Zn in soils of both the series were positively and significantly correlated with dry matter yield, Zn concentration and Zn uptake, these extractants could be used for extraction of Zn in acid soils. However based on higher correlation coefficient values, ABDTPA was found to be best extractant for extraction of Zn in acid soils.

**Acknowledgements**

The study was supported by the grant from Indian Council of Agricultural Research, New Delhi. We thank the Director of ICAR-Indian Institute of Soil Science, Bhopal, Madhya Pradesh, India for providing necessary facilities for conducting the research work. We acknowledge the help rendered by Ms. P. Singh, Mr. R. Singh and Mr. D. K. Verma during the execution of the work. The authors thank the topical editor and the anonymous reviewers for their useful suggestions for improvement of the manuscript.

**Table 1** Some selected characteristics of the experimental soils and farmyard manure.

| Characteristics | Experimental soils | |
| --- | --- | --- |
| | Hariharapur series | Debatoli series |
| Taxonomic classification | Oxic Haplustalfs | Udic Rhodustalfs |
| pH (1:2.5) | 4.50 | 5.80 |
| EC (dS m$^{-1}$) | 0.14 | 0.23 |
| Organic carbon (%) | 0.31 | 0.22 |
| Clay (%) | 12.1 | 14.2 |
| Silt (%) | 15.0 | 11.6 |
| Sand (%) | 73.2 | 75.1 |
| CaCO$_3$ (%) | 20.0 | 32.0 |
| CEC (cmol(p$^+$) kg$^{-1}$) | 3.90 | 5.10 |
| Lime requirement (g kg$^{-1}$) | 3.34 | 1.51 |
| DTAP-Zn (mg kg$^{-1}$) | 0.47 | 1.45 |
| | Farmyard manure | |
| Total organic carbon (%) | 0.22 | |
| Total N (%) | 0.48 | |
| Total P (%) | 0.10 | |
| Total K (%) | 0.55 | |
| Total Zn (mg kg$^{-1}$) | 12 | |

*Critical concentration of DTPA-Zn is 0.80 mg kg$^{-1}$

**T able 2** Effect of FYM, lime and Zn application on dry matter yield, Zn concentration and Zn uptake by maize.
[Figure]

| Treatments | No FYM | | | | FYM (10 t ha⁻¹) | | | | |
|---|---|---|---|---|---|---|---|---|---|
| | No Zn | 2.5 mg Zn kg⁻¹ | 5.0 mg Zn kg⁻¹ | Mean | No Zn | 2.5 mg Zn kg⁻¹ | 5.0 mg Zn kg⁻¹ | Mean | Overall mean |
| Hariharapur series | | | | | | | | | |
| | | | | Dry matter (g pot⁻¹) | | | | | |
| No lime | 1.64 | 2.02 | 2.04 | 1.90a | 2.06 | 2.60 | 2.23 | 2.30d | 2.10 |
| 1/10th LR | 2.43 | 2.37 | 2.16 | 2.32b | 2.21 | 2.74 | 2.66 | 2.53ef | 2.43 |
| 1/3rd LR | 2.88 | 2.87 | 2.96 | 2.83c | 2.57 | 2.89 | 3.66 | 2.98f | 2.91 |
| 2/3rd LR | 2.65 | 2.37 | 2.66 | 2.64c | 2.40 | 2.40 | 3.01 | 2.66d | 2.65 |
| LR | 1.77 | 2.06 | 2.52 | 2.12ab | 1.94 | 2.05 | 2.71 | 2.23d | 2.18 |
| Mean | 2.27aa | 2.34aa | 2.47bb | - | 2.23cc | 2.53cce | 2.85dde | - | - |
| LSD (0.01) | Lime = 0.30, Zn level = 0.11, FYM level = 0.25, Lime x Zn level =0. 50,  Lime x FYM level = 0.61, Zn level x FYM level = 0.42 | | | | | | | | |
| | | | | Zn concentration (mg kg⁻¹) | | | | | |
| No lime | 54.0 | 84.0 | 112 | 83.3a | 57.4 | 104 | 119 | 93.2e | 88.4 |
| 1/10th LR | 53.3 | 87.4 | 113 | 84.6a | 59.2 | 99.5 | 119 | 92.7e | 88.6 |
| 1/3rd LR | 38.5 | 63.5 | 75.0 | 59.0b | 46.3 | 72.8 | 80.0 | 66.4f | 62.7 |
| 2/3rd LR | 27.4 | 52.7 | 60.8 | 47.0c | 35.4 | 59.8 | 67.6 | 54.g | 50.6 |
| LR | 25.2 | 44.8 | 54.2 | 41.4d | 31.2 | 48.9 | 58.1 | 46.1h | 43.7 |
| Mean | 39.7aa | 66.5bb | 83.0cc | - | 45.9dd | 76.9ee | 88.8ff | - | - |
| LSD (0.01) | Lime = 3.50, Zn level = 0.11, FYM level = 2.00, Lime x Zn level =3.21,  Lime x FYM level = 5.70, Zn level x FYM level = 3.15 | | | | | | | | |
| | | | | Zn uptake (mg pot⁻¹) | | | | | |
| No lime | 0.11 | 0.14 | 0.23 | 0.16a | 0.12 | 0.27 | 0.26 | 0.22f | 0.19 |
| 1/10th LR | 0.13 | 0.21 | 0.24 | 0.19b | 0.13 | 0.27 | 0.32 | 0.24g | 0.22 |
| 1/3rd LR | 0.10 | 0.18 | 0.22 | 0.17c | 0.11 | 0.21 | 0.29 | 0.20h | 0.19 |
| 2/3rd LR | 0.08 | 0.13 | 0.16 | 0.12d | 0.09 | 0.14 | 0.20 | 0.15i | 0.13 |
| LR | 0.05 | 0.09 | 0.14 | 0.09e | 0.06 | 0.10 | 0.16 | 0.11j | 0.10 |
| Mean | 0.09aa | 0.15bb | 0.20cc | - | 0.10dd | 0.20dd | 0.25dd | - | - |
| LSD (0.01) | Lime = 0.002, Zn level = 0.005, FYM level = 0.004, Lime x Zn level =0.008,  Lime x FYM level = 0.007, Zn level x FYM level = 0.012 | | | | | | | | |

Debatoli series

| | Dry matter (g pot$^{-1}$) | | | | | | | | |
|---|---|---|---|---|---|---|---|---|---|
| No lime | 2.84 | 3.55 | 4.19 | 3.53a | 3.45 | 3.72 | 3.44 | 3.57d | 3.55 |
| 1/10$^{th}$ LR | 3.37 | 3.94 | 4.52 | 3.94b | 3.56 | 4.06 | 4.21 | 3.91d | 3.93 |
| 1/3$^{rd}$ LR | 3.71 | 4.32 | 4.54 | 4.19b | 3.80 | 4.84 | 4.46 | 4.37d | 4.28 |
| 2/3$^{rd}$ LR | 3.55 | 3.67 | 4.43 | 3.88b | 3.53 | 3.74 | 3.76 | 3.68d | 3.78 |
| LR | 3.27 | 3.54 | 3.46 | 3.42c | 3.46 | 3.59 | 3.55 | 3.54d | 3.48 |
| Mean | 3.35aa | 3.80bb | 4.23cc | - | 3.56dd | 3.99dd | 3.88dd | - | - |
| LSD (0.01) | Lime = 0.32, Zn level = 0.22, FYM level = ns, Lime x Zn level =0. 58,  Lime x FYM level = ns, Zn level x FYM level =ns | | | | | | | | |
| | Zn concentration (mg kg$^{-1}$) | | | | | | | | |
| No lime | 62.2 | 85.0 | 119 | 88.7a | 71.0 | 86.2 | 126 | 94.4f | 91.6 |
| 1/10$^{th}$ LR | 60.4 | 78.4 | 105 | 81.3b | 70.7 | 84.3 | 116 | 90.3g | 85.8 |
| 1/3$^{rd}$ LR | 55.3 | 68.9 | 94.8 | 73.0c | 71.6 | 77.3 | 97.9 | 82.3h | 77.6 |
| 2/3$^{rd}$ LR | 47.8 | 66.5 | 75.2 | 63.2d | 52.4 | 69.5 | 80.2 | 67.4i | 65.3 |
| LR | 39.7 | 60.6 | 64.8 | 55.0e | 44.8 | 62.6 | 70.6 | 59.4j | 57.2 |
| Mean | 53.1aa | 71.9bb | 91.8cc | - | 62.1dd | 76.0ee | 98.1ff | - | - |
| LSD (0.01) | Lime = 1.80, Zn level = 0.20, FYM level = 1.50, Lime x Zn level =2.10,  Lime x FYM level = 3.80, Zn level x FYM level = 2.10 | | | | | | | | |
| | Zn uptake (mg pot$^{-1}$) | | | | | | | | |
| No lime | 0.18 | 0.30 | 0.50 | 0.33a | 0.25 | 0.32 | 0.44 | 0.34e | 0.33 |
| 1/10$^{th}$ LR | 0.20 | 0.31 | 0.47 | 0.33a | 0.24 | 0.34 | 0.49 | 0.36f | 0.34 |
| 1/3$^{rd}$ LR | 0.21 | 0.30 | 0.43 | 0.31b | 0.27 | 0.37 | 0.44 | 0.36f | 0.34 |
| 2/3$^{rd}$ LR | 0.17 | 0.24 | 0.33 | 0.25c | 0.19 | 0.26 | 0.30 | 0.25g | 0.25 |
| LR | 0.13 | 0.21 | 0.23 | 0.19d | 0.15 | 0.23 | 0.25 | 0.21h | 0.20 |
| Mean | 0.18aa | 0.27bb | 0.39cc | - | 0.22dd | 0.30ee | 0.38ff | - | - |
| LSD (0.01) | Lime = 0.03, Zn level = 0.11, FYM level = 0.02, Lime x Zn level =ns,  Lime x FYM level = 0.08, Zn level x FYM level = ns | | | | | | | | |

*Letters indicate observed differences among the means of different treatments

**T able 3** Soil pH, EC and OC content as influence by FYM, lime and Zn application.

| Treatments | No FYM | | | | FYM (10 t ha$^{-1}$) | | | | Overall mean |
|---|---|---|---|---|---|---|---|---|---|
| | No Zn | 2.5 mg Zn kg$^{-1}$ | 5.0 mg Zn kg$^{-1}$ | Mean | No Zn | 2.5 mg Zn kg$^{-1}$ | 5.0 mg Zn kg$^{-1}$ | Mean | Overall mean |
| Hariharapur series | | | | | | | | | |
| | | | | | pH | | | | |
| No lime | 4.56 | 4.57 | 4.61 | 4.58a | 5.16 | 5.10 | 5.34 | 5.20f | 4.89 |
| 1/10$^{th}$ LR | 4.80 | 5.01 | 4.83 | 4.88b | 5.46 | 5.42 | 5.44 | 5.44f | 5.16 |
| 1/3$^{rd}$ LR | 5.69 | 6.14 | 5.57 | 5.80c | 5.93 | 6.49 | 5.97 | 6.13g | 5.97 |
| 2/3$^{rd}$ LR | 6.45 | 6.53 | 6.62 | 6.53d | 6.92 | 7.08 | 6.57 | 6.86h | 6.70 |
| LR | 7.23 | 7.25 | 6.99 | 7.16e | 7.37 | 7.17 | 7.38 | 7.31h | 7.23 |
| Mean | 5.75aa | 5.90aa | 5.72aa | - | 6.17bb | 6.25bb | 6.14bb | - | - |
| LSD (0.01) | Lime = 0.19, Zn level = ns, FYM level = 0.25, Lime x Zn level = ns, Lime x FYM level = 0.51, Zn level x FYM level = ns | | | | | | | | |
| | | | | | EC (dS m$^{-1}$) | | | | |
| No lime | 0.14 | 0.11 | 0.13 | 0.13a | 0.13 | 0.15 | 0.14 | 0.14a | 0.13 |
| 1/10$^{th}$ LR | 0.14 | 0.10 | 0.10 | 0.12a | 0.15 | 0.11 | 0.12 | 0.13a | 0.12 |
| 1/3$^{rd}$ LR | 0.13 | 0.13 | 0.11 | 0.12a | 0.12 | 0.10 | 0.14 | 0.12a | 0.12 |
| 2/3$^{rd}$ LR | 0.12 | 0.13 | 0.11 | 0.12a | 0.12 | 0.15 | 0.10 | 0.12a | 0.12 |
| LR | 0.13 | 0.14 | 0.12 | 0.13a | 0.15 | 0.14 | 0.15 | 0.15a | 0.14 |
| Mean | 0.13aa | 0.12aa | 0.11aa | - | 0.13aa | 0.13aa | 0.13aa | - | 0.13 |
| LSD (0.01) | Lime = ns, Zn level = ns, FYM level = ns, Lime x Zn level = ns, Lime x FYM level = ns, Zn level x FYM level = ns | | | | | | | | |
| | | | | | OC (%) | | | | |
| No lime | 0.26 | 0.27 | 0.25 | 0.26a | 0.32 | 0.37 | 0.34 | 0.34b | 0.30 |
| 1/10$^{th}$ LR | 0.27 | 0.24 | 0.27 | 0.26a | 0.33 | 0.34 | 0.39 | 0.35b | 0.31 |
| 1/3$^{rd}$ LR | 0.25 | 0.24 | 0.27 | 0.25a | 0.31 | 0.36 | 0.37 | 0.35b | 0.30 |
| 2/3$^{rd}$ LR | 0.27 | 0.25 | 0.23 | 0.25a | 0.30 | 0.34 | 0.32 | 0.32b | 0.29 |
| LR | 0.24 | 0.21 | 0.22 | 0.22a | 0.25 | 0.34 | 0.33 | 0.31b | 0.27 |
| Mean | 0.26aa | 0.24aa | 0.25aa | - | 0.30bb | 0.35bb | 0.35bb | - | - |
| LSD (0.01) | Lime = ns, Zn level = ns, FYM level = 0.03, Lime x Zn level = ns, Lime x FYM level = ns, Zn level x FYM level = ns | | | | | | | | |

Debatoli series pH

| | | | | | | | | | |
|---|---|---|---|---|---|---|---|---|---|
| No lime | 5.88 | 5.85 | 5.77 | 5.83a | 6.14 | 6.17 | 6.45 | 6.25f | 6.04 |
| 1/10[th] LR | 5.93 | 5.88 | 5.94 | 5.92b | 6.28 | 6.42 | 6.56 | 6.42f | 6.17 |
| 1/3[rd] LR | 6.38 | 6.21 | 6.21 | 6.27c | 6.44 | 6.57 | 6.58 | 6.53f | 6.40 |
| 2/3[rd] LR | 6.64 | 6.67 | 6.6 | 6.64d | 6.76 | 6.75 | 6.65 | 6.73g | 6.68 |
| LR | 6.96 | 6.99 | 6.9 | 6.95e | 7.27 | 6.87 | 7.14 | 7.09g | 7.02 |
| Mean | 6.36aa | 6.32aa | 6.28aa | - | 6.58bb | 6.56bb | 6.67bb | - | - |

LSD (0.01)  Lime = 0.17, Zn level = ns, FYM level = 0.20, Lime x Zn level = ns,  Lime x FYM level = 0.47, Zn level x FYM level = ns

EC (dS m$^{-1}$)

| | | | | | | | | | |
|---|---|---|---|---|---|---|---|---|---|
| No lime | 0.23 | 0.22 | 0.27 | 0.24a | 0.21 | 0.26 | 0.23 | 0.23a | 0.24 |
| 1/10[th] LR | 0.27 | 0.27 | 0.23 | 0.25a | 0.21 | 0.23 | 0.20 | 0.21a | 0.24 |
| 1/3[rd] LR | 0.23 | 0.23 | 0.24 | 0.23a | 0.17 | 0.29 | 0.25 | 0.24a | 0.24 |
| 2/3[rd] LR | 0.23 | 0.21 | 0.21 | 0.21a | 0.23 | 0.19 | 0.24 | 0.22a | 0.22 |
| LR | 0.24 | 0.17 | 0.29 | 0.23a | 0.19 | 0.30 | 0.26 | 0.25a | 0.24 |
| Mean | 0.24aa | 0.22aa | 0.25aa | - | 0.20aa | 0.25aa | 0.24aa | - | - |

LSD (0.01)  Lime = ns, Zn level = ns, FYM level = 0.04, Lime x Zn level = ns,  Lime x FYM level = ns, Zn level x FYM level = ns

OC (%)

| | | | | | | | | | |
|---|---|---|---|---|---|---|---|---|---|
| No lime | 0.21 | 0.28 | 0.22 | 0.24a | 0.22 | 0.29 | 0.30 | 0.27b | 0.25 |
| 1/10[th] LR | 0.22 | 0.22 | 0.21 | 0.22a | 0.28 | 0.28 | 0.28 | 0.28b | 0.25 |
| 1/3[rd] LR | 0.21 | 0.25 | 0.24 | 0.23a | 0.28 | 0.26 | 0.29 | 0.28b | 0.26 |
| 2/3[rd] LR | 0.18 | 0.22 | 0.25 | 0.21a | 0.31 | 0.25 | 0.28 | 0.28b | 0.25 |
| LR | 0.21 | 0.25 | 0.26 | 0.24a | 0.28 | 0.30 | 0.28 | 0.28b | 0.26 |
| Mean | 0.21aa | 0.24aa | 0.24aa | - | 0.27bb | 0.27bb | 0.29bb | - | - |

LSD (0.01)  Lime = ns, Zn level = ns, FYM level = 0.04, Lime x Zn level = ns,  Lime x FYM level = ns, Zn level x FYM level = ns

*Letters indicate observed differences among the means of different treatments

**T able 4** Effect of FYM, lime and Zn application on extractable Zn in soils.

| Treatments | No FYM | | | | FYM (10 t ha$^{-1}$) | | | | Overall mean |
|---|---|---|---|---|---|---|---|---|---|
| | No Zn | 2.5 mg Zn kg$^{-1}$ | 5.0 mg Zn kg$^{-1}$ | Mean | No Zn | 2.5 mg Zn kg$^{-1}$ | 5.0 mg Zn kg$^{-1}$ | Mean | |
| Hariharapur series | | | | | | | | | |
| | | | | DTPA-Zn (mg kg$^{-1}$) | | | | | |
| No lime | 0.40 | 1.44 | 2.95 | 1.60a | 0.88 | 1.68 | 3.21 | 1.92f | 1.76 |
| 1/10$^{th}$ LR | 0.40 | 1.24 | 2.30 | 1.31b | 0.66 | 1.67 | 3.20 | 1.84f | 1.58 |
| 1/3$^{rd}$ LR | 0.38 | 1.06 | 1.64 | 1.03c | 0.61 | 1.62 | 2.68 | 1.64fh | 1.33 |
| 2/3$^{rd}$ LR | 0.37 | 0.86 | 1.45 | 0.89d | 0.44 | 1.59 | 2.55 | 1.53gh | 1.21 |
| LR | 0.34 | 0.77 | 1.25 | 0.79e | 0.44 | 1.27 | 2.53 | 1.41gh | 1.10 |
| Mean | 0.38aa | 1.08bb | 1.92cc | - | 0.61dd | 1.57ee | 2.83ff | - | - |
| LSD (0.01) | Lime = 0.02, Zn level = 0.25, FYM level = 0.20, Lime x Zn level =0. 35, Lime x FYM level = 0.28, Zn level x FYM level = 0.47 | | | | | | | | |
| | | | | Mehlich 1-Zn (mg kg$^{-1}$) | | | | | |
| No lime | 0.78 | 1.68 | 3.85 | 2.10a | 1.23 | 3.70 | 5.08 | 3.34f | 2.72 |
| 1/10$^{th}$ LR | 0.77 | 1.66 | 3.74 | 2.06b | 1.17 | 3.20 | 4.88 | 3.08f | 2.57 |
| 1/3$^{rd}$ LR | 0.74 | 1.50 | 3.27 | 1.84c | 1.05 | 2.64 | 4.79 | 2.83gi | 2.33 |
| 2/3$^{rd}$ LR | 0.66 | 1.48 | 2.26 | 1.47d | 1.03 | 2.54 | 4.49 | 2.69hi | 2.08 |
| LR | 0.51 | 1.24 | 1.92 | 1.22e | 0.94 | 2.54 | 4.25 | 2.58hi | 1.90 |
| Mean | 0.69aa | 1.51bb | 3.01cc | - | 1.09dd | 2.92ee | 4.70ff | - | - |
| LSD (0.01) | Lime = 0.10, Zn level = 0.42, FYM level = 0.25, Lime x Zn level =0. 55, Lime x FYM level = 0.37, Zn level x FYM level = 0.70 | | | | | | | | |
| | | | | 0.1 M HCl-Zn (mg kg$^{-1}$) | | | | | |
| No lime | 0.90 | 2.50 | 4.62 | 2.67a | 1.50 | 3.81 | 6.24 | 3.85f | 3.26 |
| 1/10$^{th}$ LR | 0.89 | 2.31 | 4.61 | 2.60b | 1.34 | 3.72 | 6.20 | 3.75fi | 3.18 |
| 1/3$^{rd}$ LR | 0.84 | 2.25 | 4.28 | 2.46c | 1.33 | 3.39 | 5.68 | 3.47gi | 2.96 |
| 2/3$^{rd}$ LR | 0.84 | 2.18 | 3.94 | 2.32d | 1.22 | 3.05 | 5.62 | 3.30g | 2.81 |
| LR | 0.84 | 1.93 | 3.91 | 2.23e | 1.06 | 3.03 | 5.43 | 3.17g | 2.70 |

| | | | | | | | | | |
|---|---|---|---|---|---|---|---|---|---|
| Mean | 0.86aa | 2.23bb | 4.27cc | - | 1.29dd | 3.40ee | 5.83ff | - | - |
| LSD (0.01) | Lime = 0.02, Zn level = 0.30, FYM level = 0.27, Lime x Zn level =0. 37, Lime x FYM level = 0.30, Zn level x FYM level = 0.60 | | | | | | | | |

ABDTPA-Zn (mg kg$^{-1}$)

| | | | | | | | | | |
|---|---|---|---|---|---|---|---|---|---|
| No lime | 0.71 | 2.03 | 4.06 | 2.27a | 1.16 | 2.54 | 3.98 | 2.56f | 2.42 |
| 1/10$^{th}$ LR | 0.68 | 1.98 | 3.19 | 1.95b | 1.11 | 2.43 | 3.92 | 2.49f | 2.22 |
| 1/3$^{rd}$ LR | 0.59 | 1.70 | 2.62 | 1.64c | 1.00 | 2.43 | 3.84 | 2.42f | 2.03 |
| 2/3$^{rd}$ LR | 0.52 | 1.52 | 2.29 | 1.44d | 0.95 | 2.37 | 3.61 | 2.31f | 1.88 |
| LR | 0.49 | 1.25 | 2.12 | 1.29e | 0.93 | 2.21 | 3.31 | 2.15f | 1.72 |
| Mean | 0.60aa | 1.70bb | 2.85cc | - | 1.03dd | 2.40ee | 3.73ff | - | - |
| LSD (0.01) | Lime = 0.05, Zn level = 0.28, FYM level = 0.32, Lime x Zn level =0. 32, Lime x FYM level = 0.41, Zn level x FYM level = 0.62 | | | | | | | | |

Debatoli series

DTPA-Zn (mg kg$^{-1}$)

| | | | | | | | | | |
|---|---|---|---|---|---|---|---|---|---|
| No lime | 1.45 | 2.62 | 3.29 | 2.45a | 1.63 | 2.80 | 4.33 | 2.92f | 2.69 |
| 1/10$^{th}$ LR | 1.30 | 2.32 | 2.93 | 2.18b | 1.37 | 2.54 | 4.01 | 2.64fh | 2.41 |
| 1/3$^{rd}$ LR | 1.08 | 1.94 | 2.91 | 1.98bd | 1.32 | 2.37 | 3.79 | 2.49fh | 2.24 |
| 2/3$^{rd}$ LR | 0.99 | 1.78 | 2.80 | 1.86cd | 1.08 | 2.25 | 2.95 | 2.09gh | 1.98 |
| LR | 0.77 | 1.72 | 2.73 | 1.74c | 0.99 | 2.21 | 2.48 | 1.89g | 1.82 |
| Mean | 1.12aa | 2.08bb | 2.93cc | - | 1.28dd | 2.43ee | 3.51ff | - | - |
| LSD (0.01) | Lime = 0.21, Zn level = 0.50, FYM level = 0.35, Lime x Zn level =0. 75, Lime x FYM level = 0.78, Zn level x FYM level = 0.98 | | | | | | | | |

Mehlich 1-Zn (mg kg$^{-1}$)

| | | | | | | | | | |
|---|---|---|---|---|---|---|---|---|---|
| No lime | 1.73 | 3.61 | 6.78 | 4.04a | 2.64 | 4.78 | 6.78 | 4.73a | 4.39 |
| 1/10$^{th}$ LR | 1.63 | 3.60 | 6.59 | 3.94b | 2.44 | 4.20 | 6.28 | 4.31b | 4.12 |
| 1/3$^{rd}$ LR | 1.51 | 3.44 | 6.12 | 3.69c | 2.42 | 4.10 | 6.21 | 4.24b | 3.97 |
| 2/3$^{rd}$ LR | 1.49 | 3.33 | 4.13 | 2.98d | 2.40 | 4.06 | 5.69 | 4.05b | 3.52 |
| LR | 1.26 | 3.15 | 4.06 | 2.82e | 2.37 | 3.74 | 5.46 | 3.86b | 3.34 |
| Mean | 1.53aa | 3.43bb | 5.54cc | - | 2.45dd | 4.18ee | 6.08ff | - | - |
| LSD (0.01) | Lime = 0.09, Zn level = 0.50, FYM level = 0.28, Lime x Zn level =0. 45, Lime x FYM level = 0.42, Zn level x FYM level = 0.85 | | | | | | | | |

**0.1 M HCl-Zn (mg kg$^{-1}$)**

| | | | | | | | | | |
|---|---|---|---|---|---|---|---|---|---|
| No lime | 2.35 | 4.26 | 4.66 | 3.76a | 2.80 | 4.54 | 6.69 | 4.68e | 4.22 |
| 1/10th LR | 2.32 | 4.42 | 5.34 | 4.03b | 2.75 | 4.70 | 6.93 | 4.79e | 4.41 |
| 1/3rd LR | 2.22 | 4.40 | 6.07 | 4.23c | 2.86 | 5.25 | 7.61 | 5.24f | 4.74 |
| 2/3rd LR | 2.23 | 3.87 | 7.46 | 4.52d | 2.91 | 5.14 | 7.01 | 5.02f | 4.77 |
| LR | 2.22 | 4.53 | 6.96 | 4.57d | 2.85 | 6.06 | 7.79 | 5.57g | 5.07 |
| Mean | 2.27aa | 4.30bb | 6.10cc | - | 2.83dd | 5.14ee | 7.21ff | - | - |
| LSD (0.01) | Lime = 0.06, Zn level = 0.35, FYM level = 0.37, Lime x Zn level =0. 45, Lime x FYM level = 0.45, Zn level x FYM level = 0.79 | | | | | | | | |

**ABDTPA-Zn (mg kg$^{-1}$)**

| | | | | | | | | | |
|---|---|---|---|---|---|---|---|---|---|
| No lime | 2.10 | 3.19 | 4.23 | 3.18a | 2.12 | 3.34 | 5.17 | 3.54e | 3.36 |
| 1/10th LR | 1.82 | 3.46 | 4.19 | 3.16a | 1.98 | 3.37 | 5.89 | 3.75e | 3.46 |
| 1/3rd LR | 1.61 | 2.77 | 4.60 | 2.99b | 1.93 | 3.46 | 5.17 | 3.52e | 3.26 |
| 2/3rd LR | 1.36 | 2.05 | 5.12 | 2.84c | 1.75 | 3.02 | 4.26 | 3.01f | 2.93 |
| LR | 1.22 | 2.17 | 4.22 | 2.54d | 1.53 | 3.36 | 4.42 | 3.10f | 2.82 |
| Mean | 1.62aa | 2.73bb | 4.47cc | - | 1.86dd | 3.31ee | 4.98ff | - | - |
| LSD (0.01) | Lime = 0.10, Zn level = 0.35, FYM level = 0.20, Lime x Zn level =0. 47, Lime x FYM level = 0.40, Zn level x FYM level = 0.70 | | | | | | | | |

*Letters indicate observed differences among the means of different treatments

**Table 5** Pearson's correlation coefficient values revealing relationship among soil properties, dry matter yield, Zn concentration, Zn uptake and extracted Zn in soils (n = 90).

| | pH | EC | OC | Dry matter yield | Zn conc. | Zn uptake | DTPA-Zn | Mehlich 1-Zn | 0.1 M HCl-Zn | ABDTPA-Zn |
|---|---|---|---|---|---|---|---|---|---|---|
| | | | | | Hariharapur series | | | | | |
| pH | 1 | | | | | | | | | |
| EC | 0.058 | 1 | | | | | | | | |
| OC | -0.089 | -0.084 | 1 | | | | | | | |
| Dry matter yield | 0.059 | 0.093 | 0.221* | 1 | | | | | | |
| Zn conc. | -0.590** | -0.029 | 0.232* | 0.047 | 1 | | | | | |
| Zn uptake | -0.397** | 0.036 | 0.294** | 0.605** | 0.792** | 1 | | | | |
| DTPA-Zn | 0.010 | -0.073 | 0.211* | 0.391** | 0.610** | 0.523** | 1 | | | |
| Mehlich 1-Zn | 0.130 | -0.045 | 0.272** | 0.281** | 0.510** | 0.545** | 0.897** | 1 | | |
| 0.1 M HCl-Zn | 0.046 | -0.076 | 0.242* | 0.260* | 0.633** | 0.626** | 0.871** | 0.929** | 1 | |
| ABDTPA-Zn | -0.011 | -0.013 | 0.136 | 0.285** | 0.656** | 0.673** | 0.887** | 0.922** | 0.923** | 1 |
| | | | | | Debatoli series | | | | | |
| pH | 1 | | | | | | | | | |
| EC | 0.032 | 1 | | | | | | | | |
| OC | 0.113 | -0.098 | 1 | | | | | | | |
| Dr matter yield | -0.154 | 0.096 | 0.011 | 1 | | | | | | |
| Zn conc. | -0.343** | 0.042 | 0.158 | 0.384** | 1 | | | | | |
| Zn uptake | -0.326** | 0.086 | 0.110 | 0.727** | 0.905** | 1 | | | | |
| DTPA-Zn | -0.087 | 0.061 | 0.290** | 0.133 | 0.741** | 0.715** | 1 | | | |
| Mehlich 1-Zn | 0.168 | 0.091 | 0.317** | 0.330** | 0.589** | 0.568** | 0.811** | 1 | | |
| 0.1 M HCl-Zn | 0.188 | 0.130 | 0.294** | 0.333** | 0.562** | 0.545** | 0.822** | 0.937** | 1 | |
| ABDTPA-Zn | -0.074 | 0.108 | 0.193 | 0.419** | 0.772** | 0.748** | 0.889** | 0.890** | 0.887** | 1 |

*p ≤ 0.05; **p ≤ 0.01.

Fig. 1. Dry matter yield, Zn concentration and Zn uptake by maize as influenced by interaction of Zn application and lime rate in Hariharapur and Debatoli series. Error bars represent ± SE.

Fig. 2. Extractable Zn by different extractants as influenced by interaction of Zn application and lime rate in Hariharapur and Debatoli series. Error bars represent ± SE.

[Figure]

[Figure]

[Figure]

**(d)**

[Figure]

**(e)**

[Figure]

Fig. 1. Dry matter yield, Zn concentration and Zn uptake by maize as influenced by interaction of Zn application and lime rate in Hariharapur and Debatoli series. Error bars represent ± SE.

**(a)**

[Figure]

**(b)**

[Figure]

**(c)**

[Figure]

**(d)**

[Figure]

Fig. 2. Extractable Zn by different extractants as influenced by interaction of Zn application and lime rate in Hariharapur and Debatoli series. Error bars represent ± SE.

---

## Author Comment (AC2) · 11 Nov 2016

Dear Sir,

We thank you very much for providing constructive and useful suggestion for our manuscript. We have modified the manuscript as per the given suggestions. The details of our responses and revisions are given below.

Comment: My overall assessment of the revised version of this manuscript, published on 07/10/2016, is that it addresses most of the questions raised by the reviewers, albeit some minor ones remain.

Our response: Thank you very much sire for your observation. We have also addressed minor problems as indicated by you and corrected the manuscript accordingly.

Comment: First I list those points raised by both reviewers that, in my view, have been address properly. 1- Make a clear definition of the situation and objectives of the experiment to avoid the impression that the manuscript presents three related (but not properly C1 SOILD Interactive comment Printer-friendly version Discussion paper coordinated) experiments. The changes made by the authors with a new version of introduction and M&Methods, including clearly stated objectives have addressed properly this issue. 2- The need for a better description of the experiments and methods, among them a better description of the pots and manure used. This has also been properly addressed. 3- Elimination of duplication of results presented in Tables and Graphs. This has also been properly addressed with the elimination of some Figures. The fourth, and major, issue raised by both reviewers was the need for a much improved presentation and discussion of the results. From reviewer 1 I quote, among some of them, "a) Irrelevant results were included (e.g., adding farmyard manure increased the soil OC, the addition of lime increased soil pH, ..., adding Zn (and FYM) to soil increased Zn concentration in plant)" b) "No critical levels of Zn in soil and/or plant tissues were indicated". c) "Was the concentration of Zn in plants for unfavorable treatment below the critical values (literature)? Was there observed Zn deficiency symptoms in the plants with lower Zn concentration? –" d) "The Tables do not clarify the results of statistical analysis (comparison of means). The differences observed between means of the different treatments should be indicated by adding the corresponding letter (a, b, c...) to each mean value" I also quote some of the major comments by reviewer 2. "a) To evaluate the interaction between the two variables (FYM, and lime dose) in the statistical model" b) "Been more critical when extrapolating optimum lime application from initial stages of the crop. 60 days in a 4 l pot, to an adult plant exploring a larger soil volume not considered in their experiment" c) "Use same symbol in the same soil to facilitate identification to reader in Figures."

Our response: We do agree with you.

Comment: Many of these have been addressed but a few of them remain problematic. These issues can be summarized in: a- The statistical results presented in Tables 2 to 4 remain

difficult to understand. A better explanation of the statistical model used (a three factors analysis of variance with interaction among these factors by pairs as it seems reading the Tables?) should be included in the material and methods section (section 2.3) and also in an improved, more comprehensive, caption for these Tables.

Our response: We do agree with you. We have incorporated it in materials and method section (section 2.3) of the manuscript. We have modified the caption of the tables as suggested.

Comment: In its present form it is clear how to interpretate the difference between means for any given FYM and soil according to the LR and the Zinc levels. However it remains complicated to understand the statistical significance (or not) of the different treatments, and their interactions. For instance, in Table 2 Hariharapur soil series, dry matter results, what is the interpretation of the LSD(0.01) for Lime x FYM level = 0.61 in your results. Is it a significant or a non-significant interaction across the experimental results? This need be revised.

Our response: We have modified the results section of the manuscript by incorporating the information about the interaction effects of lime, Zn and FYM applications. A value for LSD (0.01) is mentioned in the table for significant results and "ns" is mentioned for non-significant results. Accordingly we have described the results. The interaction effect of Lime x FYM level on dry matter is significant having LSD (0.01) value 0.61 for Hariharapur series soil but the same is not significant for Debatoli series soil. We have incorporated this information in the manuscript.

Comment:  Also indicate in the caption that double letters (e.g. aa, bb and so on) are used as single symbol for the mean values across the Zinc levels.

Our response: We have incorporated it in the captions of the tables.

 Comment: 2- Graphs remain non-intuitive. Please try to use a more straightforward design. For instance use the same color in all the lines, use the symbol to identify the soil series. I mean the same symbol for the same soil series (e.g. solid square for Hariharapur and non-solid circle for Debatoli) and use the line style to differentiate between added or non-added FYM (e.g. continuous for added FYM and dotted for non-added FYM). Using for different colors and symbols makes more complicated a quick interpretation of the graphs.

Our response: We agree with you sir and we have modified the graphs as per the suggestion.

 Comment: 3- Please in the conclusions be a bit more cautious about the need to extrapolate these results to field recommendations.

Our response: we do agree with you and we have modified the manuscript as per comments given in the PDF version of the manuscript.

Comment: 4- Some misspellings and sections that should be double checked for proper English editing remain. I have added same comments in the PDF version of the manuscript in order to help you to deal with these final issues, and also to improve the edition in English .

Our response: We have modified the manuscript as per the comments/suggestion given in the manuscript. We have also gone through the manuscript and double checked the English and cross checked the references. We have rectified the existing problems therein.

Comment: So my recommendation is that this revised version merits publication in SOIL but after the authors revise this, now minor changes, before final publication.

Our response: Thank you very much sir. We have revised the manuscript as per the suggestions.

With above modifications, we are hereby submitting the revised manuscript for your kind perusal and consideration.

With kind regards,

Sanjib Kumar Behera

**Effect of lime, farmyard manure and zinc application on soil properties, dry matter yield, zinc concentration and uptake by maize and extractable zinc in Alfisols**

**Sanjib K. Behera[1], Arvind K. Shukla[2], Brahma S. Dwivedi[3], Brij L. Lakaria[2]**

[1]ICAR-Indian Institute of Oil Palm Research, Pedavegi, West Godavari District, Andhra Pradesh 534450, India
[2]ICAR-Indian Institute of Soil Science, Nabibagh, Berasia Road, Bhopal, Madhya Pradesh 462038, India
[3]ICAR-Indian Agricultural Research Institute, Pusa, New Delhi, 110012, India

*Correspondence to*: Sanjib K. Behera (sanjibkumarbehera123@gmail.com)

**Abstract.** Zinc (Zn) deficiency is widespread in all types of soils of world including acid soils affecting crop production and nutritional quality of edible plant parts. There is, however, limited information available regarding effects of lime, farmyard manure (FYM) and Zn addition to acid soils on dry matter yield, Zn concentration and uptake by maize (*Zea mays* L.), soil properties and extractable Zn by different extractants. Green house pot experiments were carried out in two acid soils to study the effect of five levels of lime (0, 1/10 lime requirement (LR), 1/3 LR, 2/3 LR and LR), three levels of Zn concentration (0, 2.5 and 5.0 mg Zn kg$^{-1}$ soil) and two levels of FYM (0 and 10 t ha$^{-1}$) addition on dry matter yield, Zn concentration and uptake by maize plant grown up to 60 days, soil pH, EC and OC content and extractable Zn in soil. Lime rate of 1/3$^{rd}$ LR was found to be optimum as dry matter yield of maize increased significantly with lime application up to 1/3$^{rd}$ LR in soils of both the series and decreased subsequently. Addition of FYM with and without lime increased dry matter yield. Application of Zn up to 5.0 mg kg$^{-1}$ to soil increased dry matter yield with and without FYM application in soils of Hariharapur series. Addition of higher doses of lime significantly reduced Zn concentration in maize crop grown in soils of both the series. Mean Zn uptake values were at par for no lime, 1/10$^{th}$ LR and 1/3$^{rd}$ LR with and without FYM application and it was significantly higher than Zn uptake by 2/3$^{rd}$ LR and LR treatments. However, FYM application improved Zn uptake by maize crop. Increased level of lime

application reduced Zn extracted by DTPA, Mehlich 1, 0.1 N HCl and ABDTPA extractants. However, application of FYM along with lime improved Zn extraction. The amount of Zn extracted by different extractants followed the order DTPA-Zn < ABDTPA-Zn < Mehlich-1 Zn < 0.1 M HCl. Zn extracted by different extractants like DTPA, ABDTPA, Mehlich 1 and 0.1 M HCl was positively and significantly correlated amongst themselves and with dry matter yield, Zn concentration and Zn uptake by maize. Among the extractants, ABDTPA was found to be the best extractant for extraction of Zn in acid soils.

*Keywords:* Alfisol, Dry matter yield, Farmyard manure, Lime, Zinc concentration

**1 Introduction**

[revised manuscript text omitted]

The information from the present study would be useful for assessment of extractable Zn and its management in acid soils where Zn availability is one of the main problems and Zn application is imminent and application of lime and FYM is a common practice. Keeping above facts in view, the present study was carried out (i) to evaluate the influence of lime, FYM and Zn addition on dry matter yield, Zn concentration and uptake by maize (*Zea mays* L.) crop and (ii) to evaluate the influence of lime, FYM and Zn addition to acid soils on soil pH, EC and OC content, extractable Zn as evaluated by different extractants.

**2  Material and methods**

**2.1 Soil and farmyard manure characteristics**

The bulk surface (0-15 cm depth) soils collected from Hariharpur series (Oxic Haplustalf, Alfisol (Soil Survey Staff, 2014)) and Debatoli series (Udic Rhodostalf, Alfisol (Soil Survey Staff, 2014)) of Bhubaneswar and Ranchi (India), respectively were used in the study. The collected soils were air dried and stone and debris were removed and then ground to pass a 2 mm sieve and analysed for selected properties (Table 1). Soil properties like pH and EC were measured in 1:2.5 (w/v) soil-water suspensions using pH meter and EC meter following half an hour equilibrium (Jackson, 1973). Soil organic carbon (OC) content was estimated by

chromic acid digestion-back titration method (Walkley and Black, 1934). The clay, silt and sand per cent of soils were determined by hydrometer method (Bouyoucos, 1962). Calcium carbonate ($CaCO_3$) content was determined by rapid titration method (Puri, 1930) and cation exchange capacity (CEC) by neutral normal ammonium acetate method (Richards, 1954). Lime requirement (LR) of the soil was estimated by extractant buffer method (Shoemaker et al., 1961). The plant available Zn in soils was extracted by DTPA method (Lindsay and Norvell, 1978). Estimation of Zn concentration was done on the clear extract by atomic absorption spectrophotometer (AAS). After drying of FYM at 70 $^o$C for 24 h followed by grinding to pass through 20 mesh sieve, one gram of ground FYM was dry-ashed at 450 $^o$C for 2h. Ashed samples were extracted using 0.5 N HCl. Zn concentration was determined in filtered extracts. The total OC (loss on ignition), N (Kjeldahl method), P (nitric-perchloric 9:4 digestion) and K (nitric-perchloric 9:4 digestion) concentrations in FYM were estimated according to Tandon (2009) (Table 1).

*2.2 Green house study, soil and plant analysis*

Pot experiments were carried out in two Hariharapur and Debatoli series soils. The experiments were carried out in plastic pots (each with diameter of 20 cm) having 4 kg of soil with five levels of LR (0, 1/10 LR, 1/3 LR, 2/3 LR and LR), three levels of Zn concentration (0, 2.5 and 5.0 mg Zn kg$^{-1}$ soil) and two levels of fresh FYM (35% moisture) (0 and 4.5 g FYM kg$^{-1}$ soil viz., 0 and 10 t FYM ha$^{-1}$). Locally available FYM was used for the study and it was decomposed mixture of left over fodder fed to farm animals, animal dung and urine. All the pots received basal treatments of N-$P_2O_5$-$K_2O$ @ 150-60-40 kg ha$^{-1}$ (equivalent to 66.7-26.7-17.8 mg N-$P_2O_5$-$K_2O$ kg$^{-1}$ soil, respectively). Fertilizer N, P and K were applied through analytical grade urea, calcium dihydrogen orthophosphate and muriate of potash, respectively. Lime and Zn were added to soil through laboratory grade $CaCO_3$ and $ZnSO_4$ respectively. All nutrients were mixed in soil thoroughly before sowing of seeds. The soil in

each pot was then irrigated to field capacity with deionized water and kept for incubation for one week. Each treatment combination was replicated thrice in a factorial completely randomized design. Four seeds of cv. KH 101 of maize were sown in each pot. Two seedlings of maize per each pot were maintained after emergence. Pots were irrigated with water daily as per requirement of water on weight basis to maintain the field capacity. Above-ground biomass of plants from each pot was harvested at the end of 60 days of growth.

Harvested above-ground biomass of each pot was washed in deionized water, and then dried in oven at 70 $^{o}$C for 48 h. After drying, dry matter yield (DMY) of each pot was recorded. Dried plant material was then ground in a stainless steel Wiley mill, and digested in a di-acid mixture of $HNO_3$ and $HClO_4$ (Jackson, 1973). Zn concentration was then determined in aqueous extracts of the digested plant material by atomic absorption spectrophotometer (AAS). Zn uptake was calculated as DMY multiplied by the Zn concentration.

Soil sample from each pot were collected after harvesting of maize plants. Collected soil samples were processed and analyzed for pH, EC, OC content and DTPA-Zn concentration following the methods described above. The plant available Zn in soils was also extracted by DTPA (Lindsay and Norvell, 1978), Mehlich 1 (Perkins, 1970), 0.1 M HCl (Sorensen et al., 1971) and ABDTPA (Soltanpour and Schwab, 1977) extractants by following the respective prescribed methods. Estimation of Zn concentration was done on the clear extract by AAS.

*2.3 Statistical analysis*

The data regarding soil properties, DMY, Zn concentration, Zn uptake and extracted Zn by different extractants subjected to analysis of variance method (Gomez and Gomez, 1984). A three factors analysis of variance with interaction among these factors by pairs was used for the study. Least square difference (LSD) at $P \leq .01$ was used to compare among the treatment means. Pearson's correlation coefficient values were estimated to establish relationship

among soil properties, DMY, Zn concentration, Zn uptake and extracted Zn by different extractants.

**3 Results**

**3.1 Dry matter yield**

DMY of maize increased significantly with lime application up to $1/3^{rd}$ LR (Table 2, Fig. 1 a) in soils of both the series. This indicated that lime application @ $1/3^{rd}$ of LR was optimum for these soils in the early stages of the crop. Application of higher doses of lime ($2/3^{rd}$ LR and LR) did not result in increased DMY. However, this finding needs to be verified by conducting field experiment. The mean DMY in $1/3^{rd}$ LR treatment without FYM and with FYM was 139% and 149% of control respectively in Harihpur series soils. Similarly in Debatoli series soil, the mean DMY was 84% and 120% of control without and with FYM application respectively in combination with $1/3^{rd}$ LR. Application of graded doses of Zn upto 5.0 mg kg$^{-1}$ to soil increased DMY with and without FYM application in Hariharapur series. Whereas in Debatoli series, application of graded doses of Zn up to 5 mg kg$^{-1}$ without FYM and application of Zn @ 2.5 mg kg$^{-1}$ with FYM enhanced DMY. Interaction effects of lime, Zn and FYM application were significant in Hariharapur series soil. Whereas, interaction effect of lime and Zn application was significant in Debatoli series soil.

**3.2 Zinc concentration and uptake by maize**

Addition of higher doses of lime significantly reduced Zn concentration in maize crop grown in soils of both the series (Table 2, Fig. 1 b). In contrast, application of Zn (@ 2.5 and 5.0 mg kg$^{-1}$) and FYM (@ 10 t ha$^{-1}$) increased Zn concentration in maize crop significantly in soils of both the series (Table 2, Fig 1c). In soils of Hariharapur series, application Zn @ 2.5 and 5 mg kg$^{-1}$ without and with FYM augmented Zn concentration in maize by 67.5 and 93.5 to 109 % respectively, as compared to control (No Zn). Similarly, increased Zn concentrations of 22

to 35 and 58 to 73% were recorded with application of Zn @ 2.5 and 5 mg kg$^{-1}$ without and with FYM respectively in comparison to no Zn control in soils of Debatoli series. However, the Zn concentration in maize under all the treatments were well above the critical Zn concentration of 15 to 22 mg kg$^{-1}$ for maize crop (Alloway, 2008) and no visual Zn deficiency symptoms in plants were recorded. Zn concentration in maize crop was significantly influenced by interaction effect of lime, Zn and FYM application in soils of both the series. Mean Zn uptake values were at par for no lime, 1/10$^{th}$ LR and 1/3$^{rd}$ LR with and without FYM application and it was significantly higher than Zn uptake by 2/3$^{rd}$ LR and LR treatments in soils of both the series (Table 2, Fig. 1 d). However, Zn and FYM application improved Zn uptake by maize crop in soils of both series (Fig. 1 e). Addition of Zn @ 2.5 and 5 mg kg$^{-1}$ enhanced Zn uptake by 67 to 100 and 122 to 150% respectively as compared to no Zn control in soils of Hariharapur series. Whereas, the enhancements in Zn uptake were 36 to50, 73 to 117% due to application of Zn @ 2.5 and 5 mg kg$^{-1}$ respectively as compared to no Zn control in soils of Debatoli series. Interaction effect of of lime, Zn and FYM application was significant on Zn uptake in Hariharpur series soil.

*3.3 Soil properties*

Application of lime at different rates significantly increased pH in soils of both Hariharapur and Debatoli series (Table 3). With addition of graded doses of limes viz. from no lime, 1/10$^{th}$ LR, 1/3$^{rd}$ LR, 2/3$^{rd}$ LR and LR, soil pH increased from 4.58 to 7.16 (without FYM addition) and from 4.89 to 7.23 (with FYM addition) in Hariharapur series and from 5.83 to 6.95 (without FYM addition) and from 6.04 to 7.02 (with FYM addition) in Debatoli series. Application of FYM without lime increased soil pH in both soils (Table 3). Interaction effect of combined application of lime and FYM on soil pH was significant. Soil pH values obtained by addition of 2/3$^{rd}$ LR and LR along with FYM were at par. Addition of Zn did not have any effect on soil pH. Sole application of lime, FYM and Zn and their interaction did

not influence soil EC levels in soils of both the series (Table 3). However application of FYM increased soil OC content in soils of both series. Addition of lime and Zn and their interaction did not influence soil OC as expected.

*3.4 Extractable zinc in post-harvest soil*

Data regarding amount Zn extracted by DTPA, Mehlich 1, 0.1 M HCl and ABDTPA extractants in post harvest soil are given in Table 4 and Figure 2. Perusal of data revealed significant effect of individual application of lime, FYM and Zn and their interaction on extracted Zn by different extractants. The amount of extracted Zn by DTPA, Mehlich 1, and ABDTPA extractants decreased with increased level of lime application in soils of both the series (Fig. 2 a, b, d). But addition of FYM (@ 10 t ha$^{-1}$) in combination of different levels of lime led to marked enhancement of extracted Zn by different extractants in both the soils compared to only application of different lime levels (Table 4). Application Zn at different levels viz. 2.5 and 5.0 mg kg$^{-1}$ with and without FYM increased the concentration of extracted Zn by the different extractants. The amount of Zn extracted by DTPA, Mehlich 1, 0.1 M HCl and ABDTPA extractant varied from 1.10 to 1.76, 1.90 to 2.72, 2.70 to 3.26 and 1.72 to 2.42 mg kg$^{-1}$ respectively, under different levels of lime application across FYM and Zn application in soils of Hariharpur series. Whereas, the Zn extracted by DTPA, Mehlich 1, 0.1 M HCl and ABDTPA extractant varied from 1.82 to 2.69, 3.34 to 4.39, 4.22 to 5.07 and 2.82 to 3.36 mg kg$^{-1}$ respectively, under different levels of lime application across FYM and Zn application in soils of Debatoli series. In both the series, the extracted Zn followed the order DTPA-Zn < ABDTPA-Zn < Mehlich1-Zn < 0.1 M HCl-Zn.

**4 Discussion**

Significant increase in DMY was recorded with application of lime up to 1/3$^{rd}$ LR. Increase in DMY with lime application up to 1/3$^{rd}$ LR may be ascribed to increase in soil pH and

positive influence on nutrient availability in soil (Tisdale, 2005). Our finding is in line with the observations made by Barman et al. (2014) who reported lime application at 1/3rd LR was optimum for obtaining cauliflower yield in Typic Fluvaquent soil of West Bengal, India. There was reduction in DMY with lime application at $2/3^{rd}$ LR and LR in soils of both the series. This may be ascribed to reduced availability Zn in soil with $2/3^{rd}$ LR and LR rate of lime application and adverse effect on other soil properties. This needs to be verified by conducting filed experiment. 
[revised manuscript text omitted]

**5   Conclusion**

Lime application of 1/3LR was found to be optimum for amelioration of the acid soils evaluated in the initial stages of the crop in pot experiments. This results merit confirmation in field conditions for the whole crop season. The concentration of Zn in maize tissue and extracted Zn by different extractants like DTPA, Mehlich 1, 0.1 M HCl and ABDTPA in both the soils reduced with lime application. Application of FYM along with lime improved the Zn concentration in maize plant and extractable Zn in soils. Since   DTPA, Mehlich 1, 0.1 M HCl and ABDTPA extractable Zn in soils of both the series were positively and significantly correlated with dry matter yield, Zn concentration and Zn uptake, these extractants could be used for extraction of Zn in acid soils. However based on higher correlation coefficient values, ABDTPA was found to be best extractant for extraction of Zn in acid soils.

**Acknowledgements.** The study was supported by the grant from Indian Council of Agricultural Research, New Delhi. We thank the Director of ICAR-Indian Institute of Soil Science, Bhopal, Madhya Pradesh, India for providing necessary facilities for conducting the research work. We acknowledge the help rendered by Ms. P. Singh, Mr. R. Singh and Mr. D. K. Verma during the execution of the work. The authors thank the topical editor and the anonymous reviewers for their useful suggestions for improvement of the manuscript.

**Table 1** Some selected characteristics of the experimental soils and farmyard manure.

| Characteristics | Experimental soils | |
|---|---|---|
| | Hariharapur series | Debatoli series |
| Taxonomic classification | Oxic Haplustalfs | Udic Rhodustalfs |
| pH (1:2.5) | 4.50 | 5.80 |
| EC (dS m$^{-1}$) | 0.14 | 0.23 |
| Organic carbon (%) | 0.31 | 0.22 |
| Clay (%) | 12.1 | 14.2 |
| Silt (%) | 15.0 | 11.6 |
| Sand (%) | 73.2 | 75.1 |
| CaCO$_3$ (%) | 20.0 | 32.0 |
| CEC (cmol(p$^+$) kg$^{-1}$) | 3.90 | 5.10 |
| Lime requirement (g kg$^{-1}$) | 3.34 | 1.51 |
| DTAP-Zn (mg kg$^{-1}$) | 0.47 | 1.45 |
| | Farmyard manure | |
| Total organic carbon (%) | 0.22 | |
| Total N (%) | 0.48 | |
| Total P (%) | 0.10 | |
| Total K (%) | 0.55 | |
| Total Zn (mg kg$^{-1}$) | 12 | |

*Critical concentration of DTPA-Zn is 0.60 mg kg$^{-1}$

**T able 2** Effects of FYM, lime and Zn application on dry matter yield, Zn concentration and Zn uptake by maize differences among the means of different treatments. Double letters (i.e. aa, bb and so on) are used as single sym the Zn levels. Symbol ns indicates non-significant result.

| Treatments | No FYM | | | | FYM ( | | |
|---|---|---|---|---|---|---|---|
| | No Zn | 2.5 mg Zn kg$^{-1}$ | 5.0 mg Zn kg$^{-1}$ | Mean | No Zn | 2.5 mg Zn kg$^{-1}$ | 5.0 mg Z |
| Hariharapur series | | | | | | | |
| | | | Dry matter (g pot$^{-1}$) | | | | |
| No lime | 1.64 | 2.02 | 2.04 | 1.90a | 2.06 | 2.60 | 2.2: |
| 1/10$^{th}$ LR | 2.43 | 2.37 | 2.16 | 2.32b | 2.21 | 2.74 | 2.6( |
| 1/3$^{rd}$ LR | 2.88 | 2.87 | 2.96 | 2.83c | 2.57 | 2.89 | 3.6( |
| 2/3$^{rd}$ LR | 2.65 | 2.37 | 2.66 | 2.64c | 2.40 | 2.40 | 3.0 |
| LR | 1.77 | 2.06 | 2.52 | 2.12ab | 1.94 | 2.05 | 2.7 |
| Mean | 2.27aa | 2.34aa | 2.47bb | - | 2.23cc | 2.53cce | 2.85d |
| LSD (0.01) | Lime = 0.30, Zn level = 0.11, FYM level = 0.25, Lime x Zn level =0. 50, Lime x FYM level = 0.61, Z | | | | | | |
| | | | Zn concentration (mg kg$^{-1}$) | | | | |
| No lime | 54.0 | 84.0 | 112 | 83.3a | 57.4 | 104 | 119 |
| 1/10$^{th}$ LR | 53.3 | 87.4 | 113 | 84.6a | 59.2 | 99.5 | 119 |
| 1/3$^{rd}$ LR | 38.5 | 63.5 | 75.0 | 59.0b | 46.3 | 72.8 | 80.( |
| 2/3$^{rd}$ LR | 27.4 | 52.7 | 60.8 | 47.0c | 35.4 | 59.8 | 67.( |
| LR | 25.2 | 44.8 | 54.2 | 41.4d | 31.2 | 48.9 | 58.1 |
| Mean | 39.7aa | 66.5bb | 83.0cc | - | 45.9dd | 76.9ee | 88.8: |
| LSD (0.01) | Lime = 3.50, Zn level = 0.11, FYM level = 2.00, Lime x Zn level =3.21, Lime x FYM level = 5.70, Z | | | | | | |
| | | | Zn uptake (mg pot$^{-1}$) | | | | |
| No lime | 0.11 | 0.14 | 0.23 | 0.16a | 0.12 | 0.27 | 0.2( |
| 1/10$^{th}$ LR | 0.13 | 0.21 | 0.24 | 0.19b | 0.13 | 0.27 | 0.3: |
| 1/3$^{rd}$ LR | 0.10 | 0.18 | 0.22 | 0.17c | 0.11 | 0.21 | 0.2! |
| 2/3$^{rd}$ LR | 0.08 | 0.13 | 0.16 | 0.12d | 0.09 | 0.14 | 0.2( |
| LR | 0.05 | 0.09 | 0.14 | 0.09e | 0.06 | 0.10 | 0.1( |

| | | | | | | | |
|---|---|---|---|---|---|---|---|
| Mean | 0.09aa | 0.15bb | 0.20cc | - | 0.10dd | 0.20dd | 0.25c |

LSD (0.01)    Lime = 0.002, Zn level = 0.005, FYM level = 0.004, Lime x Zn level =0.008,  Lime x FYM level = 0.0

Debatoli series

**Dry matter (g pot$^{-1}$)**

| | | | | | | | |
|---|---|---|---|---|---|---|---|
| No lime | 2.84 | 3.55 | 4.19 | 3.53a | 3.45 | 3.72 | 3.44 |
| 1/10$^{th}$ LR | 3.37 | 3.94 | 4.52 | 3.94b | 3.56 | 4.06 | 4.21 |
| 1/3$^{rd}$ LR | 3.71 | 4.32 | 4.54 | 4.19b | 3.80 | 4.84 | 4.46 |
| 2/3$^{rd}$ LR | 3.55 | 3.67 | 4.43 | 3.88b | 3.53 | 3.74 | 3.76 |
| LR | 3.27 | 3.54 | 3.46 | 3.42c | 3.46 | 3.59 | 3.55 |
| Mean | 3.35aa | 3.80bb | 4.23cc | - | 3.56dd | 3.99dd | 3.88c |

LSD (0.01)        Lime = 0.32, Zn level = 0.22, FYM level = ns, Lime x Zn level =0. 58,  Lime x FYM level = ns, Z

**Zn concentration (mg kg$^{-1}$)**

| | | | | | | | |
|---|---|---|---|---|---|---|---|
| No lime | 62.2 | 85.0 | 119 | 88.7a | 71.0 | 86.2 | 126 |
| 1/10$^{th}$ LR | 60.4 | 78.4 | 105 | 81.3b | 70.7 | 84.3 | 116 |
| 1/3$^{rd}$ LR | 55.3 | 68.9 | 94.8 | 73.0c | 71.6 | 77.3 | 97.9 |
| 2/3$^{rd}$ LR | 47.8 | 66.5 | 75.2 | 63.2d | 52.4 | 69.5 | 80.2 |
| LR | 39.7 | 60.6 | 64.8 | 55.0e | 44.8 | 62.6 | 70.6 |
| Mean | 53.1aa | 71.9bb | 91.8cc | - | 62.1dd | 76.0ee | 98.1 |

LSD (0.01)        Lime = 1.80, Zn level = 0.20, FYM level = 1.50, Lime x Zn level =2.10,  Lime x FYM level = 3.8

**Zn uptake (mg pot$^{-1}$)**

| | | | | | | | |
|---|---|---|---|---|---|---|---|
| No lime | 0.18 | 0.30 | 0.50 | 0.33a | 0.25 | 0.32 | 0.44 |
| 1/10$^{th}$ LR | 0.20 | 0.31 | 0.47 | 0.33a | 0.24 | 0.34 | 0.49 |
| 1/3$^{rd}$ LR | 0.21 | 0.30 | 0.43 | 0.31b | 0.27 | 0.37 | 0.44 |
| 2/3$^{rd}$ LR | 0.17 | 0.24 | 0.33 | 0.25c | 0.19 | 0.26 | 0.30 |
| LR | 0.13 | 0.21 | 0.23 | 0.19d | 0.15 | 0.23 | 0.25 |
| Mean | 0.18aa | 0.27bb | 0.39cc | - | 0.22dd | 0.30ee | 0.38 |

LSD (0.01)        Lime = 0.03, Zn level = 0.11, FYM level = 0.02, Lime x Zn level =ns,  Lime x FYM level = 0.08,

**T able 3** Effects of FYM, lime and Zn application on soil pH, EC and OC content. Letters indicate observed diff
different treatments. Double letters (i.e. aa, bb and so on) are used as single symbol for the mean values across th
non-significant result.

| Treatments | No FYM | | | | FYM (10 | | |
|---|---|---|---|---|---|---|---|
| | No Zn | 2.5 mg Zn kg$^{-1}$ | 5.0 mg Zn kg$^{-1}$ | Mean | No Zn | 2.5 mg Zn kg$^{-1}$ | 5.0 mg Z |
| Hariharapur series | | | | | | | |
| | | | | | pH | | |
| No lime | 4.56 | 4.57 | 4.61 | 4.58a | 5.16 | 5.10 | 5.3⁴ |
| 1/10$^{th}$ LR | 4.80 | 5.01 | 4.83 | 4.88b | 5.46 | 5.42 | 5.4⁴ |
| 1/3$^{rd}$ LR | 5.69 | 6.14 | 5.57 | 5.80c | 5.93 | 6.49 | 5.9⁷ |
| 2/3$^{rd}$ LR | 6.45 | 6.53 | 6.62 | 6.53d | 6.92 | 7.08 | 6.5⁷ |
| LR | 7.23 | 7.25 | 6.99 | 7.16e | 7.37 | 7.17 | 7.3⁸ |
| Mean | 5.75aa | 5.90aa | 5.72aa | - | 6.17bb | 6.25bb | 6.14⁠ᵇ |
| LSD (0.01) | Lime = 0.19, Zn level = ns, FYM level = 0.25, Lime x Zn level = ns, Lime x FYM level = 0.51, | | | | | | |
| | | | | | EC (dS m$^{-1}$) | | |
| No lime | 0.14 | 0.11 | 0.13 | 0.13a | 0.13 | 0.15 | 0.1⁴ |
| 1/10$^{th}$ LR | 0.14 | 0.10 | 0.10 | 0.12a | 0.15 | 0.11 | 0.1² |
| 1/3$^{rd}$ LR | 0.13 | 0.13 | 0.11 | 0.12a | 0.12 | 0.10 | 0.1⁴ |
| 2/3$^{rd}$ LR | 0.12 | 0.13 | 0.11 | 0.12a | 0.12 | 0.15 | 0.1⁠(⁠ |
| LR | 0.13 | 0.14 | 0.12 | 0.13a | 0.15 | 0.14 | 0.1⁵ |
| Mean | 0.13aa | 0.12aa | 0.11aa | - | 0.13aa | 0.13aa | 0.13⁠ₐ |
| LSD (0.01) | Lime = ns, Zn level = ns, FYM level = ns, Lime x Zn level = ns, Lime x FYM level = ns, Zn lev | | | | | | |
| | | | | | OC (%) | | |
| No lime | 0.26 | 0.27 | 0.25 | 0.26a | 0.32 | 0.37 | 0.3⁴ |
| 1/10$^{th}$ LR | 0.27 | 0.24 | 0.27 | 0.26a | 0.33 | 0.34 | 0.3⁹ |
| 1/3$^{rd}$ LR | 0.25 | 0.24 | 0.27 | 0.25a | 0.31 | 0.36 | 0.3⁷ |
| 2/3$^{rd}$ LR | 0.27 | 0.25 | 0.23 | 0.25a | 0.30 | 0.34 | 0.3² |
| LR | 0.24 | 0.21 | 0.22 | 0.22a | 0.25 | 0.34 | 0.3³ |

| | | | | | | | |
|---|---|---|---|---|---|---|---|
| Mean | 0.26aa | 0.24aa | 0.25aa | - | 0.30bb | 0.35bb | 0.35b |

LSD (0.01)   Lime = ns, Zn level = ns, FYM level = 0.03, Lime x Zn level = ns,  Lime x FYM level = ns, Zn l

Debatoli series

pH

| | | | | | | | |
|---|---|---|---|---|---|---|---|
| No lime | 5.88 | 5.85 | 5.77 | 5.83a | 6.14 | 6.17 | 6.4 |
| 1/10th LR | 5.93 | 5.88 | 5.94 | 5.92b | 6.28 | 6.42 | 6.56 |
| 1/3rd LR | 6.38 | 6.21 | 6.21 | 6.27c | 6.44 | 6.57 | 6.58 |
| 2/3rd LR | 6.64 | 6.67 | 6.6 | 6.64d | 6.76 | 6.75 | 6.65 |
| LR | 6.96 | 6.99 | 6.9 | 6.95e | 7.27 | 6.87 | 7.14 |
| Mean | 6.36aa | 6.32aa | 6.28aa | - | 6.58bb | 6.56bb | 6.67b |

LSD (0.01)     Lime = 0.17, Zn level = ns, FYM level = 0.20, Lime x Zn level = ns,  Lime x FYM level = 0.47,

EC (dS m$^{-1}$)

| | | | | | | | |
|---|---|---|---|---|---|---|---|
| No lime | 0.23 | 0.22 | 0.27 | 0.24a | 0.21 | 0.26 | 0.23 |
| 1/10th LR | 0.27 | 0.27 | 0.23 | 0.25a | 0.21 | 0.23 | 0.20 |
| 1/3rd LR | 0.23 | 0.23 | 0.24 | 0.23a | 0.17 | 0.29 | 0.25 |
| 2/3rd LR | 0.23 | 0.21 | 0.21 | 0.21a | 0.23 | 0.19 | 0.24 |
| LR | 0.24 | 0.17 | 0.29 | 0.23a | 0.19 | 0.30 | 0.26 |
| Mean | 0.24aa | 0.22aa | 0.25aa | - | 0.20aa | 0.25aa | 0.24a |

LSD (0.01)     Lime = ns, Zn level = ns, FYM level = 0.04, Lime x Zn level = ns,  Lime x FYM level = ns, Zn l

OC (%)

| | | | | | | | |
|---|---|---|---|---|---|---|---|
| No lime | 0.21 | 0.28 | 0.22 | 0.24a | 0.22 | 0.29 | 0.30 |
| 1/10th LR | 0.22 | 0.22 | 0.21 | 0.22a | 0.28 | 0.28 | 0.28 |
| 1/3rd LR | 0.21 | 0.25 | 0.24 | 0.23a | 0.28 | 0.26 | 0.29 |
| 2/3rd LR | 0.18 | 0.22 | 0.25 | 0.21a | 0.31 | 0.25 | 0.28 |
| LR | 0.21 | 0.25 | 0.26 | 0.24a | 0.28 | 0.30 | 0.28 |
| Mean | 0.21aa | 0.24aa | 0.24aa | - | 0.27bb | 0.27bb | 0.29b |

LSD (0.01)     Lime = ns, Zn level = ns, FYM level = 0.04, Lime x Zn level = ns, Lime x FYM level = ns, Zn l

**T able 4** Effects of FYM, lime and Zn application on extractable Zn in soils. Letters indicate observed difference treatments. Double letters (i.e. aa, bb and so on) are used as single symbol for the mean values across the Zn leve

| Treatments | No FYM | | | | FYM ( | | |
|---|---|---|---|---|---|---|---|
| | No Zn | 2.5 mg Zn kg$^{-1}$ | 5.0 mg Zn kg$^{-1}$ | Mean | No Zn | 2.5 mg Zn kg$^{-1}$ | 5.0 mg Z |
| Hariharapur series | | | | | | | |
| | | | | DTPA-Zn (mg kg$^{-1}$) | | | |
| No lime | 0.40 | 1.44 | 2.95 | 1.60a | 0.88 | 1.68 | 3.21 |
| 1/10$^{th}$ LR | 0.40 | 1.24 | 2.30 | 1.31b | 0.66 | 1.67 | 3.20 |
| 1/3$^{rd}$ LR | 0.38 | 1.06 | 1.64 | 1.03c | 0.61 | 1.62 | 2.68 |
| 2/3$^{rd}$ LR | 0.37 | 0.86 | 1.45 | 0.89d | 0.44 | 1.59 | 2.55 |
| LR | 0.34 | 0.77 | 1.25 | 0.79e | 0.44 | 1.27 | 2.53 |
| Mean | 0.38aa | 1.08bb | 1.92cc | - | 0.61dd | 1.57ee | 2.83 |
| LSD (0.01) | Lime = 0.02, Zn level = 0.25, FYM level = 0.20, Lime x Zn level =0. 35,  Lime x FYM level = 0. | | | | | | |
| | | | | Mehlich 1-Zn (mg kg$^{-1}$) | | | |
| No lime | 0.78 | 1.68 | 3.85 | 2.10a | 1.23 | 3.70 | 5.08 |
| 1/10$^{th}$ LR | 0.77 | 1.66 | 3.74 | 2.06b | 1.17 | 3.20 | 4.88 |
| 1/3$^{rd}$ LR | 0.74 | 1.50 | 3.27 | 1.84c | 1.05 | 2.64 | 4.79 |
| 2/3$^{rd}$ LR | 0.66 | 1.48 | 2.26 | 1.47d | 1.03 | 2.54 | 4.49 |
| LR | 0.51 | 1.24 | 1.92 | 1.22e | 0.94 | 2.54 | 4.25 |
| Mean | 0.69aa | 1.51bb | 3.01cc | - | 1.09dd | 2.92ee | 4.70 |
| LSD (0.01) | Lime = 0.10, Zn level = 0.42, FYM level = 0.25, Lime x Zn level =0. 55,  Lime x FYM level = 0. | | | | | | |
| | | | | 0.1 M HCl-Zn (mg kg$^{-1}$) | | | |
| No lime | 0.90 | 2.50 | 4.62 | 2.67a | 1.50 | 3.81 | 6.24 |
| 1/10$^{th}$ LR | 0.89 | 2.31 | 4.61 | 2.60b | 1.34 | 3.72 | 6.20 |
| 1/3$^{rd}$ LR | 0.84 | 2.25 | 4.28 | 2.46c | 1.33 | 3.39 | 5.68 |
| 2/3$^{rd}$ LR | 0.84 | 2.18 | 3.94 | 2.32d | 1.22 | 3.05 | 5.62 |

| | | | | | | | |
|---|---|---|---|---|---|---|---|
| LR | 0.84 | 1.93 | 3.91 | 2.23e | 1.06 | 3.03 | 5.4⁝ |
| Mean | 0.86aa | 2.23bb | 4.27cc | - | 1.29dd | 3.40ee | 5.8⁝ |
| LSD (0.01) | Lime = 0.02, Zn level = 0.30, FYM level = 0.27, Lime x Zn level =0. 37, Lime x FYM level = 0.⁝ | | | | | | |

ABDTPA-Zn (mg kg⁻¹)

| | | | | | | | |
|---|---|---|---|---|---|---|---|
| No lime | 0.71 | 2.03 | 4.06 | 2.27a | 1.16 | 2.54 | 3.9⁝ |
| 1/10th LR | 0.68 | 1.98 | 3.19 | 1.95b | 1.11 | 2.43 | 3.9⁝ |
| 1/3rd LR | 0.59 | 1.70 | 2.62 | 1.64c | 1.00 | 2.43 | 3.8⁝ |
| 2/3rd LR | 0.52 | 1.52 | 2.29 | 1.44d | 0.95 | 2.37 | 3.6⁝ |
| LR | 0.49 | 1.25 | 2.12 | 1.29e | 0.93 | 2.21 | 3.3⁝ |
| Mean | 0.60aa | 1.70bb | 2.85cc | - | 1.03dd | 2.40ee | 3.7⁝ |
| LSD (0.01) | Lime = 0.05, Zn level = 0.28, FYM level = 0.32, Lime x Zn level =0. 32, Lime x FYM level = 0.⁝ | | | | | | |

Debatoli series

DTPA-Zn (mg kg⁻¹)

| | | | | | | | |
|---|---|---|---|---|---|---|---|
| No lime | 1.45 | 2.62 | 3.29 | 2.45a | 1.63 | 2.80 | 4.3⁝ |
| 1/10th LR | 1.30 | 2.32 | 2.93 | 2.18b | 1.37 | 2.54 | 4.0⁝ |
| 1/3rd LR | 1.08 | 1.94 | 2.91 | 1.98bd | 1.32 | 2.37 | 3.7⁝ |
| 2/3rd LR | 0.99 | 1.78 | 2.80 | 1.86cd | 1.08 | 2.25 | 2.9⁝ |
| LR | 0.77 | 1.72 | 2.73 | 1.74c | 0.99 | 2.21 | 2.4⁝ |
| Mean | 1.12aa | 2.08bb | 2.93cc | - | 1.28dd | 2.43ee | 3.5⁝ |
| LSD (0.01) | Lime = 0.21, Zn level = 0.50, FYM level = 0.35, Lime x Zn level =0. 75, Lime x FYM level = 0.⁝ | | | | | | |

Mehlich 1-Zn (mg kg⁻¹)

| | | | | | | | |
|---|---|---|---|---|---|---|---|
| No lime | 1.73 | 3.61 | 6.78 | 4.04a | 2.64 | 4.78 | 6.7⁝ |
| 1/10th LR | 1.63 | 3.60 | 6.59 | 3.94b | 2.44 | 4.20 | 6.2⁝ |
| 1/3rd LR | 1.51 | 3.44 | 6.12 | 3.69c | 2.42 | 4.10 | 6.2⁝ |
| 2/3rd LR | 1.49 | 3.33 | 4.13 | 2.98d | 2.40 | 4.06 | 5.6⁝ |
| LR | 1.26 | 3.15 | 4.06 | 2.82e | 2.37 | 3.74 | 5.4⁝ |
| Mean | 1.53aa | 3.43bb | 5.54cc | - | 2.45dd | 4.18ee | 6.0⁝ |
| LSD (0.01) | Lime = 0.09, Zn level = 0.50, FYM level = 0.28, Lime x Zn level =0. 45, Lime x FYM level = 0.⁝ | | | | | | |

|  | | | | | 0.1 M HCl-Zn (mg kg$^{-1}$) | | |
|---|---|---|---|---|---|---|---|
| No lime | 2.35 | 4.26 | 4.66 | 3.76a | 2.80 | 4.54 | 6.69 |
| 1/10$^{th}$ LR | 2.32 | 4.42 | 5.34 | 4.03b | 2.75 | 4.70 | 6.93 |
| 1/3$^{rd}$ LR | 2.22 | 4.40 | 6.07 | 4.23c | 2.86 | 5.25 | 7.61 |
| 2/3$^{rd}$ LR | 2.23 | 3.87 | 7.46 | 4.52d | 2.91 | 5.14 | 7.01 |
| LR | 2.22 | 4.53 | 6.96 | 4.57d | 2.85 | 6.06 | 7.79 |
| Mean | 2.27aa | 4.30bb | 6.10cc | - | 2.83dd | 5.14ee | 7.21 |
| LSD (0.01) | Lime = 0.06, Zn level = 0.35, FYM level = 0.37, Lime x Zn level =0. 45, Lime x FYM level = 0.4 | | | | | | |
|  | | | | | ABDTPA-Zn (mg kg$^{-1}$) | | |
| No lime | 2.10 | 3.19 | 4.23 | 3.18a | 2.12 | 3.34 | 5.17 |
| 1/10$^{th}$ LR | 1.82 | 3.46 | 4.19 | 3.16a | 1.98 | 3.37 | 5.89 |
| 1/3$^{rd}$ LR | 1.61 | 2.77 | 4.60 | 2.99b | 1.93 | 3.46 | 5.17 |
| 2/3$^{rd}$ LR | 1.36 | 2.05 | 5.12 | 2.84c | 1.75 | 3.02 | 4.26 |
| LR | 1.22 | 2.17 | 4.22 | 2.54d | 1.53 | 3.36 | 4.42 |
| Mean | 1.62aa | 2.73bb | 4.47cc | - | 1.86dd | 3.31ee | 4.98 |
| LSD (0.01) | Lime = 0.10, Zn level = 0.35, FYM level = 0.20, Lime x Zn level =0. 47, Lime x FYM level = 0.4 | | | | | | |

**Table 5** Pearson's correlation coefficient values revealing relationship among soil properties, dry matter yield, Z extracted Zn in soils (n = 90).

| | pH | EC | OC | Dry matter yield | Zn conc. | Zn uptake | DTPA-Zn | Mehlich |
|---|---|---|---|---|---|---|---|---|
| | | | | | Hariharapur series | | | |
| pH | 1 | | | | | | | |
| EC | 0.058 | 1 | | | | | | |
| OC | -0.089 | -0.084 | 1 | | | | | |
| Dry matter yield | 0.059 | 0.093 | 0.221* | 1 | | | | |
| Zn conc. | -0.590** | -0.029 | 0.232* | 0.047 | 1 | | | |
| Zn uptake | -0.397** | 0.036 | 0.294** | 0.605** | 0.792** | 1 | | |
| DTPA-Zn | 0.010 | -0.073 | 0.211* | 0.391** | 0.610** | 0.523** | 1 | |
| Mehlich 1-Zn | 0.130 | -0.045 | 0.272** | 0.281** | 0.510** | 0.545** | 0.897** | 1 |
| 0.1 M HCl-Zn | 0.046 | -0.076 | 0.242* | 0.260* | 0.633** | 0.626** | 0.871** | 0.929** |
| ABDTPA-Zn | -0.011 | -0.013 | 0.136 | 0.285** | 0.656** | 0.673** | 0.887** | 0.922** |
| | | | | | Debatoli series | | | |
| pH | 1 | | | | | | | |
| EC | 0.032 | 1 | | | | | | |
| OC | 0.113 | -0.098 | 1 | | | | | |
| Dr matter yield | -0.154 | 0.096 | 0.011 | 1 | | | | |
| Zn conc. | -0.343** | 0.042 | 0.158 | 0.384** | 1 | | | |
| Zn uptake | -0.326** | 0.086 | 0.110 | 0.727** | 0.905** | 1 | | |
| DTPA-Zn | -0.087 | 0.061 | 0.290** | 0.133 | 0.741** | 0.715** | 1 | |
| Mehlich 1-Zn | 0.168 | 0.091 | 0.317** | 0.330** | 0.589** | 0.568** | 0.811** | 1 |
| 0.1 M HCl-Zn | 0.188 | 0.130 | 0.294** | 0.333** | 0.562** | 0.545** | 0.822** | 0.937** |
| ABDTPA-Zn | -0.074 | 0.108 | 0.193 | 0.419** | 0.772** | 0.748** | 0.889** | 0.890** |

*p ≤ 0.05; **p ≤ 0.01.

Fig. 1.  Dry matter yield, Zn concentration and Zn uptake by maize as influenced by interaction of Zn application and lime rate in Hariharapur and Debatoli series. Error bars represent ± SE.

Fig. 2.  Extractable Zn by different extractants as influenced by interaction of Zn application and lime rate in Hariharapur and Debatoli series. Error bars represent ± SE.

[Figure]

[Figure]

[Figure]

**(d)**

[Figure]

**(e)**

[Figure]

Fig. 1. Dry matter yield, Zn concentration and Zn uptake by maize as influenced by interaction of Zn application and lime rate in Hariharapur and Debatoli series. Error bars represent ± SE.

[Figure]

[Figure]

[Figure]

[Figure]

Fig. 2. Extractable Zn by different extractants as influenced by interaction of Zn application and lime rate in Hariharapur and Debatoli series. Error bars represent ± SE.

---

## Editor Comment (EC3) · J. A. Gomez (Editor) · 4 Dec 2016

With the improvements added by the authors in the latest version of the manuscriot, they have addreesed all the questions raised during the revision process. My recomendation is that manuscript should be accepted for final publication in SOIL